# A Unified Solution for Privacy and Communication Efficiency in Vertical Federated Learning

**Ganyu Wang**
Western University
gwang382@uwo.ca

**Bin Gu** *
Jilin University and MBZUAI
jsgubin@gmail.com

**Qingsong Zhang**
Xidian University
qszhang1995@gmail.com

**Xiang Li**
Western University
lixiang41@126.com

**Boyu Wang**
Western University
bwang@csd.uwo.ca

**Charles X. Ling** *
Western University
charles.ling@uwo.ca

## Abstract

Vertical Federated Learning (VFL) is a collaborative machine learning paradigm that enables multiple participants to jointly train a model on their private data without sharing it. To make VFL practical, privacy security and communication efficiency should both be satisfied. Recent research has shown that Zero-Order Optimization (ZOO) in VFL can effectively conceal the internal information of the model without adding costly privacy protective add-ons, making it a promising approach for privacy and efficiency. However, there are still two key problems that have yet to be resolved. First, the convergence rate of ZOO-based VFL is significantly slower compared to gradient-based VFL, resulting in low efficiency in model training and more communication round, which hinders its application on large neural networks. Second, although ZOO-based VFL has demonstrated resistance to state-of-the-art (SOTA) attacks, its privacy guarantee lacks a theoretical explanation. To address these challenges, we propose a novel cascaded hybrid optimization approach that employs a zeroth-order (ZO) gradient on the most critical output layer of the clients, with other parts utilizing the first-order (FO) gradient. This approach preserves the privacy protection of ZOO while significantly enhancing convergence. Moreover, we theoretically prove that applying ZOO to the VFL is equivalent to adding Gaussian Mechanism to the gradient information, which offers an implicit differential privacy guarantee. Experimental results demonstrate that our proposed framework achieves similar utility as the Gaussian mechanism under the same privacy budget, while also having significantly lower communication costs compared with SOTA communication-efficient VFL frameworks.

## 1 Introduction

Federated Learning [30, 22, 23, 18, 31] is an emerging technique that has raised wide attention recently. It has become one of the most important distributed learning frameworks for enabling multiple data holders to train a model collaboratively without sharing their local privacy data explicitly.

---

*Co-corresponding authors

37th Conference on Neural Information Processing Systems (NeurIPS 2023).

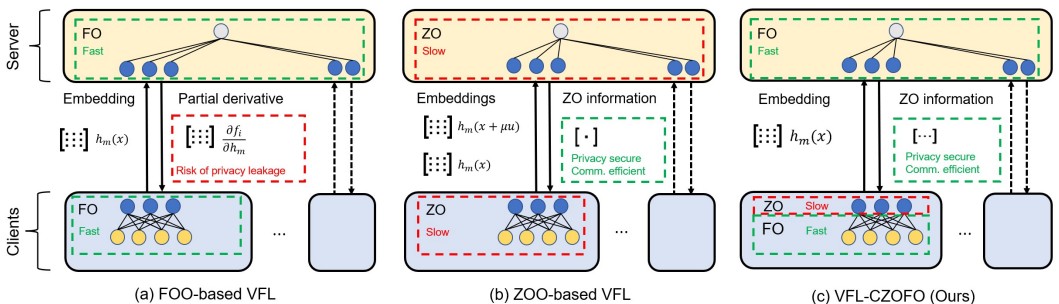

Figure 1: Comparing the Asyn-VFL Frameworks

Our research focuses on VFL, where each client possesses all the data points, but only a non-intersecting subset of the features (vertically distributed). In VFL, all participants collaborate to train a single global model. The client trains a feature extraction model that maps its local data sample to embeddings, and the server collects the embeddings from all clients to make a prediction [21, 34, 6, 39, 16, 37, 14, 42, 43, 13].

Ensuring both privacy and computation-communication efficiency is crucial for practical VFL implementations. One common approach to achieve this is starting with a basic VFL [5, 6, 28] and subsequently applying privacy protection techniques, such as Secure Multiparty Computation (SMC) [8], Differential Privacy (DP) [32, 36], and Homomorphic Encryption (HE) [36, 32] to protect privacy. However, the addition of these protection techniques often increases computation-communication costs. To avoid these costly protective add-ons, research attention has turned to Zero-Order Optimization (ZOO). Literature reports that ZOO-based VFL is able to conceal the internal information of the model (gradient/model parameter) making the SOTA privacy inference attack ineffective [41]. This finding highlights the potential of ZOO for efficient and secure VFL.

However, two critical problems remained. First, the convergence rate of ZOO-based VFL is significantly slower [41] compared to VFL frameworks optimized with gradient, resulting in a high number of communication rounds and increased communication costs. Furthermore, the convergence rate is negatively correlated to the size of the parameters optimized using ZOO [41, 27, 12], which hinders the application of ZOO-based VFL on large neural networks. Second, although ZO-based VFL has demonstrated resistance to SOTA attacks, its privacy guarantee has not been theoretically explained.

To solve the first problem, we reevaluate the advantages and disadvantages of applying first-order (FO) or zero-order (ZO) gradient methods for each component of the VFL, and find an optimal balance between efficiency and privacy protection. This analysis leads to a novel cascaded hybrid optimization framework that efficiently solves the convergence problems while fully preserving privacy protection. Specifically, our proposed framework (VFL-CZOFO) utilizes the ZO gradient for optimizing the most critical output layer of the clients, while other parts are optimized with the FO gradient[2]. To address the second problem, we theoretically prove that applying ZOO on VFL is equivalent to adding Gaussian Mechanism [7] to the gradient information and can provide a corresponding intrinsic $(\epsilon, \delta)$-DP guarantee.

For a clear illustration, a schematic diagram of our proposed framework is presented in Figure 1-(c). The advantage of our framework is two-fold. First, it greatly alleviates the slow convergence problem of ZOO-based VFL (Figure 1-b), where our framework's convergence is solely limited by the output size of the clients, rather than the parameter size of the entire model. Second, compared with the FOO-based VFL (Figure 1-a), the connection layer of our framework used ZOO, allowing the backward messages to be solely losses, thus inheriting the privacy protection of the ZOO-based VFL.

The primary innovations of our paper are as follows: 1) Our framework employs a novel cascaded hybrid optimization method, in which different optimization methods are applied to different layers of the global model in each iteration, which significantly improves the convergence rate of ZOO-based VFL while preserving privacy. To the best of our knowledge, no prior research in VFL has proposed a similar method that cascaded a hybrid optimization and exploits its advantages. 2) We provide

---

[2]Additionally, we apply compression to further reduce the communication cost of our framework.

a theoretical explanation of the intrinsic privacy guarantee of ZOO, based on $(\epsilon, \delta)$-DP, which is fundamentally different from the common DP mechanism, where noise is added to items afterward.

In summary, the contributions of our paper are:

- We propose the cascade hybrid optimization method in VFL, where ZOO is applied to the most essential part of the VFL, which provides intrinsic privacy protection and significantly improves the convergence rate compared with the ZOO-based VFL.

- Theoretically analysis shows: 1) the intrinsic $(\epsilon, \delta)$-differential privacy guarantee provided by ZOO within our framework, 2) the convergence of our framework and its superiority compared with the ZOO-based VFL, 3) the compatibility of communication compression for both forward and backward messages within our framework.

- Extensive experiments show: 1) with the same $(\epsilon, \delta)$-differential privacy guarantee, our method can achieve a similar utility as the Gaussian Mechanism, 2) Our method significantly reduces the communication cost compared with the SOTA communication-efficient VFL.

## 2 A Detailed Comparison with SOTA VFL Frameworks

Table 1 presents a comparison of our VFL framework with other SOTA VFL frameworks, including "split-learning" [34], "compressed-VFL" [5], "Syn-ZOO-VFL"[3], "VAFL" [6], "ZOO-VFL" [41]. In the table, Asyn/Syn are abbreviations for asynchronous and synchronous. The $T$ is the iteration of the model. $d$ and $d_h$ represent the size of the entire global model's parameter and the size of the client's output embeddings, respectively. The "communication size per iteration" column summarizes the number of elements in the original messages before any post-processing, such as compression. $B$ is the batch size. $q$ is the sampling times for multiple point estimation of ZOO [27].

Compared to the FOO-based VFL [34, 5, 6] our framework has two distinct advantages. First, our framework leverages the intrinsic privacy protection of ZOO, which is a result of its stochastic characteristics and the concealment of internal information. Second, we leverage the advantage of ZOO to reduce the communication cost from the server to the client where the backward message consists of merely $q$ elements, which is typically significantly smaller than the FO-based VFL with $d_h B$ elements. The key difference between our framework and the FOO-based framework is that our approach reduces communication costs while simultaneously ensuring privacy protection. In contrast, FOO-based VFL typically shares internal information (parameters/gradient) [6, 5] and then applies for extra privacy protection mechanism [36, 32].

Compared with the ZOO-based VFL [41], this framework and ours both leverage the privacy protection of ZOO in VFL. However, ZOO-based VFL suffers from a slow convergence rate of $\mathcal{O}(d/\sqrt{T})$ which is constrained by the global model parameter size $d$. This hinders its effectiveness when dealing with "larger" networks containing millions of parameters, causing higher communication costs due to increased communication rounds. To address this issue, we applied ZOO to the crucial connection layer between the server and the client. By doing so, we maintained the privacy protection offered by ZOO while significantly mitigating the slow convergence problem, reducing the constraint from the entire global model size $d$ to merely the output size of the clients $d_h$.

Table 1: Compare with Different VFL Algorithm

| Framework | Privacy | Asyn/Syn | Optimization | Convergence $\mathcal{O}(\cdot)$ | Comm. Size per Iter. (Forward + Backward) |
|---|---|---|---|---|---|
| Split learning [34] | ✗ | Syn | FO | $1/\sqrt{T}$ | $d_h B + d_h B$ |
| Compressed-VFL [5] | ✗ | Syn | FO | $1/\sqrt{T}$ | $d_h B + d_h B$ |
| VAFL [6] | ✗ | Asyn | FO | $1/\sqrt{T}$ | $d_h B + d_h B$ |
| Syn-ZOO-VFL | ✓ | Syn | ZO | $d/\sqrt{T}$ | $2d_h B + 1$ |
| ZOO-VFL [41] | ✓ | Asyn | ZO | $d/\sqrt{T}$ | $2d_h B + 1$ |
| VFL-CZOFO (Ours) | ✓ | Asyn | ZO&FO | $d_h/\sqrt{T}$ | $d_h B + q$ |

---

[3]This is the synchronous version of ZOO-VFL, the algorithm is in Appendix E.1.

## 3 Method

**Problem Definition**   In the VFL framework, there is one server and $M$ clients[4]. The server holds the label $y_i$ for each sample $i \in [n]$, ($[n] \triangleq \{1, 2 \cdots n\}$), while each client holds a non-intersecting feature set for all the samples. Specifically, the features for sample $i$ on client $m$ are denoted as $x_{m,i}$. Client $m$'s model, $h_m(w_m; x_{m,i})$, is parameterized by $w_m$ and takes local feature $x_{m,i}$ as input, outputting the embedding. The server inputs the embeddings from all clients into its own model, which is parameterized by $w_0$, and then calculates the losses for updating the parameters. The VFL framework can be viewed as solving a finite-sum problem in composite form:

$$f(w_0, \mathbf{w}; X, Y) = \frac{1}{n} \sum_{i=1}^{n} [\underbrace{\mathcal{L}(w_0, h_{1,i}, \ldots, h_{M,i}; y_i)}_{f_i(w_0, h_{1,i}, \ldots, h_{M,i})}] \text{ with } h_{m,i} = h_m(w_m; x_{m,i}) \quad \forall m \in [M]$$

(1)

where $\mathbf{w} = [w_1, \cdots w_M]$ denotes the parameter for all clients, $X$ denotes the entire feature set across all clients, and $y$ denotes the labels for all samples. For notation brevity, we define the outputs from all clients as $\Phi_i(\mathbf{w}) = [h_1(w_1; x_{1,i}), \cdots, h_M(w_M; x_{M,i})] = [h_{1,i}, \ldots, h_{M,i}]$. Therefore, the loss for the $i$-th sample[5] can be represented as $f_i(w_0, \Phi)$. $\mathcal{L}(\cdot; y_{true})$ is the loss function.

**Cascaded and Minimal Application of ZOO on the Connection Layer**   Applying ZOO on VFL resolves the privacy leakage issue from the gradient [41], but simply applying ZOO to the entire VFL has a slow convergence problem, leading to a high number of communication rounds. Specifically, the major problem of ZOO is that with more dimensions on the gradient that need to be estimated, more variance will be introduced in the estimation of the gradient [27, 3]. This is especially problematic for machine learning scenarios where even relatively small models may have millions of parameters. To address the slow convergence problem of ZOO-based VFL, we precisely apply ZOO to the client's output layer, while the rest of the model utilizes the FO gradient. This approach significantly improves the convergence of the ZOO-based VFL, since only a small portion of the entire model uses ZOO.

ZOO [12] estimates the gradient of a function by sampling a random perturbation within the function's domain of definition and evaluating its shift on the value domain. To estimate the partial derivative with respect to the output layer of the client's model, we apply the average random gradient estimation (Avg-RandGradEst) [27, 25] which utilizes $q$ i.i.d. samples $\{u_{m,i}^j\}_{j=1}^q$ on the objective function:

$$\hat{\nabla}_{h_{m,i}} f_i(w_0, h_{1,i}, \cdots, h_{M,i}) = \frac{\phi(d_{h_m})}{q\mu_m} \sum_{j=1}^{q} [\underbrace{f_i(h_{m,i} + \mu_m u_{m,i}^j) - f_i(h_{m,i})}_{\delta_{m,i}^j}] u_{m,i}^j \qquad (2)$$

where $f_i(h_{m,i} + \mu_m u_{m,i})$ is the abbreviation for $f_i(w_0, h_{1,i}, \cdots h_{m,i} + \mu_m u_{m,i} \cdots, h_{M,i})$, $d_{h_m}$ is the size of the output layer of client $m$. $\phi(d_{h_m})$ is a dimension-dependent factor [26] related to the choice of the distribution $p$ of the random vector $u_{m,i}^j$. If $p$ is the multivariate uniform distribution $\mathcal{U}(\mathcal{S}(\mathbf{0}, 1))$ on a unit sphere centered at $\mathbf{0}$ with radius 1, then $\phi(d_{h_m}) = d_{h_m}$. If $p$ is the multivariate normal distribution $\mathcal{N}(\mathbf{0}, \mathbf{I})$, then $\phi(d_{h_m}) = 1$. $q$ is the sampling times. The clients cannot calculate $\delta_{m,i}^j$ by themselves because the loss function $f_i(\cdot)$ is held by the server, therefore the server has to send the $\delta_{m,i}^j$ back to the clients[6]. To implement multiple-point estimation, the server must transmit multiple distinct values of $\delta_{m,i}^j$ to the client, with the number of values sent equaling the number of sampling points $q$ used in the Avg-RandGradEst.

The theoretical difficulty in analyzing our framework is that different optimization methods are cascaded within a single model. Unlike most VFL research [34, 5, 28, 6, 41] which updates the global model with a unified optimization method, e.g. $\nabla f_i(w_m)$ (VAFL [6]) or the $\hat{\nabla} f_i(w_m)$ (ZOO-VFL [41]), we apply a cascaded hybrid gradient via chain rule, i.e. $\hat{\nabla}_{h_m} f_i(h_{m,i}) \cdot \nabla_{w_m} h_{m,i}$, where $\hat{\nabla}_{h_m} f_i(h_{m,i})$ is the stochastic gradient estimator of the partial derivative w.r.t. the output layer of the client, and $\nabla_{w_m} h_m(w_m)$ is the local gradient of the client.

---

[4]Server and clients are the roles in the framework, in practice, one participant may take more than one role.

[5]Although sample $i$ was used throughout this paper, it is easy to generalize our approach to a mini-batch $B$.

[6]The server and the client also need to share an identical random sequence $u_{m,i}$, but this can be achieved by sharing one random seed at the beginning of the training, or use the sample ID as the random seed.

**Compress the Communication**  Furthermore, we demonstrate that compression is compatible with our approach, and can be applied to all communications in the framework, including the forward message [5] and the backward message [35]. We define the compressor of the client as $\mathcal{C}_m(\cdot)$, and the compressor of the server as $\mathcal{C}_0(\cdot)$. In each communication round, the client $m$ sends compressed embeddings instead of the raw message to the server, and the server replies with the compressed messages. We apply the uniform scale compressor [2] in the experiment. However, other compression schemes such as Top-K [24], sign-SGD [4], and lattice quantization [40] can also be applied to the different parts of our framework to further reduce the message size.

The theoretical challenge in proving the convergence with compression lies in that two errors (forward and backward) are introduced in the analysis. The first error affects the embeddings sent from the client to the server, where the error influences the server's model input, creating uncertainty in the loss value and gradients for updating the parameter. We bound this error by imposing assumptions on the curvature of the loss function, i.e. an upper bound for the norm of the Hessian matrix. The second error affects the message sent from the server to the clients, which then updates using the hybrid gradient with compression error. As a result, there are three different sources of uncertainty in the clients' parameter update: the two mentioned above, as well as the error caused by ZOO.

**Algorithm**  In this section, we propose our asynchronous VFL[7] with ZOO on the connection layer and compression on the communication messages. Starting with the client, which randomly selects a sample $i$ and sends its local model output to the server. Receiving the forward message from the client $m$, the server calculates the $\delta_{m,i}^j$ with Avg-RandGradEst (Eq. 2) and group $\delta_{m,i}^j$ with $j = 1, 2, \cdots, q$ into a vector $\Delta_{m,i}$. The server compresses $\Delta_{m,i}$ and sends it back to the client. Then the server and client update their local model with the "gradient".

---

**Algorithm 1** VFL-CZOFO

**Input:** Learning rate $\eta_m$, smoothing parameter $\mu_m$, compressor $\mathcal{C}_m$ for $m \in \{0, 1, ...M\}$.
**Output:** Parameter $w_m$ for $m \in \{0, 1, ...M\}$.
  0: Initialize model parameter $w_m$ for all participants $m \in \{0, 1, ...M\}$
  1: **while** not convergent **do**
  2:    **when** a **client** $m$ is activated, **do**:
  3:        Randomly select a sample $x_{m,i}$
  4:        Compute and send $\mathcal{C}_m(h_m(w_m; x_{m,i}))$ to server
  5:        Receive $\mathcal{C}_0(\Delta_{m,i})$ from the server (in a listen manner)
  6:        Compute $\hat{\nabla}_{h_{m,i}} \hat{f}_i(w_0, \Phi_i)$ via Avg-RandGradEst (Eq. 2)
  7:        Compute $\nabla_{w_m} h_m(w_m; x_{m,i})$ via backpropagation.
  8:        Compute $v_m = \hat{\nabla}_{h_{m,i}} \hat{f}_i(w_0, \Phi_i) \cdot \nabla_{w_m} h_{m,i}$
  9:        Update $w_m \leftarrow w_m - \eta_m v_m$
10:    **when server** receives from client $m$, **do**:
11:        Compute $\delta_{m,i}^l$ and group into $\Delta_{m,i}$
12:        Send $\mathcal{C}_0(\Delta_{m,i})$ to the client $m$
13:        Compute $v_0 = \nabla_{w_0} f_i(w_0, \Phi_i)$, (FOO)
14:        Update $w_0 \leftarrow w_0 - \eta_0 v_0$
15: **end while**

---

## 4  Security Analysis

**Threat Model**  We discuss privacy under two scenarios: "honest-but-curious" and "honest-but-colluded". In both scenarios, all participants are assumed to follow the protocol and perform their assigned tasks. Under the "honest-but-curious" scenario, a curious client attempts to obtain private information from other participants using the information they have received. In the "honest-but-colluded" scenario, some participants collude to obtain private information from other participants. The attacker in this scenario can access all information and messages from the colluding participants. In the "honest-but-colluded" scenario, the attacker's capability is either equal to or stronger than in the "honest-but-curious" scenario. This is because the attacker can acquire more information from other participants when colluding, therefore increasing their ability to infer sensitive data.

---

[7]A schematic graph and illustration of asynchronous VFL is in the Appendix section D.1

**Theorem 4.1. Differential Privacy Guarantee of Sharing the ZO Gradient:** Under the "honest-but-colluded" threat model where the attacker can access all information of all clients through the entire training process, including the client's dataset, model, and the ZO gradient received from the server. Under algorithm 1, with $q$ being sufficiently large, $\zeta$ being the maximum $l_2$-norm of the partial gradient w.r.t. any client's output through the entire training. If the following condition holds:

$$\sigma_{m_t,s} = \sqrt{\frac{1}{qd_hT} \sum_{t=0}^{T-1} \text{tr}(\Psi_{m_t}^t)} > \frac{2\zeta\sqrt{2\ln(1.25/\delta)}}{N \cdot \epsilon} \tag{3}$$

we can derive that sharing the stochastic gradient estimation from the server ensures $(\bar{\epsilon}, \bar{\delta})$-DP for all $\epsilon, \delta, \delta' > 0$, where $\bar{\epsilon} = \sqrt{2T\ln(1/\delta')}\epsilon + T\epsilon(e^\epsilon - 1)$, $\bar{\delta} = T\delta + \delta'$.

*Proof sketch:* The proof can be found in Appendix C.1. Firstly, we establish that each client's update can be modeled with an ideal sequence, where the unbiased parameter is updated with the gradient of the smoothed loss function $f_{\mathbf{u},i}(w_0^t, \Phi_i) = \mathbb{E}_{\mathbf{u}}[f_i(w_0, \Phi(\mathbf{w})) + \mu\mathbf{u}]$. Since Avg-RandGradEst in Eq. 2 is an average of $q$ independently and identically distributed samples from $\tilde{f}_i(\cdot)$, applying the central limit theorem, the estimated gradient is normally distributed for sufficiently large $q$. We can then apply the differential privacy guarantee for the Gaussian Mechanism to the estimated gradient. Finally, the accumulated privacy guarantee is derived from the advanced composition theorem.

*Remark* 4.2. This theorem tells that under the worse case where a strong attacker has obtained all of the ZO information $\Delta_{m,i}$ from all client $m \in [M]$, of all sample $i \in [N]$, during the entire training process $t \in [T]$, the attacker cannot differ a single data point in the dataset, with $(\bar{\epsilon}, \bar{\delta})$-DP guarantee.

*Remark* 4.3. Since the ZO gradient is shared from the server to clients, this theorem mainly provides a privacy guarantee for the label on the server.

*Remark* 4.4. For a weaker attacker, where the attacker can only access the information from fewer clients ("honest-but-colluded"), or only one client ("honest-but-curious"), the privacy guarantee either remains the same or is strengthened.

Additionally, in Appendix C.2, we provided a detailed discussion of the privacy protection of our approach at the framework level, plus a discussion on the SOTA inference attacks under "honest-but-curious" [10, 33, 45, 45, 44, 17, 9] and "honest-but-colluded"[10, 45, 29, 38] threat model.

## 5 Convergence Analysis

**Assumption 5.1.** The formal definition and detailed discussion of the assumptions are in Appendix B. We assume that there is a feasible optimal solution for $f(\cdot)$, $\nabla f_i(\cdot)$ is $L$-Lipschitz continuous, $f_i(\cdot)$ has an unbiased gradient, bounded Hessian $H_m$ and bounded block-coordinate gradient $\mathbf{G}_m$. The activation of the clients is independent, and the client has a uniformly bounded delay $\tau$.

**Theorem 5.2.** *Under assumption 5.1, to solve the problem 1 with algorithm 1, the following inequality holds.*

$$\frac{1}{T}\sum_{t=0}^{T-1}\mathbb{E}\left\|\nabla f\left(w_0^t, \mathbf{w}^t\right)\right\|^2 \leq \frac{4p_*\mathbb{E}\left(f^0 - f^*\right)}{T\eta} + 2\eta p_* L_*\left(4d_*\mathbf{G}_*^4 + \mu_*^2 L_*^2 d_*^2\mathbf{G}_*^2 + 2\sigma_0^2\right)$$
$$+ 7\eta^2 p_*\tau\mathbf{G}_*^2\left(4d_*\mathbf{G}_*^2 + \mu_*^2 L_*^4 d_*^2 + 2L_*^2\Gamma\right)$$
$$+ \mu_*^2 p_* L_*^2 d_*^2\mathbf{G}_*^2 + \mathcal{E}p_* H_*^2\left(6 + 16\mathbf{G}_*^2\right) + 17\Gamma p_*\mathbf{G}_*^2$$

*where $L_* = \max_m\{L, L_0, L_m\}$, $\eta_0 = \eta_m = \eta$, $\frac{1}{p_*} = \min_m p_m$, $\mu_* = \max_m\{\mu_m\}$, $d_* = \max_m\{d_{h_m}\}$, $\mathbf{G}_* = \max_m\{\mathbf{G}_0, \mathbf{G}_m\}$, $H_* = \max_m\{H_0, H_m\}$, $\mathcal{E}$ and $\Gamma$ are the upper bound for the square of the norm of the forward and backward compression error respectively. $T$ is the total number of iterations (communication rounds).*

*Remark* 5.3. If we choose $\eta = \frac{1}{\sqrt{T}}$ and $\mu_* = \frac{1}{\sqrt{T}}$. Design the compression to make $\mathcal{E} = \mathcal{O}\left(\frac{1}{\sqrt{T}}\right)$ and $\Gamma = \mathcal{O}\left(\frac{1}{\sqrt{T}}\right)$ we can derive:

$$\frac{1}{T}\sum_{t=0}^{T-1}\mathbb{E}\left\|\nabla f\left(w_0^t, \mathbf{w}^t\right)\right\|^2 = \mathcal{O}\left(\frac{d_h}{\sqrt{T}}\right)$$

where $d_h = \max_m \{d_{h_m}\}$ is the maximum output layer size of the clients.

*Remark* 5.4. This theorem proves that we significantly relieve the slow convergence problem of ZOO-based VFL [41] from $\mathcal{O}(\frac{d}{\sqrt{T}})$ to $\mathcal{O}(\frac{d_h}{\sqrt{T}})$.

# 6 Experiment

We conducted a systematic experiment on the SOTA and baseline VFL and our proposed framework. The primary objective of our study was to empirically validate the security measures of our approach and demonstrate its capability in reducing communication overhead. Furthermore, we conducted an ablation study to quantify the contributions of each component toward improving communication efficiency. Due to space limitations, extra experiment details, the CIFAR-10 experiments, extra experiments on less common datasets, and extra experiments on other aspects of our framework have been placed in appendix E. The source code for this project is available at the following URL: `https://github.com/GanyuWang/VFL-CZOFO`.

## 6.1 Experiment Setups

**Datasets**   The datasets were vertically partitioned among all participants in our experiments. Each client held a portion of the features of each sample, while the server held the corresponding labels. Both the server and the client kept the sample IDs during training. We utilized two datasets in our main experiments: MNIST [20] and CIFAR-10 [19]. For both datasets, we employed two clients in our experiments, with each client being responsible for half of the images[8] in the respective dataset. Therefore, we denoted these datasets as dist-MNIST and dist-CIFAR-10.

**Models**   In our dist-MNIST experiment, we implemented a multilayer perception (MLP). Each client utilized a one-layer Fully Connected Layer (FCL), with an input size equal to the flattened local data input size, and an output size of 64. The server employed a two-layer FCL. The first layer concatenated all the outputs from all clients and produced embeddings with 128 neurons. The second layer outputs 10 neurons for prediction. ReLU [11] was used as the activation function for the server and clients. Practically, the server kept a table of the outputs from all clients for all of the samples, i.e. $\tilde{\Phi}_i^t = [h_1(w_1^{t-\tau_{1,i}^t}; x_{1,i}), \ldots, w_M^{t-\tau_{M,i}^t}]$ was maintained by the server. The server updated the corresponding value in the table when it received $h_m(w_m^{t-\tau_{m,i}^t}; x_{m,i})$ from client $m$ during each communication round. The clients' outputs from the table were then used for prediction. Regarding the dist-CIFAR-10 dataset, each client padded its half-image to full size and trained a ResNet-18 [15]. The server then summed the outputs from all the clients.

**Frameworks for Comparison**   We compare our framework with other SOTA VFL frameworks including "split-learning" [34], "compressed-VFL" [5], "Syn-VFL-ZO"[9], "VAFL" [6], "ZOO-VFL" [41]. All frameworks utilized identical base models and training procedures. While some frameworks also introduced alternative VFL settings, e.g. server and clients both processing the labels, we only focus on the VFL setting where the server stores the labels and multiple clients possess non-intersecting features.

**Training Procedures**   For the experiment applying the split MLP model on dist-MNIST, a batch size of 64 was utilized, and the model was trained for 100 epochs. For the experiment applying the ResNet-18 on dist-CIFAR-10, a batch size of 128 was used, and the model was trained for 50 epochs. The learning rate $\eta$ was chosen from $[0.1, 0.01, 0.001, \cdots]$, and $\mu$ for ZOO was chosen from $[1, 0.1, 0.001, 0.0001, \cdots]$. We use a uniform scale compressor [2] on the forward and backward messages with different compression bits $[8, 4, 2, 1]$. The number of sampling $q$ used for Avg-RandGradEst in the dist-MNIST experiment was set to $[10, 100]$, while in the dist-CIFAR-10 experiment, it was set to either $[100, 500, 1000]$. The sampling distribution is $\mathcal{U}(\mathcal{S}(\mathbf{0}, 1))$. We used a vanilla SGD with a fixed learning rate for all VFL frameworks to ensure a fair comparison. The reported test accuracy values in the tables have been obtained from five independent runs.

---

[8]Details of the feature splitting can be found in Appendix E.1
[9]This is the synchronous version of ZOO-VFL, the algorithm is in Appendix E.1

Table 2: Sampling Times of Avg-RandGradEst and the Corresponding Differential Privacy Guarantee

| $q$ | $\sigma^2$ | $\zeta$ | $\delta$ | $\epsilon$ | $\delta'$ | $\bar{\epsilon}$ | $\bar{\delta}$ |
|-----|-----------|---------|----------|------------|-----------|------------------|----------------|
| 10  | 1.61      | 20.1    | $10^{-6}$ | $2.8 \times 10^{-3}$ | $10^{-6}$ | 85.9 | $9.4 \times 10^{-2}$ |
| 100 | 0.15      | 7.3     | $10^{-6}$ | $3.3 \times 10^{-3}$ | $10^{-6}$ | 93.6 | $9.4 \times 10^{-2}$ |

## 6.2 Evaluation on the Privacy Protection

In this part, we evaluate the privacy protection of our method and Gaussian Mechanism. The major difference between the ZOO and Gaussian Mechanism in DP is that the variance of the stochastic gradient estimation (ZOO) is intrinsically controlled by the hyper-parameter of the ZOO and the characteristic of the objective function, while in the Gaussian Mechanism, the variance is a parameter that directly controls the magnitude of the noise added.

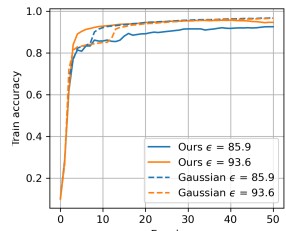

Figure 2: The Utility Comparing with the VAFL with Gaussian Mechanism

We obtain the DP guarantee w.r.t. the sampling times of ZOO in the dist-MNIST experiment and report the result in Table 2. We estimate $\sigma^2$ by freezing the model and repeatedly applying Eq.2 100 times, then calculating the variance of the estimated gradients. We obtain the maximum gradient norm $\zeta$ by recording the norm throughout the entire training process[10]. We set $\delta = 10^{-6}$ and calculate $\epsilon$ based on the given sampling times $q$. The iteration $T$ was set to the total iteration of 50 epochs, which is 93800.

In the second part, we compared the utility of our framework with the VAFL[6] using the Gaussian Mechanism and gradient clipping [1] on the partial derivative w.r.t. the client's output with the corresponding $(\epsilon, \delta)$-DP. We set the gradient norm clipping bound to 0.1 And we plotted the training accuracy for each epoch in Figure 2. As illustrated in the figure, our scheme exhibits comparable utility to VAFL with the same privacy guarantee.

## 6.3 Evaluation of Training Efficiency and Communication Cost

In this part, we compare our framework with SOTA VFL in the effectiveness of training and communication efficiency. We apply the best tuning for all other frameworks with respect to test accuracy and communication efficiency. (The experiment for dist-CIFAR-10 is in Appendix E.3)

Figure 3-(a) demonstrates that our VFL-CZOFO framework has the fastest convergence rate comparable to VAFL, demonstrating the superiority of our cascade hybrid optimization method. It is worth noting that the pure-ZOO-based VFL suffers from a slower convergence compared to other frameworks, even with an MLP model. Figure 3-(b) plots the total communication cost against the training accuracy, where the crosses indicate the point where the curve reaches 95% training accuracy for MNIST. It shows that our VFL-CZOFO framework achieves the same level of convergence with much lower communication costs.

Table 3 report the important statistic about the training. As shown in the table, our framework significantly outperforms other communication-efficient frameworks in terms of communication cost.

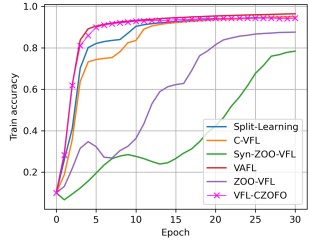

(a) MNIST by epochs

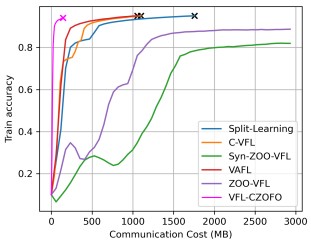

(b) MNIST by comm. cost

Figure 3: Compare with other VFL Frameworks

---

[10]Alternatively, one can also apply gradient clipping[1] to get a better privacy guarantee by bounding $\zeta$, however, we want to focus on the intrinsic privacy of our method, therefore only record the maximum norm.

Table 3: Evaluating the Test Accuracy and the Comm. Cost

| | Privacy | Test Accuracy | Cost(60%) | Cost (80%) | Cost(90%) | Cost (95%) | Cost (total) |
|---|---|---|---|---|---|---|---|
| Split learning [34] | ✗ | $98.03 \pm 0.07$ | 175 MB | 234 MB | 937 MB | 1758 MB | 5859 MB |
| Compressed-VFL [5] | ✗ | $98.17 \pm 0.13$ | 109 MB | 330 MB | 440 MB | 1099 MB | 3663 MB |
| VAFL [6] | ✗ | $97.36 \pm 0.14$ | 123 MB | 184 MB | 307 MB | 1106 MB | 6144 MB |
| VAFL[6]+DP[1] | ✓ | $95.81 \pm 0.20$ | 123 MB | 184 MB | 799 MB | 1413 MB | 6144 MB |
| Syn-ZOO-VFL | ✓ | $84.51 \pm 0.72$ | 1406 MB | 2520 MB | - | - | - |
| ZOO-VFL [41] | ✓ | $85.62 \pm 0.45$ | 860 MB | 1300 MB | - | - | - |
| Ours | ✓ | $95.00 \pm 0.21$ | **16** MB | **24** MB | **39** MB | **205** MB | 790 MB |

## 6.4 Ablation Study

**Minimal Zeroth-Order Optimization**   In this section, we evaluate the impact of different ZOO parameters on convergence and communication costs. We did not apply compression in this section. We plot the training curve against epochs in Figure 4-(a), and against the logarithm base 10 of the total backward message size in Figure 4-(b). Table 4 reports the test accuracy and communication costs. (The experiment for dist-CIFAR-10 can be found in Appendix E.3.2 with a similar conclusion.)

Figure 4 illustrates how increasing the number of sampling $q$ can enhance the convergence of our proposed method. When $q$ is set to 100, the training curve of our method closely matches that of the VAFL method. Notably, the use of ZOO in VFL results in a significant reduction in communication cost, reducing the backward cost by a factor of $40$ compared to VAFL.

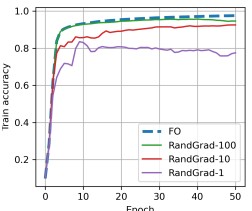 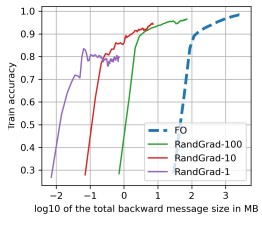

(a) MNIST by epochs    (b) MNIST by comm. cost

Figure 4: Ablation Study on ZOO

Table 4: Ablation Study on ZOO

| ZO Type | Test Accuracy | Backward Cost |
|---|---|---|
| FO | $97.36 \pm 0.14$ | 3073 MB |
| AvgRand-100 | $95.30 \pm 0.25$ | **75** MB |
| AvgRand-10 | $93.33 \pm 0.55$ | 7.5 MB |
| AvgRand-1 | $79.66 \pm 0.96$ | 0.75 MB |

**Compression of the Communication**   In this section, we investigate the impact of compression on convergence and communication costs using Avg-RandGradEst with 100 samplings. We evaluate different compression schemes on either the forward (F) or backward (B) message. Figure 5 presents the training curve plotted against the epoch and the log10 of the communication cost of the message with compression. Table 5 displays the test accuracy and the total cost at the end of the 100-epoch training period. (corresponding dist-CIFAR-10 experiment can be found in Appendix E.3.2.)

The results show that increasing the compression rate reduces communication costs but leads to a decline in test accuracy. Notably, even when compressing the gradient information to 2 bits, there is no significant decrease in test accuracy, and the message size is significantly reduced to 5.4 MB. Hence, a trade-off can be identified in practical application to achieve a balance point.

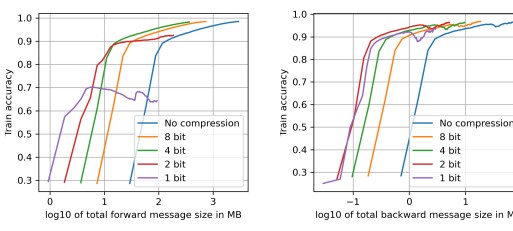

(a) MNIST by forward cost  (b)  MNIST  by  backward
cost

Figure 5: Ablation Study on Compression

Table 5: Ablation Study on Compression

| Compression Type | Test Accuracy | Message Cost |
|---|---|---|
| No Compression | $95.30 \pm 0.25$ | 3073 MB |
| F-8 bits | $95.02 \pm 0.17$ | 769 MB |
| F-4 bits | $93.35 \pm 0.23$ | 386 MB |
| F-2 bits | $79.10 \pm 0.41$ | 194 MB |
| F-1 bit | $30.20 \pm 0.45$ | 96 MB |
| B-8 bits | $94.58 \pm 0.21$ | 19 MB |
| B-4 bits | $94.50 \pm 0.10$ | 10 MB |
| B-2 bits | $94.16 \pm 0.16$ | 5.4 MB |
| B-1 bit | $90.67 \pm 0.22$ | 3.1 MB |

# 7 Limitation

While the utilization of ZOO improves communication efficiency and security, it comes at the expense of increased computational costs for the server. Specifically, the server needs to perform $q$ extra forward propagations on its local model compared to other FOO-based VFL methods. A more comprehensive analysis and supplementary experiments can be found in Appendix E.2. However, it is important to note that the server typically has higher computational capabilities, which makes the associated costs manageable and acceptable.

# 8 Conclusion

We propose a solution in VFL that improves communication efficiency and privacy simultaneously. Our method has been theoretically proven to outperform ZOO-based VFL in terms of convergence, while also providing proof of the intrinsic differential privacy guarantees. Through our experiments, we demonstrate that our method substantially reduces the communication cost of VFL in comparison to the state-of-the-art communication-efficient VFL. Furthermore, our method achieves comparable utility to VFL models that offer the same level of privacy guarantee.

## Acknowledgments and Disclosure of Funding

This work has been supported by the Natural Sciences and Engineering Research Council of Canada (NSERC), Discovery Grants program. Bin Gu was partially supported by the National Natural Science Foundation of China under Grant 62076138.

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
