# A Unified Solution for Privacy and Communication Efficiency in Vertical Federated Learning (Supplemental Material)

## A  Notation Table

Below is a notation table for the parameter used in the convergence analysis.

Table 1: Notation Table

| Basic: | |
|---|---|
| $\mathbf{w} = [w_1, w_2, \cdots, w_M]$ | The parameter for all the clients. |
| $h_{m,i} = h_m(w_m; x_{m,i})$ | The local model output of client $m$ with sample $i$ |
| $\Phi_i = \Phi_i(\mathbf{w}) = [h_1(w_1; x_{1,i}), \cdots, h_M(w_M; x_{M,i})] = [h_{1,i}, \ldots, h_{M,i}]$ | The output embeddings from all clients for sample $i$ |
| $f(w_0, \mathbf{w}) = f(w_0, \mathbf{w}, X, y)$ | The global loss function |
| $f_i(w_0, \Phi_i(\mathbf{w})) = f_i(w_0, h_{1,i}, \ldots, h_{M,i})$ | The loss function for the sample $i$ calculated by server. |

| With timestep $(t)$, clients' delay $(\tilde{\mathbf{w}})$, embedding compression $(\hat{f})$, ZOO gradient estimator $(\hat{\nabla})$ | |
|---|---|
| $w_m^t$ | The client $m$'s parameter, at global timestep $t$, |
| $\mathbf{w}^t = [w_1^t, \ldots, w_M^t]$ | The clients' parameter at global timestep $t$ |
| $\tilde{\mathbf{w}}^t = \mathbf{w}^{t-\tau_i^t} = [w_1^{t-\tau_{1,i}^t}, \ldots, w_M^{t-\tau_{M,i}^t}]$ | The delayed parameter for all the clients at global time step $t$ (and the local timestep is 0 for all $w$). |
| $\Phi_i^t = \Phi_i(\mathbf{w}^t) = [h_1(w_1^t; x_{1,i}), \cdots, h_M(w_M^t; x_{M,i})]$ | The output embeddings from all clients for sample $i$ at global timestep $t$ without delay. |
| $\tilde{\Phi}_i^t = \Phi_i(\tilde{\mathbf{w}}^t) = [h_1(w_1^{t-\tau_{1,i}^t}; x_{1,i}), \cdots, h_M(w_M^{t-\tau_{M,i}^t}; x_{M,i})]$ | The output embeddings from all clients for sample $i$ with the client delay at global timestep $t$ |
| $\Phi_i^t(w_m^t) = [h_1(w_1^t; x_{1,i}), \cdots h_m(w_m^t; x_{m,i}), \cdots h_M(w_M^t; x_{M,i})]$ | $\Phi_i^t$ substitute the client $m$'s parameter with $w_m^t$ |
| $\tilde{\Phi}_i^t(w_m^t) = [h_1(w_1^{t-\tau_{1,i}^t}; x_{1,i}), \cdots h_m(w_m^t; x_{m,i}), \cdots h_M(w_M^{t-\tau_{M,i}^t}; x_{M,i})]$ | $\tilde{\Phi}_i^t$ substitute the client $m$'s parameter with $w_m^t$ |
| $\hat{f}_i(w_0, \Phi_i) = f_i(w_0, h_{1,i} + \epsilon_{1,i}, \cdots, h_{M,i} + \epsilon_{M,i})$ | The loss function with compression error of all client's embedding. |
| $\hat{\nabla}_{h_{m,i}} f_i(w_0, \Phi_i) = \frac{\phi(d_{h_m})}{\mu_m}[f_i(h_{m,i} + \mu_m u_{m,i}) - f_i(h_{m,i})]u_{m,i}$ | The ZO gradient estimator w.r.t. the client $m$'s output |

Submitted to 37th Conference on Neural Information Processing Systems (NeurIPS 2023). Do not distribute.

# B Assumptions

Assumptions B.1 - B.4 are the basic assumptions for solving the non-convex optimization problem with stochastic gradient descent [10, 15, 21].

**Assumption B.1. Feasible Optimal Solution:** Function $f$ is bounded below that is, there exist $f^*$ such that,

$$f^* := \inf_{[w_0, \mathbf{w}] \in \mathbb{R}^d} f(w_0, \mathbf{w}) > -\infty.$$

**Assumption B.2. Lipschitz Gradient:** $\nabla f_i$ is $L$-Lipschitz continuous w.r.t. all the parameter, i.e., there exists a constant $L$ for $\forall [w_0, \mathbf{w}], [w_0', \mathbf{w}']$ such that

$$\left\| \nabla_{[w_0, \mathbf{w}]} f_i(w_0, \Phi_i(\mathbf{w})) - \nabla_{[w_0, \mathbf{w}]} f_i(w_0', \Phi_i(\mathbf{w}')) \right\| \leq L \left\| [w_0, \mathbf{w}] - [w_0', \mathbf{w}'] \right\|$$

specifically there exists an $L_m > 0$ for all parties $m = 0, \cdots, M$ such that $\nabla_{w_m} f_i$ is $L_m$-Lipschitz continuous:

$$\left\| \nabla_{w_m} f_i(w_0, \Phi_i(\mathbf{w})) - \nabla_{w_m} f_i(w_0', \Phi_i(\mathbf{w}')) \right\| \leq L_m \left\| [w_0, \mathbf{w}] - [w_0', \mathbf{w}'] \right\|$$

**Assumption B.3. Unbiased Gradient:** For $m \in 0, 1, \cdots M$ for every data sample $i$, the stochastic partial derivatives are unbiased, i.e. $\mathbb{E}_i \nabla_{w_m} f_i(w_0, \Phi_i(\mathbf{w})) = \nabla_{w_m} f(w_0, \Phi_i(\mathbf{w}))$

**Assumption B.4. Bounded Variance:** For $m = 0, 1, \cdots, M$, there exist constants $\sigma_m \leq \infty$ such that the variance of the stochastic partial derivatives are bounded: $\mathbb{E}_i \left\| \nabla_{w_m} f_i(w_0, \Phi_i(\mathbf{w})) - \nabla_{w_m} f(w_0, \mathbf{w}) \right\|^2 \leq \sigma_m^2$

Assumptions B.5 - B.6 are the base assumptions for bounding the compression of the embedding on the loss [4]. Since compression introduces error in the input of the loss function, therefore with the bounded Hessian we can derive the maximum effect of the error on the loss. And bounding the block-coordinated gradient is common in VFL analysis for bounding the gradient for the entire model when the gradient of other parts have been bounded [4, 12, 22].

**Assumption B.5. Bounded Hessian:** The Hessian for $f_i(w_0, \Phi_i(\mathbf{w}))$ is bounded, i.e. there exist positive constant $H_m$ for $m = 0, 1, \cdots M$ such that for all $[w_0, \mathbf{w}]$, the following inequalities holds:

$$\left\| \nabla^2_{w_0} f_i(w_0, \Phi_i(\mathbf{w})) \right\| \leq H_0$$
$$\left\| \nabla^2_{(w_m; x_{m,i})} f_i(w_0, \Phi_i(\mathbf{w})) \right\| \leq H_m$$

Where the norm is the spectral norm (the matrix norm induced by L2-norm[1])

**Assumption B.6. Bounded Block-coordinate Gradient:** The gradient of all the participants' local output w.r.t. their local input is bounded, i.e. there exist positive constants $\mathbf{G}_0$ for the server $m = 0$ the following inequalities holds:

$$\left\| \nabla_{[w_0, h_{1,i}, \cdots, h_{M,i}]} f_i(w_0, h_{1,i}, \cdots, h_{M,i}) \right\| \leq \mathbf{G}_0$$

and there exist positive constants $\mathbf{G}_m$ for the client $m = 1, \cdots, M$ the following inequalities hold:

$$\left\| \nabla_{w_m}(w_m; x_{m,i}) \right\| \leq \mathbf{G}_m$$

where the first inequality bounds the gradient for the server w.r.t. to all the outputs received from the clients, and the second inequality bounds the gradient for the client's outputs w.r.t. the client's local parameter.

Assumptions B.7 - B.8 are the assumptions for dealing with the asynchronous updates of our VFL framework. We assume that the activation of clients at each global round is independent and that the maximum delay is bounded [21, 5, 12]. These are reasonable assumptions for analysis.

**Assumption B.7. Independent Client:** The activated client $m_t$ for the global iteration $t$ is independent of $m_0, \cdots, m_{t-1}$ and satisfies $\mathbb{P}(m_t = m) := p_m$

**Assumption B.8. Uniformly Bounded Delay:** For each client $m$, and each sample $i$, the delay at each global iteration $t$ is bounded by a constant $\tau$. i.e. $\tau_{m,i}^t \leq \tau$

---

[1]For notation brevity, unless specific, the norm is L2-norm for the vector and spectral norm for the matrix.

## C  Security Analysis

### C.1  Differential Privacy Guarantee of Sharing the Stochastic Estimation of the Gradient

**Definition C.1.**  $(\epsilon, \delta)$-**Differential Privacy**  A randomized mechanism $\mathcal{M} : \mathcal{D} \to \mathcal{R}$ with domain $\mathcal{D}$ and range $\mathcal{R}$ satisfies $(\epsilon, \delta)$-differential privacy if for any two adjacent inputs $d, d' \in \mathcal{D}$, and for any subset of outputs $S \subseteq \mathcal{R}$ it holds that:

$$\Pr[\mathcal{M}(d) \in S] \leq e^{\epsilon}\Pr[\mathcal{M}(d') \in S] + \delta$$

Now we start the proof of the $(\epsilon, \delta)$-Differential Privacy of the training process of sharing the stochastic gradient.

The activated client $m_t$ at iteration $t$ is updated with the following equation.

$$w_{m_t}^{t+1} = w_{m_t}^t - \eta\hat{\nabla}_{h_{m_t,i}}f_i\left(w_0^t, \Phi_i\right) \cdot \nabla_{m_t}h_{m_t,i} \tag{1}$$

where

$$\hat{\nabla}_{h_{m_t,i}}f_i\left(w_0^t, \Phi_i\right) = \frac{1}{q}\sum_{j=1}^{q}\frac{1}{\mu_{m_t}}[f_i(h_{m_t,i} + \mu_{m_t}\mathbf{u}_{m_t,i}^j) - f_i(h_{m_t,i})]\mathbf{u}_{m_t,i}^j \tag{2}$$

For notation brevity, we define:

$$g_{m_t}^{t,j} \triangleq \frac{1}{\mu_{m_t}}[f_i(h_{m_t,i} + \mu_{m_t}\mathbf{u}_{m_t,i}^j) - f_i(h_{m_t,i})]\mathbf{u}_{m_t,i}^j \tag{3}$$

$$g_{m_t}^t \triangleq \hat{\nabla}_{h_{m_t,i}}f_i\left(w_0^t, \Phi_i\right) = \frac{1}{q}\sum_{j=1}^{q}g_{m_t}^{t,j} \tag{4}$$

We will show in the following lemma C.2 that the solution can be regarded as client updating with the unbiased gradient of the smoothed loss function $f_{\mathbf{u},i}\left(w_0^t, \Phi_i\right) = \mathbb{E}_{\mathbf{u}}[f_i(w_0, \Phi(\mathbf{w})) + \mu\mathbf{u}]$, but adding a stochastic noise on it, where the unbiased ideal parameter sequence of the client $m_t$ is defined as $\breve{w}_{m_t}$. Formally:

$$\breve{w}_{m_t}^{t+1} = \breve{w}_{m_t}^t - \eta\nabla_{h_{m_t,i}}f_{\mathbf{u},i}\left(w_0^t, \Phi_i\right) \cdot \nabla_{m_t}h_{m_t,i} \tag{5}$$

$$w_{m_t}^t = \breve{w}_{m_t}^t + \xi_{m_t}^t \tag{6}$$

Where $\xi^t$ is a stochastic variable.

For notation brevity, we define:

$$\breve{g}_{m_t}^t \triangleq \nabla_{h_{m_t,i}}f_{\mathbf{u},i}\left(w_0^t, \Phi_i\right) \tag{7}$$

**Lemma C.2.**  *For $t = 0, ...T - 1$, if $g_{m_t}^{t,j}$ is i.i.d. and $q$ is sufficiently large, then $w_{m_t}^t$ is distributed as:*

$$w_{m_t}^t \sim \mathcal{N}\left(\breve{w}_{m_t}^t, \frac{1}{q}\eta^2(\nabla_{m_t}h_{m_t,i})^{\mathsf{T}}\Psi_{m_t}^t\nabla_{m_t}h_{m_t,i}\right) \tag{8}$$

*and $g_{m_t}^t$ is distributed as*

$$g_{m_t}^t \sim \mathcal{N}\left(\breve{g}_{m_t}^t, \frac{1}{q}\Psi_{m_t}^t\right) \tag{9}$$

*proof:* First we show that $\mathbb{E}[w_{m_t}^t] = \breve{w}_{m_t}^t$, we prove this by Mathematical Induction. The $w_{m_t}^0 = \breve{w}_{m_t}^0$ holds by natural when initializing the parameter. Assuming $\mathbb{E}[w_{m_t}^{t-1}] = \breve{w}_{m_t}^{t-1}$, we have:

$$\mathbb{E}[w_{m_t}^t] = \mathbb{E}[w_{m_t}^{t-1} - \eta\hat{\nabla}_{h_{m_t,i}}f_i\left(w_0^t, \Phi_i\right) \cdot \nabla_{m_t}h_{m_t,i}]$$
$$= \breve{w}_{m_t}^{t-1} - \eta\mathbb{E}[\hat{\nabla}_{h_{m_t,i}}f_i\left(w_0^t, \Phi_i\right) \cdot \nabla_{m_t}h_{m_t,i}]$$

$$
\begin{aligned}
&= \breve{w}^{t-1}_{m_t} - \eta \nabla_{h_{m_t,i}} f_{\mathbf{u},i}\left(w^t_0, \Phi_i\right) \cdot \nabla_{m_t} h^t_{m_t,i} \\
&= \breve{w}^t_{m_t}
\end{aligned}
\tag{10}
$$

Where the third equality applies the lemma D.3 (Eq. 26). Therefore $\mathbb{E}[w^t_{m_t}] = \breve{w}^t_{m_t}$ for $t = 0, 1, \dots T - 1$.

For the stochastic gradient estimation $g^t_{m_t}$ which applies $q$ times of sampling on the same distribution centering on $w^t_{m_t}$. Apply the central limit theorem, and assuming the expectation and covariance matrix for each sampling is $\boldsymbol{\mu}^t_{m_t}$ and $\Psi^t_{m_t}$ respectively, we have

$$
g^t_{m_t} = \frac{1}{q} \sum_{j=1}^{q} g^{t,j}_{m_t} \sim \mathcal{N}(\boldsymbol{\mu}^t_{m_t}, \frac{1}{q}\Psi^t_{m_t})
\tag{11}
$$

The update of the $w^t_{m_t}$ is distributed as:

$$
-\eta g^t_{m_t} \nabla_{m_t} h_{m_t,i} \sim \mathcal{N}\left(-\eta \boldsymbol{\mu}^t_{m_t} \nabla_{m_t} h_{m_t,i}, \frac{1}{q}\eta^2 (\nabla_{m_t} h_{m_t,i})^{\mathsf{T}} \Psi^t_{m_t} \nabla_{m_t} h_{m_t,i}\right)
\tag{12}
$$

which is the only stochastic part, therefore we have

$$
w^t_{m_t} \sim \mathcal{N}\left(\breve{w}^t_{m_t}, \frac{1}{q}\eta^2 (\nabla_{m_t} h_{m_t,i})^{\mathsf{T}} \Psi^t_{m_t} \nabla_{m_t} h_{m_t,i}\right)
\tag{13}
$$

the first part of the lemma has been proved.

Specifically, $\mathbb{E}[g^t_{m_t}] = \breve{g}^t_{m_t}$ (lemma D.3 Eq. 26), therefore we have

$$
g^t_{m_t} = \frac{1}{q} \sum_{j=1}^{q} g^{t,j}_{m_t} \sim \mathcal{N}\left(\breve{g}^t_{m_t}, \frac{1}{q}\Psi^t_{m_t}\right)
\tag{14}
$$

the second part of the lemma has been proved. $\qquad\square$

**Lemma C.3.** $(\epsilon, \delta)$-*Differential Privacy for Gaussian mechanism: Let $\epsilon \in (0, 1)$ be arbitrary, for $c^2 > 2\ln(1.25/\delta)$ the Gaussian mechanism with parameter $\sigma > c\Delta_2/\epsilon$ is $(\epsilon, \delta)$-differential privacy.*

*proof:* The proof is in [6] Theorem A.1.

**Theorem C.4.** *Let $\epsilon \in [0, 1]$, the covariance matrix of the $g^t_{m_t}$ be $\Psi^t_{m_t}$, with the following condition holds:*

$$
\sigma_{m_t,s} = \sqrt{\frac{1}{q d_h T} \sum_{t=0}^{T-1} tr(\Psi^t_{m_t})} > \frac{2\sqrt{2\ln(1.25/\delta)}\mathbf{G}_0}{N \cdot \epsilon}
\tag{15}
$$

*under Algorithm 1, sharing the stochastic estimation of the partial gradient for each iteration satisfy $(\epsilon, \delta)$- differential privacy.*

*Proof:* From lemma C.2 we already have that:

$$
g^t_{m_t} \sim \mathcal{N}\left(\breve{g}^t_{m_t}, \frac{1}{q}\Psi^t_{m_t}\right)
\tag{16}
$$

To make the problem more trackable, assume each entry of $g^t_{m_t}$ is independent of each other and has the same variance value, and the variance is stable throughout the training process. Therefore, for each entry $s$ of $g^t_{m_t}$

$$
g^t_{m_t,s} \sim \mathcal{N}(\breve{g}_{m_t t,s}, \sigma^t_{m_t,s})
\tag{17}
$$

where $\sigma_{m_t} = \frac{1}{q d_h T} \sum_{t=0}^{T-1} tr(\Psi^t_{m_t})$ is the averaged variance for each entry.

The $l_2$-norm sensitive of $g^t_{m_t}$ is given by

$$
\Delta_{m_t,2} = \max_{\mathcal{D},\mathcal{D}'} \left\| \breve{g}^t_{m_t,\mathcal{D}} - \breve{g}^t_{m_t,\mathcal{D}'} \right\|_2
\tag{18}
$$

Assume the probability of selecting each sample in the dataset $\mathcal{D}$ (or $\mathcal{D}'$) is the same. $\mathcal{D}$ and $\mathcal{D}'$ are two neighboring dataset differing in only one sample $(x_i, y_i)$ and $(x_i', y_i')$. Without loss of generality, we set the differing sample to be the $N$-th sample. Assume $\zeta = \max_{m_t,i} \left\| \nabla_{h_{m_t,i}} f_{\mathbf{u},i}\left(w_0^t, \Phi_i\right) \right\|$ be the maximum $l_2$-norm of the partial gradient w.r.t. any client's output through the entire training.

$$
\begin{aligned}
\Delta_{m_t,2} &= \max_{\mathcal{D},\mathcal{D}'} \| \frac{1}{N} \sum_{i=0}^{N} \nabla_{h_{m_t,i}} f_{\mathbf{u},i}\left(w_0^t, \Phi_i\right) - \frac{1}{N} \sum_{i=0}^{N-1} \nabla_{h_{m_t,i}} f_{\mathbf{u},i}\left(w_0^t, \Phi_i\right) \\
&\quad - \frac{1}{N} \nabla_{h_{m_t,N}} f_{\mathbf{u},N}\left(w_0^t, h_1(w_1; x_{1,N}'), h_2(w_2; x_{2,N}'); y_N'\right) \|_2 \\
&= \frac{1}{N} \max_{\mathcal{D},\mathcal{D}'} \left\| \left[ \nabla_{h_{m_t,i}} f_{\mathbf{u},i}\left(w_0^t, \Phi_i\right) - \nabla_{h_{m_t,N}} f_{\mathbf{u},N}\left(w_0^t, h_1(w_1; x_{1,N}'), h_2(w_2; x_{2,N}'); y_N'\right) \right] \right\|_2 \\
&\leq \frac{2\zeta}{N}
\end{aligned}
\tag{19}
$$

where the inequality is based on assumption B.6.

Applying lemma C.3, with the $l_2$-norm sensitive $\Delta_{m_t,2}$ of $\breve{g}_m$. We derived the Theorem C.4. ∎

**Total Privacy**   Now we consider the total privacy of the entire training process.

**Lemma C.5.** *(Advanced Composition) For all $\epsilon, \delta, \delta'$ the class of $(\epsilon, \delta)$-DP mechanisms satisfies $(\epsilon', k\delta + \delta')$-DP under $k$-fold adaptive composition for:*

$$
\epsilon' = \sqrt{2k\ln(1/\delta')}\epsilon + k\epsilon(e^\epsilon - 1)
\tag{20}
$$

*Proof:* The proof is from [6, Theorem 3.20].

**Theorem C.6.** *Under the "honest-but-colluded" threat model where the attacker can access all information of all clients through the entire training process. Under algorithm 1, if the following condition holds:*

$$
\frac{1}{qd_hT} \sum_{t=0}^{T-1} \eta \, tr(\Psi_{m_t}^t) > \frac{2\zeta\sqrt{2\ln(1.25/\delta)}}{N \cdot \epsilon}
\tag{21}
$$

*we can derive that sharing the stochastic gradient estimation from the server ensures $(\epsilon', T\delta + \delta')$-DP for all $\epsilon, \delta, \delta' > 0$, where $\epsilon' = \sqrt{2T\ln(1/\delta')}\epsilon + T\epsilon(e^\epsilon - 1)$.*

*Proof:* Theorem C.4 has proved the $(\epsilon, \delta)$-DP for each iteration, then applying lemma C.5 with the number of composition $k = T$, the theorem is proved. ∎

**Lemma C.7.** *Given target privacy parameters $0 < \epsilon' < 1$ and $\delta' > 0$, to ensure $(\epsilon', k\delta + \delta')$ cumulative privacy loss over $k$ mechanisms, it suffices that each mechanism is $(\epsilon, \delta)$-differentially private, where*

$$
\epsilon = \epsilon'/(2\sqrt{2k\ln(1/\delta')})
\tag{22}
$$

*Proof:* The proof is from [6, Corollary 3.21]

**Corollary C.8.** *With algorithm 1, for each iteration, for all $\delta > 0$, if the following conditions holds:*

$$
\frac{1}{qd_hT} \sum_{t=0}^{T-1} \eta \, tr(\Psi_{m_t}^t) > \frac{4\zeta\sqrt{4T\ln(1.25/\delta)\cdot\ln(1/\delta')}}{N \cdot \epsilon'}
\tag{23}
$$

*we can derived $(\epsilon', T\delta + \delta')$ cumulative privacy loss over $T$ iteration,*

*Proof:* Applying lemma C.7 and plugging in $\epsilon = \epsilon'/(2\sqrt{2T\ln(1/\delta')})$

## C.2   Defense Against the State-of-the-art Privacy Inference Attack in VFL

**Privacy Protection of VFL-CZOFO at Framework Level**   VFL-CZOFO ensures the protection of *labels* on the server through two key mechanisms. Firstly, the information shared by the server with the clients is the ZO gradient with intrinsic DP rather than the vulnerable unbiased gradient. Secondly, the internal details of the server's model and the domain information associated with the labels are not disclosed to the client. Thus, the server appears as a black box to the client, allowing queries from the clients, but responds with noisy outputs to protect privacy.

VFL-CZOFO ensures the protection of the *features* on the clients by keeping the internal model information of the clients not disclosed to the server. Additionally, clients have the flexibility to utilize any model, which makes them appear as black-box models to the server. As a result, the server can only obtain the outputs from the black box (client) but does not have access to the corresponding inputs nor the ability to make adaptive queries.

**SOTA Inference Attack under "Honest-but-curious"**   We discuss two types of privacy inference attacks: the label inference attack and the feature inference attack.

A label inference attack[8, 17] under the "honest-but-curious" model involves a curious client attempting to infer the label of the dataset from the server. The "direct label inference" attack proposed by Fu et al. [8] cannot successfully attack our framework because it relies on a strong assumption that the client explicitly knows that the server simply sums the output from all clients. This allows the gradient replied by the server to directly reveal the label information by the sign of the entries. However, in our framework, we assume that the server can use any model and that the client does not have access to the server's model. Additionally, the gradient is not sent to the client in our framework. The "model completion" attack by Fu et al. [8] and the "forward embeddings" attack by Sun et al. [17] rely on the local model's representation of the unknown label, assuming that the curious client can obtain a small number of labels for the samples. Essentially, this is equivalent to using the local model and the local feature set to guess the label. The effectiveness of these attacks depends solely on the representation of the local model and features held by the client. However, with Theorem 3, we have proven the $(\epsilon, \delta)$-differential privacy of sharing the ZO gradient. Therefore, the client cannot differ one item from the server's dataset. Deep Leakage from Gradient and its variation [24, 23, 13], utilizes gradient information to optimize and reconstruct the label from the model. However, this attack does not apply to our framework. The attack assumes that the attacker has access to the target model's structure and parameters, as well as unbiased gradient information. In contrast, in our framework, participants cannot access each other's model information through the protocol. Additionally, our framework does not provide the attacker with any gradient information. Instead, the attacker can only obtain a stochastic estimation of the gradient of the ZOO.

The feature inference attack [24, 13, 7], involves the server, acting as the attacker, attempting to infer the feature from the clients under the "honest-but-curious" model. Deep leakage from gradient [24] can also be used as a feature inference attack in VFL, however, this attack cannot compromise our framework because the attacker cannot access the victim model and cannot get a certain gradient information. Model inversions attack [7] can be considered as a feature inference attack, wherein the server uses the output of the client to recover the feature. However, this attack cannot successfully compromise our framework because it relies on the attacker adaptively querying the model of the victim with specially designed input features, which is not allowed in our framework. Specifically, our framework does not provide a mechanism for the server to query the client with feature inputs. Moreover, all of the attacks mentioned above assume that the attacker obtained the domain of the label or features, however, in our framework, the client and server can collaborate without sharing the tasks information.

**"Honest-but-colluded"**   In the "honest-but-colluded" threat model, the label inference attack involves some clients colluding to infer the label from the server [8, 24]. In the worst-case scenario, all clients collude to infer the label from the server. If there is only one client, the "Honest-but-curious" threat model is equivalent to the "honest-but-colluded" model since there are no other participants to collude with. Even in the worst-case scenario where all clients collude, our framework remains resilient to the "direct label inference" attack described in Fu et al. [8]. This is because the server's model remains unknown to the clients, and the clients can only obtain a stochastic estimation of the gradient, rather than the unbiased gradient information, which makes it impossible for them to perform the attack successfully. Our framework remains resistant to the Deep Leakage from Gradient

attack [24] because this attack relies on the attacker having knowledge of the parameter and structure of the victim's model, which is not shared in our framework. Furthermore, the attacker is unable to access the unbiased gradient in our framework, which further prevents this attack from being successful.

The feature inference attack in the "honest-but-colluded" threat model is that the server colluded with some clients to infer the features from one client. In the worst-case scenario, the server colludes with all clients except the victim client. The attacker can access all the information from the colluding participants, including the model information, the dataset, and the communications between other participants, following the protocol. Luo et al. [16] propose a feature inference attack in VFL where they assume that all participants collude except the victim client. Additionally, they explicitly assume that all clients use the Logistic Regression (LR) model. However, this attack is not applicable to our framework because we do not assume a specific model for the client, and the attacker cannot access the model information of the victim, making the attack infeasible. The Reverse Multiplication Attack [19] is similar to the feature inference attack proposed by Luo et al. [16], but the target model adds Homomorphic Encryption to protect the data. In the attack, the authors assume that the "coordinate participant" who has the private key is also corrupted, enabling them to decrypt all data received from the victim. They then perform an equation-solving attack for the LR model. However, this attack is not applicable to our framework because we do not assume an LR model, and the attacker cannot access the victim's model. Therefore, our framework is secure against such attacks.

# D Convergence Analysis

## D.1 Asynchronous VFL Framework

We use an Asyn-VFL framework [5] where the server passively handles the request from the clients and replies with the necessary information instead of actively sending messages to coordinate with the training process of the clients. Asyn-VFL can be modeled with a global iteration sequence, where each iteration has four steps. As shown in Fig. 1, step 1 is that the client $m_t$ is activated and sends the forward message to the server. In step 2, the server replies with the necessary information for the client's update. Then the server does local updates based on the updated forward message and the client update its model with the backward message.

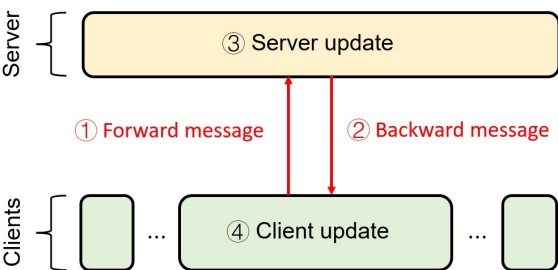

Figure 1: Asynchronous VFL framework

At any iteration $t$, only one client $m_t$ and the server updates its parameter. Therefore, there are delays between parameters updated at iteration $t$ and parameters that are not updated. In this paper, we use $\tau_{m,i}^t$ to denote the delay w.r.t. the client $m$ and the sample $i$ at global iteration $t$. At each global iteration, the client $m_t$ is activated and its delay is clear; for all other clients $m \neq m_t$, their delay count $\tau_{m,i}^t$ is increased by 1. Formally, the update rules for delay $\tau_{m,i}^t$ can be represented as following:

$$\tau_{m,i}^{t+1} = \begin{cases} 1, & m = m_t, i = i_t \\ \tau_{m,i}^t + 1, & \text{otherwise} \end{cases}$$

## D.2 Defining Compression Errors

**Definition D.1. Compression Error (forward message).** Let vector $\epsilon_{m,i}$ be the compression error of $\mathcal{C}_m(\cdot)$, on sample $i$, i.e. $\epsilon_{m,i} = \mathcal{C}_m(h_m(w_m; x_i)) - h_m(w_m; x_i)$. We denote the expected norm

of the error from the client $m$ at global iteration $t$ as $\mathcal{E}_m^t = \mathbb{E} \left\| \epsilon_{m_t, i_t} \right\|^2$, and define the upper limit $\mathcal{E} = \max_t \left\{ \mathcal{E}^t \right\}$

**Definition D.2. Compression Error (backward message).** Let $e_{m,i}$ be the compression error on $\delta_{m,i}$, i.e. $e_{m,i} = \mathcal{C}_0(\delta_{m,i}) - \delta_{m,i}$. the pseudo partial derivative that the client gets is

$$
\begin{aligned}
v_m &= \frac{\phi(d_{h_m})}{\mu_m}[f_i\left(h_{m,i} + \mu_m \mathbf{u}_{m,i}\right) - f_i\left(h_{m,i}\right) + e_{m,i}]\mathbf{u}_{m,i} \\
&= \frac{\phi(d_{h_m})}{\mu_m}[f_i\left(h_{m,i} + \mu_m \mathbf{u}_{m,i}\right) - f_i\left(h_{m,i}\right)]\mathbf{u}_{m,i} + \frac{\phi(d_{h_m})}{\mu_m}e_{m,i}\mathbf{u}_{m,i} \\
&= \hat{\nabla}_{h_{m,i}} f_i\left(w_0, \phi_i(\mathbf{w})\right) + \frac{\phi(d_{h_m})}{\mu_m}e_{m,i}\mathbf{u}_{m,i}
\end{aligned}
\tag{24}
$$

Define vector $\gamma_{m,i} = \frac{\phi(d_{h_m})}{\mu_m}e_{m,i}\mathbf{u}_{m,i}$ be the compression error on $\hat{\nabla}_{h_{m,i}} f_i\left(w_0, \phi_i(\mathbf{w})\right)$, so that

$$
v_m = \hat{\nabla}_{h_{m,i}} f_i\left(w_0, \phi_i(\mathbf{w})\right) + \gamma_{m,i}
\tag{25}
$$

and define $\gamma^t = \gamma_{m_t, i_t}$, be the error on the activated client $m_t$ and the selected sample $i_t$ at iteration $T$, and $\Gamma^t = \mathbb{E} \left\| \gamma^t \right\|^2$. The upper limit for the compression error is $\Gamma = \max_t \left\{ \Gamma^t \right\}$.

## D.3 Lemmas

**Lemma D.3.** *Zeroth-Order Optimization. For arbitrary $f \in C_L^1(\mathcal{R}^d)$, we have:*

*1) $f_\mu(x)$ is continuously differentiable, its gradient is Lipschitz continuous with $L_\mu \leq L$:*

$$
\nabla f_\mu\left(x\right) = \mathbb{E}_{\mathbf{u}} \left[ \hat{\nabla} f(x) \right]
\tag{26}
$$

*where $\mathbf{u}$ is drawn from the uniform distribution over the unit Euclidean sphere, and $\hat{\nabla} f(x) = \frac{d}{\mu} \left[ f\left(x + \mu \mathbf{u}\right) - f\left(x\right) \right] u$ is the gradient estimator, $f_\mu(x) = \mathbb{E}_{\mathbf{u}} \left[ f(x + \mu \mathbf{u}) \right]$ is the smooth approximation of $f$.*

*2) For any $x \in \mathbb{R}^d$,*

$$
\left| f_\mu\left(x\right) - f\left(x\right) \right| \leq \frac{L\mu^2}{2}
\tag{27}
$$

$$
\left\| \nabla f_\mu(x) - \nabla f(x) \right\|^2 \leq \frac{\mu^2 L^2 d^2}{4}
\tag{28}
$$

$$
\frac{1}{2} \left\| \nabla f\left(x\right) \right\|^2 - \frac{\mu^2 L^2 d^2}{4} \leq \left\| \nabla f_\mu\left(x\right) \right\|^2 \leq 2 \left\| \nabla f\left(x\right) \right\|^2 + \frac{\mu^2 L^2 d^2}{2}
\tag{29}
$$

*3) For any $x \in \mathbb{R}^d$,*

$$
\mathbb{E}_{\mathbf{u}} \left[ \left\| \hat{\nabla} f\left(x\right) \right\|^2 \right] \leq 2d \left\| \nabla f\left(x\right) \right\|^2 + \frac{\mu^2 L^2 d^2}{2}
\tag{30}
$$

Lemma D.3 helps build a connection between $f\left(\cdot\right)$ and its smooth approximation $f_{\mu_m}\left(\cdot\right)$ of the convergence analysis. Proof of this lemma is provided in [14, 9].

**Lemma D.4.** *Compression Error. Under assumption B.5 - B.6, the norm of the difference between the loss function value with and without compression error is bounded:*

$$
\mathbb{E} \left\| \nabla_{w_0} \hat{f}_i\left(w_0^t, \Phi_i^t\right) - \nabla_{w_0} f_i\left(w_0^t, \Phi_i^t\right) \right\|^2 \leq H_0^2 \mathcal{E}
$$

$$
\mathbb{E} \left\| \nabla_{w_m} \hat{f}_i\left(w_0^t, \Phi_i^t\right) - \nabla_{w_m} f_i\left(w_0^t, \Phi_i^t\right) \right\|^2 \leq \mathbf{G}_m^2 H_m^2 \mathcal{E}
$$

$$
\tag{31}
$$

*where $\mathcal{E} = \max_t \left\{ \mathcal{E}^t \right\} = \max_t \left\{ \sum_{m=1}^M \mathcal{E}_m^t \right\}$*

*proof:*

For notation brevity, we denote:

$$h_{m,i}^t = h_m(w_m^t; x_{m,i})$$
$$\hat{f}_i^t = f_i(w_0^t, c_{1,i}^t, \cdots, c_{q,i}^t) = f_i(w_0, h_{1,i}^t + \epsilon_{1,i}, \cdots, h_{M,i}^t + \epsilon_{M,i})$$
$$f_i{}^t = f_i(w_0^t, h_{1,i}^t, \cdots, h_{M,i}^t) = f_i\left(w_0^t, \Phi_i^t\right)$$
$$E_m^t = [\epsilon_0^t, \epsilon_1^t, \cdots, \epsilon_M^t]$$

$$(32)$$

Applying the chain rule to $\nabla f_i$ w.r.t. $w_m$.

$$\nabla_{w_m} \hat{f}_i^t = \nabla_{w_m} h_{m,i}^t \underbrace{\nabla_{h_{m,i}^t} \hat{f}_i^t}_{a)}$$

$$= \nabla_{w_m} h_{m,i}^t \left(\nabla_{h_{m,i}^t} f_i^t + R_0^t\right)$$

$$= \nabla_{w_m} f_i^t + \nabla_{w_m} h_{m,i}^t \cdot R_0^t \qquad (33)$$

where a) applies Taylor expansion to $\nabla_{h_{m,i}} \hat{f}_i$ around the point $[w_0^t, \Phi_i^t]$

$$\nabla_{h_{m,i}} \hat{f}_i = \nabla_{h_{m,i}} f_i + \nabla_{h_{m,i}}^2 f_i E_m + \cdots$$

$$(34)$$

Define the infinite sum of all terms higher than the second partial derivative as $R_n^t$, where $n$ denotes the number of terms in the remainder $R$:

$$R_n^t(w_0, h_{1,i} + \epsilon_{1,i}, \cdots, h_{M,i} + \epsilon_{M,i}) = \nabla_{h_{m,i}}^2 f_i E_m + \cdots$$

$$(35)$$

Therefore, for $m = [M]$,

$$\mathbb{E}\left\|\nabla_{w_m} \hat{f}_i\left(w_0^t, \Phi_i^t\right) - \nabla_{w_m} f_i\left(w_0^t, \Phi_i^t\right)\right\|^2$$

$$= \mathbb{E}\left\|\nabla_{w_m} h_{m,i}^t \cdot R_0^t\right\|^2$$

$$\leq \mathbb{E}\left\|\nabla_{w_m} h_{m,i}^t\right\|^2 \left\|R_0^t\right\|^2$$

$$\overset{1)}{\leq} \mathbb{E}\mathbf{G}_m^2 \left\|R_0^t\right\|^2$$

$$\overset{2)}{\leq} \mathbf{G}_m^2 H_m^2 \mathbb{E}\left\|E_m^t\right\|^2$$

$$\overset{3)}{=} \mathbf{G}_m^2 H_m^2 \sum_{m=1}^M \mathcal{E}_m^t$$

$$\overset{4)}{=} \mathbf{G}_m^2 H_m^2 \mathcal{E}^t$$

$$\overset{5)}{\leq} \mathbf{G}_m^2 H_m^2 \mathcal{E}$$

$$(36)$$

where 1) apply assumption B.6, 2) apply the assumption B.5 and Taylor's inequality. 3) note that the specific error to be plugged in is $E_m = [0, \epsilon_1^t, \cdots, \epsilon_M^t]$, i.e. the error for the $w_0$ is 0, 4) for brevity, we use $\mathcal{E}^t$ to denote $\sum_{m=1}^M \|\mathcal{E}_m^t\|^2$ 5) To make a succinct bound, let $\mathcal{E} = \max_t \{\mathcal{E}^t\}$ denote the maximum norm of the error caused by the compression.

For the server $\mathbb{E}\left\|\nabla_{w_0} \hat{f}_i\left(w_0^t, \Phi_i^t\right) - \nabla_{w_0} f_i\left(w_0^t, \Phi_i^t\right)\right\|^2$, the $w_0$ in $f_i(w_0, \Phi_i)$ can be regarded as $w_0$ applying the identical function $h_0(\cdot)$, i.e. $h_0(w_0) = w_0$, and passing through the same procedures above. Note that $\|\nabla_{w_0} h_0(w_0)\|^2 = 1$.

$$\mathbb{E}\left\|\nabla_{w_0} \hat{f}_i\left(w_0^t, \Phi_i^t\right) - \nabla_{w_0} f_i\left(w_0^t, \Phi_i^t\right)\right\|^2$$

$$=\mathbb{E}\left\|\nabla_{w_0}h_0^t \cdot R_0^t\right\|^2$$

$$\leq\mathbb{E}\left\|\nabla_{w_0}h_0^t\right\|^2\left\|R_0^t\right\|^2$$

$$\leq\mathbb{E}\left\|R_0^t\right\|^2$$

$$\leq H_0^2\mathbb{E}\left\|E_m^t\right\|^2$$

$$=H_0^2\sum_{m=1}^{M}\mathcal{E}_m^t$$

$$=H_0^2\mathcal{E}^t$$

$$\leq H_0^2\mathcal{E}$$

$$(37)$$

## D.4   Bound the Global Update Round

In one global round during training, the client $m_t$ is activated, and the server and the client $m_t$ update one step. Taking expectations w.r.t. the sample $i$ and the random direction $u$ for the zeroth-order optimization in one global update round.

$$\mathbb{E}_{i,u}\left[f\left(w_0^{t+1},w_1^t,\cdots,w_{m_t}^{t+1},\cdots,w_M^t\right)-f\left(w_0^t,w_1^t,\cdots,w_{m_t}^t,\cdots,w_M^t\right)\right]$$

$$\overset{1)}{\leq}\underbrace{-\eta_0\mathbb{E}_i\left\langle\nabla_{w_0}f\left(w_0^t,\mathbf{w}^t\right),\nabla_{w_0}\hat{f}_i\left(w_0^t,\tilde{\Phi}_i^t\right)\right\rangle}_{a)}$$

$$\underbrace{+\frac{1}{2}L\eta_0^2\mathbb{E}_i\left\|\nabla_{w_0}\hat{f}_i\left(w_0^t,\tilde{\Phi}_i^t\right)\right\|^2}_{b)}$$

$$\underbrace{-\eta_{m_t}\mathbb{E}_{i,u}\left\langle\nabla_{w_{m_t}}f\left(w_0^t,\mathbf{w}^t\right),\hat{\nabla}_{h_{m_t}}\hat{f}_i\left(w_0^t,\tilde{\Phi}_i(w_{m_t})\right)\nabla_{w_{m_t}}h_{m_t}(w_{m_t}^t;x_{m_t,i})\right\rangle}_{c)}$$

$$\underbrace{+\frac{1}{2}L\eta_{m_t}^2\mathbb{E}_{i,u}\left\|\hat{\nabla}_{h_{m_t}}\hat{f}_i\left(w_0^t,\tilde{\Phi}(w_{m_t}^t)\right)\nabla_{w_{m_t}}h_{m_t}(w_{m_t}^t;x_{m_t,i})\right\|^2}_{d)}$$

$$\overset{2)}{\leq}-\frac{1}{2}\eta_0\mathbb{E}_i\left\|\nabla_{w_0}f\left(w_0^t,\mathbf{w}^t\right)\right\|^2+\eta_0 H_0^2\mathcal{E}+\eta_0 L_0^2\mathbb{E}_i\left\|\tilde{\mathbf{w}}^t-\mathbf{w}^t\right\|^2$$

$$+L\eta_0^2\mathbb{E}_i\left\|\nabla_{w_0}f\left(w_0^t,\mathbf{w}^t\right)\right\|^2+L\eta_0^2\sigma_0^2+2L\eta_0^2 H_0^2\mathcal{E}+2L\eta_0^2 L_0^2\mathbb{E}_i\left\|\tilde{\mathbf{w}}^t-\mathbf{w}^t\right\|^2$$

$$-\frac{1}{2}\eta_{m_t}\left\|\nabla_{w_{m_t}}f\left(\cdot\right)\right\|^2+\frac{1}{4}\eta_{m_t}\mu_{m_t}^2 L_{m_t}^2 d_{h_{m_t}}^2\mathbf{G}_{m_t}^2+4\eta_{m_t}\mathbf{G}_{m_t}^2 H_{m_t}^2\mathcal{E}+2\eta_{m_t}L_{m_t}^2\mathbb{E}_i\left\|\tilde{\mathbf{w}}^t-\mathbf{w}^t\right\|^2$$

$$+4\eta_{m_t}\Gamma\mathbf{G}_{m_t}^2$$

$$+2L\eta_{m_t}^2 d_{h_{m_t}}\mathbf{G}_0^2\mathbf{G}_{m_t}^2+\frac{1}{2}L\eta_{m_t}^2\mu_{m_t}^2 L_{m_t}^2 d_{h_{m_t}}^2\mathbf{G}_{m_t}^2+L\eta_{m_t}^2\mathbf{G}_{m_t}^2\Gamma$$

$$\overset{3)}{\leq}-\left(\frac{1}{2}\eta_0-L\eta_0^2\right)\mathbb{E}_i\left\|\nabla_{w_0}f\left(w_0^t,\mathbf{w}^t\right)\right\|^2-\frac{1}{2}\eta_{m_t}\mathbb{E}_i\left\|\nabla_{w_{m_t}}f\left(w_0^t,\mathbf{w}^t\right)\right\|^2$$

$$+\left(\eta_0 L_0^2+2L\eta_0^2 L_0^2+2\eta_{m_t}L_{m_t}^2\right)\mathbb{E}_i\left\|\tilde{\mathbf{w}}^t-\mathbf{w}^t\right\|^2$$

$$+\eta_0 H_0^2\mathcal{E}+2L\eta_0^2 H_0^2\mathcal{E}+4\eta_{m_t}\mathbf{G}_{m_t}^2 H_{m_t}^2\mathcal{E}+4\eta_{m_t}\mathbf{G}_{m_t}^2\Gamma+L\eta_{m_t}^2\mathbf{G}_{m_t}^2\Gamma$$

$$+\frac{1}{4}\eta_{m_t}\mu_{m_t}^2 L_{m_t}^2 d_{h_{m_t}}^2\mathbf{G}_{m_t}^2+2L\eta_{m_t}^2 d_{h_{m_t}}\mathbf{G}_0^2\mathbf{G}_{m_t}^2+\frac{1}{2}L\eta_{m_t}^2\mu_{m_t}^2 L_{m_t}^2 d_{h_{m_t}}^2\mathbf{G}_{m_t}^2$$

$$+L\eta_0^2\sigma_0^2$$

$$(38)$$

where 1) applies assumption B.2 (smoothness). Then we will discuss the server's update (a&b) and the client's update (c&d) separately in the following paragraph. 2) plugging in a), b), c) and d). 3) organize the equation.

for a)

$$
\begin{aligned}
& -\eta_0 \mathbb{E}_i \left\langle \nabla_{w_0} f\left(w_0^t, \mathbf{w}^t\right), \nabla_{w_0} \hat{f}_i\left(w_0^t, \tilde{\Phi}_i^t\right) \right\rangle \\
& \overset{0)}{=} -\eta_0 \mathbb{E}_i \left\langle \nabla_{w_0} f\left(w_0^t, \mathbf{w}^t\right), \nabla_{w_0} \hat{f}_i\left(\tilde{\Phi}_i^t\right) \right\rangle \\
& = -\eta_0 \mathbb{E}_i \left\langle \nabla_{w_0} f\left(w_0^t, \mathbf{w}^t\right), \nabla_{w_0} \hat{f}_i\left(\tilde{\Phi}_i^t\right) - \nabla_{w_0} f_i\left(\Phi_i^t\right) + \nabla_{w_0} f_i\left(\Phi_i^t\right) \right\rangle \\
& = \eta_0 \mathbb{E}_i \left\langle -\nabla_{w_0} f\left(w_0^t, \mathbf{w}^t\right), \nabla_{w_0} \hat{f}_i\left(\tilde{\Phi}_i^t\right) - \nabla_{w_0} f_i\left(\Phi_i^t\right) \right\rangle - \eta_0 \mathbb{E}_i \left\langle \nabla_{w_0} f\left(w_0^t, \mathbf{w}^t\right), \nabla_{w_0} f_i\left(\Phi_i^t\right) \right\rangle \\
& \overset{1)}{=} \eta_0 \mathbb{E}_i \left\langle -\nabla_{w_0} f\left(w_0^t, \mathbf{w}^t\right), \nabla_{w_0} \hat{f}_i\left(\tilde{\Phi}_i^t\right) - \nabla_{w_0} f_i\left(\Phi_i^t\right) \right\rangle - \eta_0 \mathbb{E}_i \left\| \nabla_{w_0} f\left(w_0^t, \mathbf{w}^t\right) \right\|^2 \\
& \overset{2)}{\leq} -\frac{1}{2} \eta_0 \mathbb{E}_i \left\| \nabla_{w_0} f\left(w_0^t, \mathbf{w}^t\right) \right\|^2 + \frac{1}{2} \eta_0 \mathbb{E}_i \left\| \nabla_{w_0} \hat{f}_i\left(\tilde{\Phi}_i^t\right) - \nabla_{w_0} f_i\left(\Phi_i^t\right) \right\|^2 \\
& = -\frac{1}{2} \eta_0 \mathbb{E}_i \left\| \nabla_{w_0} f\left(w_0^t, \mathbf{w}^t\right) \right\|^2 \\
& \quad + \frac{1}{2} \eta_0 \mathbb{E}_i \left\| \nabla_{w_0} \hat{f}_i\left(\tilde{\Phi}_i^t\right) - \nabla_{w_0} f_i\left(\tilde{\Phi}_i^t\right) + \nabla_{w_0} f_i\left(\tilde{\Phi}_i^t\right) - \nabla_{w_0} f_i\left(\Phi_i^t\right) \right\|^2 \\
& \overset{3)}{\leq} -\frac{1}{2} \eta_0 \mathbb{E}_i \left\| \nabla_{w_0} f\left(w_0^t, \mathbf{w}^t\right) \right\|^2 \\
& \quad + \eta_0 \mathbb{E}_i \left\| \nabla_{w_0} \hat{f}_i\left(\tilde{\Phi}_i^t\right) - \nabla_{w_0} f_i\left(\tilde{\Phi}_i^t\right) \right\|^2 + \eta_0 \mathbb{E}_i \left\| \nabla_{w_0} f_i\left(\tilde{\Phi}_i^t\right) - \nabla_{w_0} f_i\left(\Phi_i^t\right) \right\|^2 \\
& \overset{4)}{\leq} -\frac{1}{2} \eta_0 \mathbb{E}_i \left\| \nabla_{w_0} f\left(w_0^t, \mathbf{w}^t\right) \right\|^2 + \eta_0 H_0^2 \mathcal{E} + \eta_0 L_0^2 \mathbb{E}_i \left\| \tilde{\mathbf{w}}^t - \mathbf{w}^t \right\|^2
\end{aligned}
$$

$$(39)$$

where 0) For notation brevity, we mark $f_i\left(w_0^t, \tilde{\Phi}_i^t\right)$ as $f_i\left(\tilde{\Phi}_i^t\right)$, (omit the common parameters), 1) applies assumption B.3 (unbiased gradient), 2) $\langle a, b \rangle \leq \frac{1}{2}\|a\|^2 + \frac{1}{2}\|b\|^2$, 3) $\|a + b\|^2 \leq 2\|a\|^2 + 2\|b\|^2$, 4) lemma D.4 and assumption B.2 (smoothness).

for b)

$$
\begin{aligned}
& \frac{1}{2} \mathbb{E}_i \left\| \nabla_{w_0} \hat{f}_i\left(w_0^t, \tilde{\Phi}_i^t\right) \right\|^2 \\
& \overset{0)}{=} \frac{1}{2} \mathbb{E}_i \left\| \nabla_{w_0} \hat{f}_i\left(\tilde{\Phi}_i^t\right) - \nabla_{w_0} f_i\left(\Phi_i^t\right) + \nabla_{w_0} f_i\left(\Phi_i^t\right) \right\|^2 \\
& \overset{1)}{\leq} \mathbb{E}_i \left\| \nabla_{w_0} \hat{f}_i\left(\tilde{\Phi}_i^t\right) - \nabla_{w_0} f_i\left(\Phi_i^t\right) \right\|^2 + \mathbb{E}_i \left\| \nabla_{w_0} f_i\left(\Phi_i^t\right) \right\|^2 \\
& \overset{2)}{\leq} \mathbb{E}_i \left\| \nabla_{w_0} \hat{f}_i\left(\tilde{\Phi}_i^t\right) - \nabla_{w_0} f_i\left(\Phi_i^t\right) \right\|^2 + \mathbb{E}_i \left\| \nabla_{w_0} f\left(\mathbf{w}^t\right) \right\|^2 + \sigma_0^2 \\
& = \mathbb{E}_i \left\| \nabla_{w_0} f\left(\mathbf{w}^t\right) \right\|^2 + \sigma_0^2 \\
& \quad + \mathbb{E}_i \left\| \nabla_{w_0} \hat{f}_i\left(\tilde{\Phi}_i^t\right) - \nabla_{w_0} f_i\left(\tilde{\Phi}_i^t\right) + \nabla_{w_0} f_i\left(\tilde{\Phi}_i^t\right) - \nabla_{w_0} f_i\left(\Phi_i^t\right) \right\|^2 \\
& \overset{3)}{\leq} \mathbb{E}_i \left\| \nabla_{w_0} f\left(\mathbf{w}^t\right) \right\|^2 + \sigma_0^2 \\
& \quad + 2 \mathbb{E}_i \left\| \nabla_{w_0} \hat{f}_i\left(\tilde{\Phi}_i^t\right) - \nabla_{w_0} f_i\left(\tilde{\Phi}_i^t\right) \right\|^2 + 2 \mathbb{E}_i \left\| \nabla_{w_0} f_i\left(\tilde{\Phi}_i^t\right) - \nabla_{w_0} f_i\left(\Phi_i^t\right) \right\|^2 \\
& \overset{4)}{\leq} \mathbb{E}_i \left\| \nabla_{w_0} f\left(\mathbf{w}^t\right) \right\|^2 + \sigma_0^2 + 2 H_0^2 \mathcal{E} + 2 L_0^2 \mathbb{E}_i \left\| \tilde{\mathbf{w}}^t - \mathbf{w}^t \right\|^2
\end{aligned}
$$

$$(40)$$

where 0) For notation brevity, we mark $f_i\left(w_0^t, \tilde{\Phi}_i^t\right)$ as $f_i\left(\tilde{\Phi}_i^t\right)$, (omit the common parameters), 1) applies $\|a + b\|^2 \leq 2\|a\|^2 + 2\|b\|^2$, 2) applies $\mathbb{E}(X^2) = \mathbb{E}(X)^2 + \text{Var}(X)$, assumption B.3 (unbiased gradient) and assumption B.4 (bounded variance), i.e. $\mathbb{E}_i \left\| \nabla_{w_0} f_i\left(\Phi_i^t\right) \right\|^2 =$

247  $\left\|\nabla_{w_0} f(\Phi_i^t)\right\|^2 + \mathrm{Var}(\nabla_{w_0} f_i(\Phi_i)) \leq \left\|\nabla_{w_0} f(\mathbf{w}^t)\right\|^2 + \sigma_0^2$, 3) applies $\|a+b\|^2 \leq 2\|a\|^2 + 2\|b\|^2$,
248  4) lemma D.4 and assumption B.2 (smoothness).

249  then b) is

$$L\eta_0^2 \left( \frac{1}{2} \mathbb{E}_i \left\| \nabla_{w_0} \hat{f}_i \left( w_0^t, \tilde{\Phi}_i^t \right) \right\|^2 \right)$$
$$= L\eta_0^2 \mathbb{E}_i \left\| \nabla_{w_0} f\left(\mathbf{w}^t\right) \right\|^2 + L\eta_0^2 \sigma_0^2 + 2L\eta_0^2 H_0^2 \mathcal{E} + 2L\eta_0^2 L_0^2 \mathbb{E}_i \left\| \tilde{\mathbf{w}}^t - \mathbf{w}^t \right\|^2$$

$$(41)$$

250  for c)

$$-\eta_{m_t} \mathbb{E}_{i,u} \left\langle \nabla_{w_{m_t}} f\left(w_0^t, \mathbf{w}^t\right), \left[\hat{\nabla}_{h_{m_t,i}^t} \hat{f}_i\left(w_0^t, \Phi_i(w_{m_t}^t)\right) + \gamma^t\right] \cdot \nabla_{w_{m_t}} h_{m_t}(w_{m_t}^t; x_{m,i}) \right\rangle$$

$$\overset{0)}{=} -\eta_{m_t} \mathbb{E}_{i,u} \left\langle \nabla_{w_{m_t}} f(\cdot), \left[\hat{\nabla}_{h_{m_t,i}^t} \hat{f}_i\left(\tilde{\Phi}_i(w_{m_t}^t)\right) + \gamma^t\right] \cdot \nabla_{w_{m_t}} h_{m_t,i}^t \right\rangle$$

$$\overset{1)}{=} -\eta_{m_t} \mathbb{E}_i \left\langle \nabla_{w_{m_t}} f(\cdot), \nabla_{h_{m_t,i}^t} \hat{f}_{\mu_{m_t},i}\left(\tilde{\Phi}_i(w_{m_t}^t)\right) \cdot \nabla_{w_{m_t}} h_{m_t,i}^t + \gamma^t \cdot \nabla_{w_{m_t}} h_{m_t,i}^t \right\rangle$$

$$= -\eta_{m_t} \mathbb{E}_i \left\langle \nabla_{w_{m_t}} f(\cdot), \nabla_{h_{m_t,i}^t} \hat{f}_{\mu_{m_t},i}\left(\tilde{\Phi}_i(w_{m_t}^t)\right) \cdot \nabla_{w_{m_t}} h_{m_t,i}^t + \gamma^t \cdot \nabla_{w_{m_t}} h_{m_t,i}^t - \nabla_{w_{m_t}} f_i\left(\Phi_i(w_{m_t}^t)\right) \right.$$
$$\left. + \nabla_{w_{m_t}} f_i\left(\Phi_i(w_{m_t}^t)\right) \right\rangle$$

$$= \eta_{m_t} \mathbb{E}_i \left\langle -\nabla_{w_{m_t}} f(\cdot), \nabla_{h_{m_t,i}^t} \hat{f}_{\mu_{m_t},i}\left(\tilde{\Phi}_i(w_{m_t}^t)\right) \cdot \nabla_{w_{m_t}} h_{m_t,i}^t - \nabla_{w_{m_t}} f_i\left(\Phi_i(w_{m_t}^t)\right) + \gamma^t \cdot \nabla_{w_{m_t}} h_{m_t,i}^t \right\rangle$$
$$- \eta_{m_t} \mathbb{E}_i \left\langle \nabla_{w_{m_t}} f(\cdot), \nabla_{w_{m_t}} f_i\left(\Phi_i(w_{m_t}^t)\right) \right\rangle$$

$$\overset{2)}{=} \eta_{m_t} \mathbb{E}_i \left\langle -\nabla_{w_{m_t}} f(\cdot), \nabla_{h_{m_t,i}^t} \hat{f}_{\mu_{m_t},i}\left(\tilde{\Phi}_i(w_{m_t}^t)\right) \cdot \nabla_{w_{m_t}} h_{m_t,i}^t - \nabla_{w_{m_t}} f_i\left(\Phi_i(w_{m_t}^t)\right) + \gamma^t \cdot \nabla_{w_{m_t}} h_{m_t,i}^t \right\rangle$$
$$- \eta_{m_t} \mathbb{E}_i \left\| \nabla_{w_{m_t}} f(\cdot) \right\|^2$$

$$\overset{3)}{\leq} -\frac{1}{2} \eta_{m_t} \left\| \nabla_{w_{m_t}} f(\cdot) \right\|^2$$
$$+ \frac{1}{2} \eta_{m_t} \mathbb{E}_i \left\| \nabla_{h_{m_t,i}^t} \hat{f}_{\mu_{m_t},i}\left(\tilde{\Phi}_i(w_{m_t}^t)\right) \cdot \nabla_{w_{m_t}} h_{m_t,i}^t - \nabla_{w_{m_t}} f_i\left(\Phi_i(w_{m_t}^t)\right) + \gamma^t \cdot \nabla_{w_{m_t}} h_{m_t,i}^t \right\|^2$$

$$= -\frac{1}{2} \eta_{m_t} \left\| \nabla_{w_{m_t}} f(\cdot) \right\|^2$$
$$+ \frac{1}{2} \eta_{m_t} \mathbb{E}_i \| \nabla_{h_{m_t,i}^t} \hat{f}_{\mu_{m_t},i}\left(\tilde{\Phi}_i(w_{m_t}^t)\right) \cdot \nabla_{w_{m_t}} h_{m_t,i}^t - \nabla_{w_{m_t}} \hat{f}_i\left(\tilde{\Phi}_i(w_{m_t}^t)\right) + \nabla_{w_{m_t}} \hat{f}_i\left(\tilde{\Phi}_i(w_{m_t}^t)\right)$$
$$- \nabla_{w_{m_t}} f_i\left(\tilde{\Phi}_i(w_{m_t}^t)\right) + \nabla_{w_{m_t}} f_i\left(\tilde{\Phi}_i(w_{m_t}^t)\right) - \nabla_{w_{m_t}} f_i\left(\Phi_i(w_{m_t}^t)\right) + \gamma^t \cdot \nabla_{w_{m_t}} h_{m_t,i}^t \|^2$$

$$\overset{4)}{\leq} -\frac{1}{2} \eta_{m_t} \left\| \nabla_{w_{m_t}} f(\cdot) \right\|^2$$
$$+ \eta_{m_t} \mathbb{E}_i \left\| \nabla_{h_{m_t,i}^t} \hat{f}_{\mu_{m_t},i}\left(\tilde{\Phi}_i(w_{m_t}^t)\right) \cdot \nabla_{w_{m_t}} h_{m_t,i}^t - \nabla_{w_{m_t}} \hat{f}_i\left(\tilde{\Phi}_i(w_{m_t}^t)\right) \right\|^2$$
$$+ 4\eta_{m_t} \mathbb{E}_i \left\| \nabla_{w_{m_t}} \hat{f}_i\left(\tilde{\Phi}_i(w_{m_t}^t)\right) - \nabla_{w_{m_t}} f_i\left(\tilde{\Phi}_i(w_{m_t}^t)\right) \right\|^2 + 4\eta_{m_t} \mathbb{E}_i \left\| \gamma^t \cdot \nabla_{w_{m_t}} h_{m_t,i}^t \right\|^2$$
$$+ 2\eta_{m_t} \mathbb{E}_i \left\| \nabla_{w_{m_t}} f_i\left(\tilde{\Phi}_i(w_{m_t}^t)\right) - \nabla_{w_{m_t}} f_i\left(\Phi_i(w_{m_t}^t)\right) \right\|^2$$

$$\leq -\frac{1}{2} \eta_{m_t} \left\| \nabla_{w_{m_t}} f(\cdot) \right\|^2$$
$$+ \eta_{m_t} \mathbb{E}_i \left\| \nabla_{h_{m_t,i}^t} \hat{f}_{\mu_{m_t},i}\left(\tilde{\Phi}_i(w_{m_t}^t)\right) - \nabla_{h_{m_t,i}^t} \hat{f}_i\left(\tilde{\Phi}_i(w_{m_t}^t)\right) \right\|^2 \left\| \nabla_{w_{m_t}} h_{m_t,i}^t \right\|^2$$
$$+ 4\eta_{m_t} \mathbb{E}_i \left\| \nabla_{w_{m_t}} \hat{f}_i\left(\tilde{\Phi}_i(w_{m_t}^t)\right) - \nabla_{w_{m_t}} f_i\left(\tilde{\Phi}_i(w_{m_t}^t)\right) \right\|^2 + 4\eta_{m_t} \mathbb{E}_i \left\| \gamma^t \cdot \nabla_{w_{m_t}} h_{m_t,i}^t \right\|^2$$
$$+ 2\eta_{m_t} \mathbb{E}_i \left\| \nabla_{w_{m_t}} f_i\left(\tilde{\Phi}_i(w_{m_t}^t)\right) - \nabla_{w_{m_t}} f_i\left(\Phi_i(w_{m_t}^t)\right) \right\|^2$$

$$\overset{5)}{\leq} -\frac{1}{2} \eta_{m_t} \left\| \nabla_{w_{m_t}} f(\cdot) \right\|^2 + \eta_{m_t} \frac{\mu_{m_t}^2 L_{m_t}^2 d_{h_{m_t}}^2}{4} \mathbf{G}_{m_t}^2$$

$$+ 4\eta_{m_t} \mathbb{E}_i \left\| \nabla_{w_{m_t}} \hat{f}_i \left( \tilde{\Phi}_i(w_{m_t}^t) \right) - \nabla_{w_{m_t}} f_i \left( \tilde{\Phi}_i(w_{m_t}^t) \right) \right\|^2 + 4\eta_{m_t} \mathbb{E}_i \left\| \gamma^t \cdot \nabla_{w_{m_t}} h_{m_t,i}^t \right\|^2$$

$$+ 2\eta_{m_t} \mathbb{E}_i \left\| \nabla_{w_{m_t}} f_i \left( \tilde{\Phi}_i(w_{m_t}^t) \right) - \nabla_{w_{m_t}} f_i \left( \Phi_i(w_{m_t}^t) \right) \right\|^2$$

$$\overset{6)}{\leq} -\frac{1}{2}\eta_{m_t} \left\| \nabla_{w_{m_t}} f(\cdot) \right\|^2 + \frac{1}{4}\eta_{m_t} \mu_{m_t}^2 L_{m_t}^2 d_{h_{m_t}}^2 \mathbf{G}_{m_t}^2 + 4\eta_{m_t} \mathbf{G}_{m_t}^2 H_{m_t}^2 \mathcal{E} + 2\eta_{m_t} L_{m_t}^2 \mathbb{E}_i \left\| \tilde{\mathbf{w}}^t - \mathbf{w}^t \right\|^2$$

$$+ 4\eta_{m_t} \mathbb{E}_i \left\| \gamma^t \cdot \nabla_{w_{m_t}} h_{m_t,i}^t \right\|^2$$

$$\leq -\frac{1}{2}\eta_{m_t} \left\| \nabla_{w_{m_t}} f(\cdot) \right\|^2 + \frac{1}{4}\eta_{m_t} \mu_{m_t}^2 L_{m_t}^2 d_{h_{m_t}}^2 \mathbf{G}_{m_t}^2 + 4\eta_{m_t} \mathbf{G}_{m_t}^2 H_{m_t}^2 \mathcal{E} + 2\eta_{m_t} L_{m_t}^2 \mathbb{E}_i \left\| \tilde{\mathbf{w}}^t - \mathbf{w}^t \right\|^2$$

$$+ 4\eta_{m_t} \mathbb{E}_i \left\| \gamma^t \right\|^2 \left\| \nabla_{w_{m_t}} h_{m_t,i}^t \right\|^2$$

$$\overset{7)}{\leq} -\frac{1}{2}\eta_{m_t} \left\| \nabla_{w_{m_t}} f(\cdot) \right\|^2 + \frac{1}{4}\eta_{m_t} \mu_{m_t}^2 L_{m_t}^2 d_{h_{m_t}}^2 \mathbf{G}_{m_t}^2 + 4\eta_{m_t} \mathbf{G}_{m_t}^2 H_{m_t}^2 \mathcal{E} + 2\eta_{m_t} L_{m_t}^2 \mathbb{E}_i \left\| \tilde{\mathbf{w}}^t - \mathbf{w}^t \right\|^2$$

$$+ 4\eta_{m_t} \Gamma \mathbf{G}_{m_t}^2$$

$$(42)$$

where 0) for notation brevity, we omit the common parameters, $f(w_0^t, \mathbf{w}^t) = f(\cdot)$, $\hat{f}_i \left( w_0^t, \tilde{\Phi}_i(w_{m_t}^t) \right) = \hat{f}_i \left( \tilde{\Phi}_i(w_{m_t}^t) \right)$, $h_{m_t}(w_{m_t}^t; x_{m_t,i}) = h_{m_t,i}^t$, 1) applies Eq.26 in lemma D.3, 2) applies assumption B.3 (unbiased gradient) and we use $f\left( w_{m_t}^t \right)$ to denote $f\left( w_0^{t,0}, w_1^{t,0} \cdots w_{m_t}^{t,0}, \cdots, w_M^{t,0} \right)$, 3) $\langle a, b \rangle \leq \frac{1}{2}\|a\|^2 + \frac{1}{2}\|b\|^2$, 4) applying $\|a+b\|^2 \leq 2\|a\|^2 + 2\|b\|^2$ recursively, 5) applies Eq. 28 in lemma D.3 and the assumption B.6 (bounded block-coordinated gradient), 6) applies lemma D.4 and assumption B.2. 7) applies assumption B.6 (bounded block-coordinated gradient) and definition D.2.

for d)

$$\frac{1}{2}\mathbb{E}_{i,u} \left\| \left[ \hat{\nabla}_{h_{m_t}} \hat{f}_i \left( w_0^t, \tilde{\Phi}(w_{m_t}^t) \right) + \gamma^t \right] \cdot \nabla_{w_{m_t}} h_{m_t}(w_{m_t}^t; x_{m_t,i}) \right\|^2$$

$$\overset{0)}{=} \frac{1}{2}\mathbb{E}_{i,u} \left\| \left[ \hat{\nabla}_{h_{m_t}} \hat{f}_i \left( \tilde{\Phi}(w_{m_t}^t) \right) + \gamma^t \right] \cdot \nabla_{w_{m_t}} h_{m_t,i}^t \right\|^2$$

$$\leq \frac{1}{2}\mathbb{E}_{i,u} \left\| \hat{\nabla}_{h_{m_t}} \hat{f}_i \left( \tilde{\Phi}(w_{m_t}^t) \right) + \gamma^t \right\|^2 \left\| \nabla_{w_{m_t}} h_{m_t,i}^t \right\|^2$$

$$\overset{1)}{\leq} \mathbb{E}_{i,u} \left\| \hat{\nabla}_{h_{m_t}} \hat{f}_i \left( \tilde{\Phi}(w_{m_t}^t) \right) \right\|^2 \left\| \nabla_{w_{m_t}} h_{m_t,i}^t \right\|^2 + \mathbb{E}_{i,u} \left\| \gamma^t \right\|^2 \left\| \nabla_{w_{m_t}} h_{m_t,i}^t \right\|^2$$

$$\overset{2)}{\leq} \mathbf{G}_{m_t}^2 \mathbb{E}_{i,u} \left\| \hat{\nabla}_{h_{m_t}} \hat{f}_i \left( \tilde{\Phi}(w_{m_t}^t) \right) \right\|^2 + \mathbb{E}_{i,u} \left\| \gamma^t \right\|^2 \mathbf{G}_{m_t}^2$$

$$\overset{3)}{\leq} \mathbf{G}_{m_t}^2 \left( 2d_{h_{m_t}} \left\| \nabla_{h_{m_t}} \hat{f}_i \left( \tilde{\Phi}(w_{m_t}^t) \right) \right\|^2 + \frac{1}{2}\mu_{m_t}^2 L_{m_t}^2 d_{h_{m_t}}^2 \right) + \mathbb{E}_{i,u} \left\| \gamma^t \right\|^2 \mathbf{G}_{m_t}^2$$

$$= 2\mathbf{G}_{m_t}^2 d_{h_{m_t}} \left\| \nabla_{h_{m_t}} \hat{f}_i \left( \tilde{\Phi}(w_{m_t}^t) \right) \right\|^2 + \frac{1}{2}\mu_{m_t}^2 L_{m_t}^2 d_{h_{m_t}}^2 \mathbf{G}_{m_t}^2 + \mathbb{E}_{i,u} \left\| \gamma^t \right\|^2 \mathbf{G}_{m_t}^2$$

$$\overset{4)}{\leq} 2d_{h_{m_t}} \mathbf{G}_0^2 \mathbf{G}_{m_t}^2 + \frac{1}{2}\mu_{m_t}^2 L_{m_t}^2 d_{h_{m_t}}^2 \mathbf{G}_{m_t}^2 + \Gamma \mathbf{G}_{m_t}^2$$

$$(43)$$

where 0) for notation brevity, we omit the common parameters, i.e. $f(w_0^t, \mathbf{w}^t) = f(\cdot)$, $\hat{f}_i \left( w_0^t, \tilde{\Phi}_i(w_{m_t}^t) \right) = \hat{f}_i \left( \tilde{\Phi}_i(w_{m_t}^t) \right)$, $h_{m_t}(w_{m_t}^t; x_{m_t,i}) = h_{m_t,i}^t$, 1) applies $\|a+b\|^2 \leq 2\|a\|^2 + 2\|b\|^2$, 2) applies assumption B.6 (bounded block-coordinated gradient), 3) applies Eq. 30 in lemma D.3, 4) $\Gamma = \max_t \{\Gamma^t\}$.

Then d) is

$$L\eta_{m_t}^2 \left( \frac{1}{2}\mathbb{E}_{i,u} \left\| \hat{\nabla}_{h_{m_t}} \hat{f}_i \left( w_0^t, \tilde{\Phi}(w_{m_t}^t) \right) \nabla_{w_{m_t}} h_{m_t}(w_{m_t}^t; x_{m_t,i}) \right\|^2 \right)$$

$$\leq 2L\eta_{m_t}^2 d_{h_{m_t}} \mathbf{G}_0^2 \mathbf{G}_{m_t}^2 + \frac{1}{2}L\eta_{m_t}^2 \mu_{m_t}^2 L_{m_t}^2 d_{h_{m_t}}^2 \mathbf{G}_{m_t}^2 + L\eta_{m_t}^2 \Gamma \mathbf{G}_{m_t}^2$$

## D.5 Combine the Gradient

Start with the Eq. 38, additionally taking expectation w.r.t. activated client $m_t$, and applying the assumption B.7 (independent client).

$$\mathbb{E}_{m_t,i,u} \left[ f \left( w_0^{t+1}, w_1^t, \cdots, w_{m_t}^{t+1}, \cdots, w_M^t \right) - f \left( w_0^t, w_1^t, \cdots, w_{m_t}^t, \cdots, w_M^t \right) \right]$$

$$\leq - \left( \frac{1}{2} \eta_0 - L \eta_0^2 \right) \mathbb{E}_i \left\| \nabla_{w_0} f \left( w_0^t, \mathbf{w}^t \right) \right\|^2 - \frac{1}{2} \sum_{m=1}^{M} p_m \eta_m \mathbb{E}_i \left\| \nabla_{w_m} f \left( w_0^t, \mathbf{w}^t \right) \right\|^2$$

$$+ \left( \eta_0 L_0^2 + 2 L \eta_0^2 L_0^2 + 2 \sum_{m=1}^{M} p_m \eta_m L_m^2 \right) \mathbb{E}_i \left\| \tilde{\mathbf{w}}^t - \mathbf{w}^t \right\|^2$$

$$+ \eta_0 H_0^2 \mathcal{E} + 2 L \eta_0^2 H_0^2 \mathcal{E} + 4 \sum_{m=1}^{M} p_m \eta_m \mathbf{G}_m^2 H_m^2 \mathcal{E} + 4 \sum_{m=1}^{M} p_m \eta_m \mathbf{G}_m^2 \Gamma + L \sum_{m=1}^{M} p_m \eta_m^2 \mathbf{G}_m^2 \Gamma$$

$$+ \frac{1}{4} \sum_{m=1}^{M} p_m \eta_m \mu_m^2 L_m^2 d_{h_m}^2 \mathbf{G}_m^2 + 2 \sum_{m=1}^{M} p_m L \eta_m^2 d_{h_m} \mathbf{G}_0^2 \mathbf{G}_m^2 + \frac{1}{2} \sum_{m=1}^{M} p_m L \eta_m^2 \mu_m^2 L_m^2 d_{h_m}^2 \mathbf{G}_m^2$$

$$+ L \eta_0^2 \sigma_0^2$$

$$\overset{1)}{\leq} - \left( \frac{1}{2} \eta_0 - L \eta_0^2 \right) \mathbb{E}_i \left\| \nabla_{w_0} f \left( w_0^t, \mathbf{w}^t \right) \right\|^2 - \frac{1}{2} \sum_{m=1}^{M} p_m \eta_m \mathbb{E}_i \left\| \nabla_{w_m} f \left( w_0^t, \mathbf{w}^t \right) \right\|^2$$

$$+ \left( \eta_0 L_0^2 + 2 L \eta_0^2 L_0^2 + 2 \sum_{m=1}^{M} p_m \eta_m L_m^2 \right) \mathbb{E}_i \left\| \tilde{\mathbf{w}}^t - \mathbf{w}^t \right\|^2$$

$$+ Q_1$$

$$\overset{2)}{\leq} - \frac{1}{4} \eta_0 \mathbb{E}_i \left\| \nabla_{w_0} f \left( w_0^t, \mathbf{w}^t \right) \right\|^2 - \frac{1}{4} \sum_{m=1}^{M} p_m \eta_m \mathbb{E}_i \left\| \nabla_{w_m} f \left( w_0^t, \mathbf{w}^t \right) \right\|^2$$

$$+ \left( \eta_0 L_0^2 + 2 L \eta_0^2 L_0^2 + 2 \sum_{m=1}^{M} p_m \eta_m L_m^2 \right) \mathbb{E}_i \left\| \tilde{\mathbf{w}}^t - \mathbf{w}^t \right\|^2$$

$$+ Q_1$$

$$\overset{3)}{\leq} - \frac{1}{4} \min \left\{ \eta_0, p_m \eta_m \right\} \mathbb{E}_i \left\| \nabla f \left( w_0^t, \mathbf{w}^t \right) \right\|^2$$

$$+ \left( \eta_0 L_0^2 + 2 L \eta_0^2 L_0^2 + 2 \sum_{m=1}^{M} p_m \eta_m L_m^2 \right) \mathbb{E}_i \left\| \tilde{\mathbf{w}}^t - \mathbf{w}^t \right\|^2$$

$$+ Q_1$$

(45)

where 1) for notation brevity, denotes the line 3-5 (constants) as $Q_1$, 2) let $\eta_0 \leq \frac{1}{4L}$ then $-\frac{1}{2}\eta_0 + L\eta_0^2 < -\frac{1}{4}\eta_0$, and $-\frac{1}{2} \sum_{m=1}^{M} p_m \eta_m \mathbb{E}_i \left\| \nabla_{w_{m_t}} f \left( w_0^t, \mathbf{w}^t \right) \right\|^2 \leq -\frac{1}{4} \sum_{m=1}^{M} p_m \eta_m \mathbb{E}_i \left\| \nabla_{w_{m_t}} f \left( w_0^t, \mathbf{w}^t \right) \right\|^2$, 3) uses the orthogonality of $\nabla f$, i.e. $\left\| \nabla f \left( w_0, \mathbf{w} \right) \right\|^2 = \left\| \nabla_{w_0} f \left( w_0, \mathbf{w} \right) \right\|^2 + \sum_{m=1}^{M} \left\| \nabla_{w_m} f \left( w_0, \mathbf{w} \right) \right\|^2$.

## D.6 Define the Lyapunov Function to Eliminate the Client's Delay.

Define a Lyapunov function.

$$M^t = f \left( w_0^t, \mathbf{w}^t \right) + \sum_{i=1}^{\tau} \theta_i \left\| \mathbf{w}^{t+1-i} - \mathbf{w}^{t-i} \right\|^2 \tag{46}$$

Taking expectation w.r.t. the activated client $m_t$, sample index $i$, and the random direction $u$.

$$\mathbb{E}\left(M^{t+1} - M^t\right)$$

$$=\mathbb{E}\left[f\left(w_0^{t+1}, \mathbf{w}^{t+1}\right) + \sum_{i=1}^{\tau}\theta_i\left\|\mathbf{w}^{t+1+1-i} - \mathbf{w}^{t+1-i}\right\|^2\right] - \mathbb{E}\left[f\left(w_0^t, \mathbf{w}^t\right) + \sum_{i=1}^{\tau}\theta_i\left\|\mathbf{w}^{t+1-i} - \mathbf{w}^{t-i}\right\|^2\right]$$

$$=\mathbb{E}\left[f\left(w_0^{t+1}, \mathbf{w}^{t+1}\right) - f\left(w_0^t, \mathbf{w}^t\right)\right] + \sum_{i=1}^{\tau}\theta_i\mathbb{E}\left\|\mathbf{w}^{t+1+1-i} - \mathbf{w}^{t+1-i}\right\|^2 - \sum_{i=1}^{\tau}\theta_i\left\|\mathbf{w}^{t+1-i} - \mathbf{w}^{t-i}\right\|^2$$

$$\overset{1)}{\leq} -\frac{1}{4}\min\left\{\eta_0, p_m\eta_m\right\}\mathbb{E}\left\|\nabla f\left(w_0^t, \mathbf{w}^t\right)\right\|^2 + Q_1$$

$$+ \left(\eta_0 L_0^2 + 2L\eta_0^2 L_0^2 + 2\sum_{m=1}^{M}p_m\eta_m L_m^2\right)\underbrace{\mathbb{E}\left\|\tilde{\mathbf{w}}^t - \mathbf{w}^t\right\|^2}_{a)}$$

$$+ \underbrace{\sum_{i=1}^{\tau}\theta_i\mathbb{E}\left\|\mathbf{w}^{t+1+1-i} - \mathbf{w}^{t+1-i}\right\|^2 - \sum_{i=1}^{\tau}\theta_i\left\|\mathbf{w}^{t+1-i} - \mathbf{w}^{t-i}\right\|^2}_{b)}$$

$$\overset{2)}{\leq} -\frac{1}{4}\min\left\{\eta_0, p_m\eta_m\right\}\mathbb{E}\left\|\nabla f\left(w_0^t, \mathbf{w}^t\right)\right\|^2 + Q_1$$

$$+ \left(\eta_0 L_0^2 + 2L\eta_0^2 L_0^2 + 2\sum_{m=1}^{M}p_m\eta_m L_m^2\right)\tau\sum_{i=1}^{\tau}\mathbb{E}\left\|\mathbf{w}^{t+1-i} - \mathbf{w}^{t-i}\right\|^2$$

$$+ \theta_1\mathbb{E}\left\|\mathbf{w}^{t+1} - \mathbf{w}^t\right\|^2 + \sum_{i=1}^{\tau-1}(\theta_{i+1} - \theta_i)\mathbb{E}\left\|\mathbf{w}^{t+1-i} - \mathbf{w}^{t-i}\right\|^2 - \theta_\tau\mathbb{E}\left\|\mathbf{w}^{t+1-\tau} - \mathbf{w}^{t-\tau}\right\|^2$$

$$\leq -\frac{1}{4}\min\left\{\eta_0, p_m\eta_m\right\}\mathbb{E}\left\|\nabla f\left(w_0^t, \mathbf{w}^t\right)\right\|^2 + Q_1$$

$$+ \theta_1\mathbb{E}\left\|\mathbf{w}^{t+1} - \mathbf{w}^t\right\|^2$$

$$+ \sum_{i=1}^{\tau-1}\left(\theta_{i+1} - \theta_i + \eta_0 L_0^2 + 2L\eta_0^2 L_0^2 + 2\sum_{m=1}^{M}p_m\eta_m L_m^2\right)\mathbb{E}\left\|\mathbf{w}^{t+1-i} - \mathbf{w}^{t-i}\right\|^2$$

$$- \left[\theta_\tau - \left(\eta_0 L_0^2 + 2L\eta_0^2 L_0^2 + 2\sum_{m=1}^{M}p_m\eta_m L_m^2\right)\right]\mathbb{E}\left\|\mathbf{w}^{t+1-\tau} - \mathbf{w}^{t-\tau}\right\|^2$$

$$\tag{47}$$

where 1) plugging in Eq. 45, 2) plugging in a) and b).

For a) in Eq. 47:

$$\mathbb{E}\left\|\tilde{\mathbf{w}}^t - \mathbf{w}^t\right\|^2 \overset{1)}{\leq} \mathbb{E}\left\|\sum_{i=1}^{\tau}\left(\mathbf{w}^{i+1} - \mathbf{w}^i\right)\right\|^2 \overset{2)}{\leq} \tau\sum_{i=1}^{\tau}\mathbb{E}\left\|\mathbf{w}^{t+1-i} - \mathbf{w}^{t-i}\right\|^2 \tag{48}$$

where 1) applies assumption B.8 (uniformly bounded delay), 2) applies Cauchy-Schwarz inequality, i.e. $\left(\sum_{i=0}^{n-1}x_i\right)^2 = \left(\sum_{i=0}^{n-1}1\cdot x_i\right)^2 \leq n\sum_{i=0}^{n-1}x_i^2$.

For b) in Eq. 47:

$$\sum_{i=1}^{\tau}\theta_i\mathbb{E}\left\|\mathbf{w}^{t+1+1-i} - \mathbf{w}^{t+1-i}\right\|^2 - \sum_{i=1}^{\tau}\theta_i\mathbb{E}\left\|\mathbf{w}^{t+1-i} - \mathbf{w}^{t-i}\right\|^2$$

$$=\theta_1\mathbb{E}\left\|\mathbf{w}^{t+1} - \mathbf{w}^t\right\|^2 + \sum_{i=1}^{\tau-1}(\theta_{i+1} - \theta_i)\mathbb{E}\left\|\mathbf{w}^{t+1-i} - \mathbf{w}^{t-i}\right\|^2 - \theta_\tau\mathbb{E}\left\|\mathbf{w}^{t+1-\tau} - \mathbf{w}^{t-\tau}\right\|^2 \tag{49}$$

279 Let $\theta_1 \geq \tau \left( \eta_0 L_0^2 + 2L\eta_0^2 L_0^2 + 2 \sum_{m=1}^{M} p_m \eta_m L_m^2 \right)$ and design the recurrent relation for $\theta_i$

$$\theta_{i+1} = \theta_i - \left( \eta_0 L_0^2 + 2L\eta_0^2 L_0^2 + 2 \sum_{m=1}^{M} p_m \eta_m L_m^2 \right) \tag{50}$$

280 It follows that

$$\theta_\tau - \left( \eta_0 L_0^2 + 2L\eta_0^2 L_0^2 + 2 \sum_{m=1}^{M} p_m \eta_m L_m^2 \right) = \theta_1 - \tau \left( \eta_0 L_0^2 + 2L\eta_0^2 L_0^2 + 2 \sum_{m=1}^{M} p_m \eta_m L_m^2 \right) \geq 0 \tag{51}$$

281 Applying Eq. 50 and Eq. 51 to Eq. 47

$$
\begin{aligned}
&\mathbb{E}\left( M^{t+1} - M^t \right) \\
&\leq -\frac{1}{4} \min\{\eta_0, p_m \eta_m\} \mathbb{E} \left\| \nabla f\left(w_0^t, \mathbf{w}^t\right) \right\|^2 + Q_1 \\
&\quad + \tau \left( \eta_0 L_0^2 + 2L\eta_0^2 L_0^2 + 2 \sum_{m=1}^{M} p_m \eta_m L_m^2 \right) \underbrace{\mathbb{E} \left\| \mathbf{w}^{t+1} - \mathbf{w}^t \right\|^2}_{c)} \\
&\leq -\frac{1}{4} \min\{\eta_0, p_m \eta_m\} \mathbb{E} \left\| \nabla f\left(w_0^t, \mathbf{w}^t\right) \right\|^2 + Q_1 \\
&\quad + \tau \left( \eta_0 L_0^2 + 2L\eta_0^2 L_0^2 + 2 \sum_{m=1}^{M} p_m \eta_m L_m^2 \right) \left( 2 \sum_{m=1}^{M} p_m \eta_m^2 d_{h_m} \mathbf{G}_0^2 \mathbf{G}_m^2 \right. \\
&\qquad\qquad\qquad\qquad\qquad \left. + \frac{1}{2} \sum_{m=1}^{M} p_m \eta_m^2 \mu_m^2 L_m^2 d_{h_m}^2 \mathbf{G}_m^2 + \sum_{m=1}^{M} p_m \eta_m^2 \mathbf{G}_m^2 \Gamma \right) \\
&\overset{1)}{=} -\frac{1}{4} \min\{\eta_0, p_m \eta_m\} \mathbb{E} \left\| \nabla f\left(w_0^t, \mathbf{w}^t\right) \right\|^2 + Q_1 + Q_2
\end{aligned}
\tag{52}
$$

282 where 1) mark the second line as $Q_2$ for notation brevity.

283 For c),

$$
\begin{aligned}
&\mathbb{E}_{m_t, i, u} \left\| \mathbf{w}^{t+1} - \mathbf{w}^t \right\|^2 \\
&\overset{1)}{=} \mathbb{E}_{m_t, i, u} \left\| \eta_{m_t} \left[ \hat{\nabla}_{h_{m_t}} \hat{f}_i \left( w_0^t, \tilde{\Phi}(w_{m_t}^t) \right) + \gamma^t \right] \nabla_{w_{m_t}} h_{m_t}(w_{m_t}^t; x_{m,i}) \right\|^2 \\
&\overset{2)}{\leq} 2 \sum_{m=1}^{M} p_m \eta_m^2 d_{h_m} \mathbf{G}_0^2 \mathbf{G}_m^2 + \frac{1}{2} \sum_{m=1}^{M} p_m \eta_m^2 \mu_m^2 L_m^2 d_{h_m}^2 \mathbf{G}_m^2 + \sum_{m=1}^{M} p_m \eta_m^2 \mathbf{G}_m^2 \Gamma
\end{aligned}
\tag{53}
$$

284 where 1) the update rule for the communication round, 2) applies the exactly same procedures in
285 Eq. 43 and applies assumption B.7 (independent client).

## D.7 Bound the Gradient $\nabla f\left(w_0^t, \mathbf{w}^t\right)$

287 Start with Eq. 52:

$$
\begin{aligned}
&\mathbb{E}\left( M^{t+1} - M^t \right) \\
&\leq -\frac{1}{4} \min\{\eta_0, p_m \eta_m\} \mathbb{E} \left\| \nabla f\left(w_0^t, \mathbf{w}^t\right) \right\|^2 + Q_1 + Q_2
\end{aligned}
\tag{54}
$$

288 Summing over the global iteration $t = 0, 1, \ldots T-1$, arrange the equation and divided it by $T$ from
289 both sides.

$$\frac{1}{4T} \min\{\eta_0, p_m \eta_m\} \sum_{t=0}^{T-1} \mathbb{E} \left\| \nabla f\left(w_0^t, \mathbf{w}^t\right) \right\|^2$$

$$\leq \frac{\mathbb{E}\left(M^0 - M^T\right)}{T} + Q_1 + Q_2$$

$$\overset{1)}{\leq} \frac{\mathbb{E}\left(f^0 - f^*\right)}{T} + Q_1 + Q_2$$

(55)

where 1) $\mathbb{E}\left(M^0 - M^T\right) = f\left(w_0^0, \mathbf{w}^0\right) - f\left(w_0^T, \mathbf{w}^T\right) - \sum_{i=1}^{\tau} \theta_i \left\|\mathbf{w}^{T-i} - \mathbf{w}^{T-i}\right\|^2 \leq f\left(w_0^0, \mathbf{w}^0\right) - f\left(w_0^T, \mathbf{w}^T\right) \leq f^0 - f^*$, we use $f^0$ to denote $f\left(w_0^0, \mathbf{w}^0\right)$ and applying assumption B.1.

Dividing $\zeta = \frac{1}{4} \min\{\eta_0, p_m \eta_m\}$ from both sides:

$$\frac{1}{T}\sum_{t=0}^{T-1} \mathbb{E}\left\|\nabla f\left(w_0^t, \mathbf{w}^t\right)\right\|^2$$

$$\leq \frac{\mathbb{E}\left(f^0 - f^*\right)}{T\zeta} + \frac{Q_1}{\zeta} + \frac{Q_2}{\zeta}$$

$$\leq \frac{\mathbb{E}\left(f^0 - f^*\right)}{T\zeta}$$

$$+ \frac{1}{\zeta}\left(\eta_0 H_0^2 + 2L\eta_0^2 H_0^2 + 4\sum_{m=1}^{M} p_m \eta_m \mathbf{G}_m^2 H_m^2\right)\mathcal{E} + \left(4\sum_{m=1}^{M} p_m \eta_m \mathbf{G}_m^2 + L\sum_{m=1}^{M} p_m \eta_m^2 \mathbf{G}_m^2\right)\Gamma$$

$$+ \frac{1}{4\zeta}\sum_{m=1}^{M} p_m \eta_m \mu_m^2 L_m^2 d_{h_m}^2 \mathbf{G}_m^2 + \frac{2}{\zeta}\sum_{m=1}^{M} p_m L\eta_m^2 d_{h_m} \mathbf{G}_0^2 \mathbf{G}_m^2 + \frac{1}{2\zeta}\sum_{m=1}^{M} p_m L\eta_m^2 \mu_m^2 L_m^2 d_{h_m}^2 \mathbf{G}_m^2$$

$$+ \frac{1}{\zeta} L\eta_0^2 \sigma_0^2$$

$$+ \frac{\tau}{\zeta}\left(\eta_0 L_0^2 + 2L\eta_0^2 L_0^2 + 2\sum_{m=1}^{M} p_m \eta_m L_m^2\right)\left(2\sum_{m=1}^{M} p_m \eta_m^2 d_{h_m} \mathbf{G}_0^2 \mathbf{G}_m^2\right.$$

$$\left.+ \frac{1}{2}\sum_{m=1}^{M} p_m \eta_m^2 \mu_m^2 L_m^2 d_{h_m}^2 \mathbf{G}_m^2 + \sum_{m=1}^{M} p_m \eta_m^2 \mathbf{G}_m^2 \Gamma\right)$$

(56)

where 1) plugging in $Q_1$.

To simplify the result, let $L_* = \max_m\{L, L_0, L_m\}$, $\eta_0 = \eta_m = \eta \leq \frac{1}{4L_*}$, $\frac{1}{p_*} = \min_m p_m$, $\mu_* = \max_m\{\mu_m\}$, $d_* = \max_m\{d_{h_m}\}$, $\mathbf{G}_* = \max_m\{\mathbf{G}_0, \mathbf{G}_m\}$, $H_* = \max_m\{H_0, H_m\}$, then $\zeta = \frac{1}{4}\min\{\eta_0, p_m\eta_m\} = \frac{\eta}{4p_*}$. Eq. 56 can be further simplified:

$$\frac{1}{T}\sum_{t=0}^{T-1} \mathbb{E}\left\|\nabla f\left(w_0^t, \mathbf{w}^t\right)\right\|^2$$

$$\overset{1)}{\leq} \frac{4p_* \mathbb{E}\left(f^0 - f^*\right)}{T\eta}$$

$$+ 4p_*\left(H_*^2 + 2L\eta H_*^2 + 4\sum_{m=1}^{M} p_m \mathbf{G}_*^2 H_*^2\right)\mathcal{E} + 4p_*\left(4\sum_{m=1}^{M} p_m \mathbf{G}_*^2 + L\sum_{m=1}^{M} p_m \eta \mathbf{G}_*^2\right)\Gamma$$

$$+ p_*\sum_{m=1}^{M} p_m \mu_*^2 L_*^2 d_*^2 \mathbf{G}_*^2 + 8p_*\sum_{m=1}^{M} p_m L\eta d_* \mathbf{G}_*^4 + 2p_*\sum_{m=1}^{M} p_m L\eta \mu_*^2 L_*^2 d_*^2 \mathbf{G}_*^2$$

$$+ 4p_* L\eta \sigma_0^2$$

$$+ 4p_*\tau\left(L_*^2 + 2L\eta L_*^2 + 2\sum_{m=1}^{M} p_m L_*^2\right)\left(2\sum_{m=1}^{M} p_m \eta^2 d_* \mathbf{G}_*^4 + \frac{1}{2}\sum_{m=1}^{M} p_m \eta^2 \mu_*^2 L_*^2 d_*^2 \mathbf{G}_*^2 + \sum_{m=1}^{M} p_m \eta^2 \mathbf{G}_*^2 \Gamma\right)$$

$$\overset{2)}{\leq} \frac{4p_* \mathbb{E}\left(f^0 - f^*\right)}{T\eta}$$

$$
\begin{aligned}
&+ 4p_* \left( H_*^2 + 2L\eta H_*^2 + 4\mathbf{G}_*^2 H_*^2 \right) \mathcal{E} + 4p_* \left( 4\mathbf{G}_*^2 + L\eta \mathbf{G}_*^2 \right) \Gamma \\
&+ p_* \mu_*^2 L_*^2 d_*^2 \mathbf{G}_*^2 + 8p_* L\eta d_* \mathbf{G}_*^4 + 2p_* L\eta \mu_*^2 L_*^2 d_*^2 \mathbf{G}_*^2 \\
&+ 4p_* L\eta \sigma_0^2 \\
&+ 4p_* \tau \left( L_*^2 + 2L\eta L_*^2 + 2L_*^2 \right) \left( 2\eta^2 d_* \mathbf{G}_*^4 + \frac{1}{2} \eta^2 \mu_*^2 L_*^2 d_*^2 \mathbf{G}_*^2 + \eta^2 \mathbf{G}_*^2 \Gamma \right) \\
\overset{3)}{\leq} & \frac{4p_* \mathbb{E} \left( f^0 - f^* \right)}{T\eta} \\
&+ p_* \left( 6H_*^2 + 16\mathbf{G}_*^2 H_*^2 \right) \mathcal{E} + 17p_* \mathbf{G}_*^2 \Gamma \\
&+ p_* \mu_*^2 L_*^2 d_*^2 \mathbf{G}_*^2 + 8p_* L\eta d_* \mathbf{G}_*^4 + 2p_* \eta \mu_*^2 L_*^3 d_*^2 \mathbf{G}_*^2 \\
&+ 4p_* L\eta \sigma_0^2 \\
&+ 28p_* \tau \eta^2 d_* \mathbf{G}_*^4 + 7p_* \tau \eta^2 \mu_*^2 L_*^4 d_*^2 \mathbf{G}_*^2 + 14p_* \tau L_*^2 \eta^2 \mathbf{G}_*^2 \Gamma \\
\overset{4)}{\leq} & \frac{4p_* \mathbb{E} \left( f^0 - f^* \right)}{T\eta} \\
&+ \eta \left( 8p_* L d_* \mathbf{G}_*^4 + 2p_* \mu_*^2 L_*^3 d_*^2 \mathbf{G}_*^2 + 4p_* L \sigma_0^2 \right) \\
&+ \eta^2 \left( 28p_* \tau d_* \mathbf{G}_*^4 + 7p_* \tau \mu_*^2 L_*^4 d_*^2 \mathbf{G}_*^2 + 14p_* \tau L_*^2 \mathbf{G}_*^2 \Gamma \right) \\
&+ \mu_*^2 \left( p_* L_*^2 d_*^2 \mathbf{G}_*^2 \right) \\
&+ \mathcal{E} \left( 6p_* H_*^2 + 16p_* \mathbf{G}_*^2 H_*^2 \right) \\
&+ \Gamma \left( 17p_* \mathbf{G}_*^2 \right)
\end{aligned}
$$

$$(57)$$

where 1) plugs in the above variables $L_*, \eta, p_*, \zeta, \mu_*$, 2) applies $\sum_{m=1}^M p_m = 1$, 3) simplify by $\eta \leq \frac{1}{4L_*}$, 4) collect $\eta, \mu_*, \mathcal{E}$

Suppose we set $\eta = \frac{1}{\sqrt{T}}$, $\mu_* = \frac{1}{\sqrt{T}}$, and design the compression to make $\mathcal{E} = \mathcal{O} \left( \frac{1}{\sqrt{T}} \right)$ and $\Gamma = \mathcal{O} \left( \frac{1}{\sqrt{T}} \right)$ [2] the above equation becomes

$$
\begin{aligned}
&\frac{1}{T} \sum_{t=0}^{T-1} \mathbb{E} \left\| \nabla f \left( w_0^t, \mathbf{w}^t \right) \right\|^2 \\
\leq & \frac{1}{\sqrt{T}} \left( 4p_* \mathbb{E} \left( f^0 - f^* \right) + 8p_* L d_* \mathbf{G}_*^4 + 4p_* L \sigma_0^2 + 6p_* H_*^2 + 16p_* \mathbf{G}_*^2 H_*^2 + 17p_* \mathbf{G}_*^2 \right) \\
&+ \frac{1}{T} \left( 28p_* \tau d_* \mathbf{G}_*^4 + p_* L_*^2 d_*^2 \mathbf{G}_*^2 \right) \\
&+ \frac{1}{T^{\frac{3}{2}}} \left( 2p_* L_*^3 d_*^2 \mathbf{G}_*^2 + 14p_* \tau L_*^2 \mathbf{G}_*^2 \right) \\
&+ \frac{1}{T^2} \left( 7p_* \tau \mu_*^2 L_*^4 d_*^2 \mathbf{G}_*^2 \right)
\end{aligned}
$$

$$(58)$$

Therefore,

$$
\frac{1}{T} \sum_{t=0}^{T-1} \mathbb{E} \left\| \nabla f \left( w_0^t, \mathbf{w}^t \right) \right\|^2 = \mathcal{O} \left( \frac{d_h}{\sqrt{T}} \right) \tag{59}
$$

where $d_h = d_* = \max_m \{ d_{h_m} \}$ (for clear notation), $T$ is the number of communication rounds.

The proof of Theorem 5.2 is complete. ∎

---

[2]Refer to C-VFL [4] about how to design the compression to achieve the compression errors of $\mathcal{O}(\frac{1}{\sqrt{T}})$.

 # E    Experiment Details and Extra Experiments

## E.1    Experiment Details

**Experiment Hardware and Software**    The experiments were conducted on a Linux server with Intel(R) Xeon(R) Silver 4114 CPU @ 2.20GHz and the experiment is run on one Nvidia Tesla P100 graphic card. PyTorch was used as the deep learning framework. We re-implement the framework by ourselves because all of the frameworks [18, 4, 5, 21] we compared were not open-source, and re-implementing the code helped make a fair comparison which eliminated the differences in implementation details of various methods.

**Feature Splitting Details**    Regarding the dist-MNIST experiment in Section 6, we flattened the image and then equally distributed the dimensions among each client. Specifically, the first client received the upper half of each image, while the second client was allocated the lower half.

Regarding the dist-CIFAR-10 experiment in Section 6 (and section E.3 in this Appendix), we split the image by the last dimension. Therefore, the first client was assigned the left half, while the second client received the right half of each image.

**Syn-ZOO-VFL**

---

**Algorithm 1** The Synchronous Modification of ZOO-VFL [21]

---

0: Initialize variables for workers $m \in [M]$
1: **for** $t = 0, ..., T - 1$ **do**
2:     Random sample a sample $i$ (or batch $B$).
3:     **for** client $m$ in $[M]$ in parallel **do**
4:         Client $m$ compute and send $h_{m,i} = h_m(w_m; x_{m,i})$ and $\hat{h}_{m,i} = h_m(w_m + \mu\mathbf{u}_{m,i}; x_{m,i})$ to the server.
5:         The server calculates $\delta_m = f_i(w_0, ...\hat{h}_{m,i}...) - f_i(w_0, h_{1,i}, ...h_{M,i})$ and send back to the client.
6:         Client $m$ calculate the stochastic gradient w.r.t. its local parameter $w_m$ with the $\delta_m$ received from the server: $\hat{\nabla}_{w_m} f_i(\cdot) = \frac{\phi(d_m)}{\mu} \delta_m \mathbf{u}_{m,i}$
7:         Client $m$ update its parameter with gradient descent $w_m \leftarrow w_m - \eta_m \hat{\nabla}_{w_m} f_i(\cdot)$
8:     **end for**
9:     The server calculates its local stochastic gradient estimation via $\hat{\nabla}_{w_0} f_i(\cdot) = \frac{\phi(d_0)}{\mu} \left[ f_i(w_0 + \mu\mathbf{u}_{0,i}, ...\hat{h}_{m,i}...) - f_i(w_0, h_{1,i}, ...h_{M,i}; y_i) \right] \mathbf{u}_{0,i}$
10:     The server update its local parameter with gradient descent $w_0 \leftarrow w_0 - \eta_0 \hat{\nabla}_{w_0} f_i(\cdot)$
11: **end for**

---

## E.2    Computation Cost on Extra Propagation on the Server

Our method has extra computation cost on the server compared with other methods, however, the difference is negligible given the powerful computation performance of the server.

We repeat the experiment on dist-MNIST with the default setting (2 clients). To make the result more obvious, we **disable the GPU** to conduct this experiment, and we record the computational time as an index of the computational cost. We assume that the network latency is the same for all frameworks, and ignoring other minor operations in the implementation. The major factor which influences the computation cost is the propagation through the network.

The table below shows a comparison of the computation cost between different frameworks. Letter "F" means forward propagation, "B" means backward propagation, and the numeral preceding the letter indicates the number of propagations, for all frameworks, we only count the propagation time.

Table 2: Computational Cost for Extra Propagation

| Framework | Client | Server | Client Comp. Time per Epoch (s) | Server Comp. per Epoch (s) |
|---|---|---|---|---|
| Split learning [18] | F+B | F+B | 0.86 | 0.90 |
| Syn-ZOO-VFL | 2F | 3F | 0.64 | 1.00 |
| Compressed-VFL [4] | F+B | F+B | 0.86 | 0.89 |
| VAFL [5] | 2F | F+B | 1.10 | 1.52 |
| ZOO-VFL [21] | 2F | 3F | 0.92 | 1.49 |
| VAFL[5]+DP[3] | F+B | F+B | 1.10 | 1.52 |
| Ours | F+B | 101F+B | 1.15 | 49.02 |

### E.3 Dist-CIFAR-10 Experiments

### E.3.1 Comparing with SOTA Frameworks

Following the training procedure outlined in section 6, we utilized the optimal configuration across all frameworks. Table 3 presents a summary of the test accuracy and communication metrics at various stages of convergence. Our achieved test accuracy is comparable to the SOTA VFL methodology. Furthermore, our communication costs are significantly lower than those reported by the leading VFL communication efficiency research. In contrast, the pure ZOO-based VFL is unable to attain convergence to a practical model due to the large dimensionality of the model for optimization.

Table 3: Test Accuracy and Evaluation of the Total Communication Cost.

| | Privacy Security | Test Accuracy | Cost (80%) | Cost (total) |
|---|---|---|---|---|
| Split learning [18] | ✗ | $84.31 \pm 0.28$ | 107 MB | 381 MB |
| Compressed-VFL [4] | ✗ | $84.10 \pm 0.18$ | 67 MB | 240 MB |
| VAFL [5] | ✗ | $83.16 \pm 0.03$ | 184 MB | 400 MB |
| Syn-ZOO-VFL | ✓ | $18.08 \pm 0.33$ | - | - |
| ZOO-VFL [21] | ✓ | $17.96 \pm 0.92$ | - | - |
| Ours | ✓ | $82.82 \pm 0.29$ | 21 MB | 45 MB |

(-) represents that the model cannot converge to a usable model after the entire training process.

Figure 2 illustrates a plot of the training accuracy against epoch (Figure 2-a) and communication cost (Figure 2-b). As depicted in (a), our framework exhibits a convergence rate comparable to that of other frameworks. Specifically, regarding the communication cost, as indicated in (b), our communication cost is significantly lower than that of other communication-efficient algorithms.

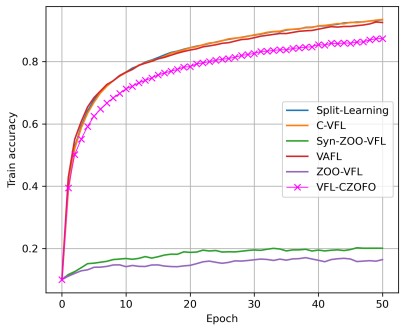

(a) Dist-CIFAR-10 by epochs

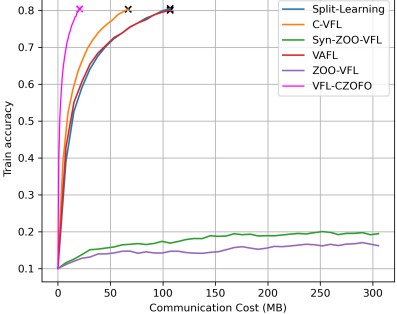

(b) Dist-CIFAR-10 by comm. cost

Figure 2: Comparing with other VFL Framework on Dist-CIFAR10

 The cross means that the training accuracy reaches 80%.

### E.3.2 Dist-CIFAR-10 Ablation Study

**Ablation Study on Zeroth Order Optimization** Figure 3 performs an ablation study on the application of ZOO on the connection layer. We implemented the Avg-RandGradEst using various sampling times $q$. The results indicate that exclusively applying ZOO yields communication costs comparable to those of FOO-based VFL in each communication budget.

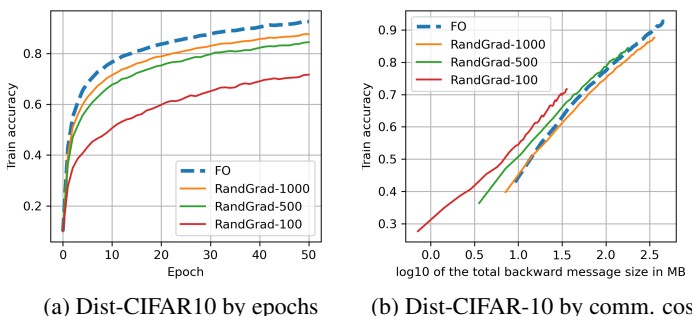

(a) Dist-CIFAR10 by epochs   (b) Dist-CIFAR-10 by comm. cost

Figure 3: Ablation Study on ZO with Dist-CIFAR-10

Table 4 presents the test accuracy of the method and the total backward cost of implementing ZOO on the output layer. The table shows that the application of ZOO decreases the total communication required for the entire training process. With all the sampling times $q$ provided in the table, communication costs are reduced with a slight utility trade-off.

Table 4: Ablation Study on ZO with Dist-CIFAR-10

| ZO Type | Test Accuracy | Backward Cost |
|---|---|---|
| FO | $83.16 \pm 0.03$ | 200 MB |
| RandGradEst-1000 | $82.10 \pm 0.28$ | 156 MB |
| RandGradEst-500 | $81.28 \pm 0.17$ | 78 MB |
| RandGradEst-100 | $72.83 \pm 0.20$ | 16 MB |

**Ablation Study on Compression** Figure 4 displays the results of the ablation study on communication for both forward and backward messages. The plot represents the training accuracy against the communication cost. The results indicate that the utilization of a certain degree of compression (8, 4, 2 bits) led to a reduction in communication costs without significantly affecting the convergence of the model.

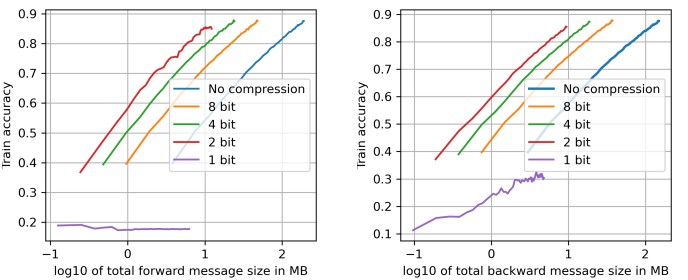

(a) Dist-CIFAR-10 by forward cost   (b) Dist-CIFAR-10 by backward cost

Figure 4: Ablation Study on Compression with Dist-CIFAR-10

Table 5 presents the test accuracy and total communication cost for compression on forward and backward messages. The results suggest that compressing to a certain degree does not significantly impact test accuracy, but it considerably reduces the communication cost. Therefore, we may implement compression to a certain level, such as 4 bits for both the forward and backward messages.

Table 5: Ablation Study on Compression with Dist-CIFAR-10

| Compression | Test Accuracy | Forward/Backward Cost |
|---|---|---|
| No Compression | $82.10 \pm 0.28$ | 200 MB |
| Forward-8 bit | $81.90 \pm 0.24$ | 50 MB |
| Forward-4 bit | $82.64 \pm 0.29$ | 25 MB |
| Forward-2 bit | $81.55 \pm 0.23$ | 13 MB |
| Forward-1 bit | $17.68 \pm 0.70$ | 6 MB |
| Backward-8 bit | $82.03 \pm 0.25$ | 39 MB |
| Backward-4 bit | $82.31 \pm 0.30$ | 20 MB |
| Backward-2 bit | $81.15 \pm 0.34$ | 10 MB |
| Backward-1 bit | $31.03 \pm 0.67$ | 5 MB |

## E.4 Experiments on GiveMeSomeCredit Dataset

To simulate a real-world VFL scenario, we utilize the GiveMeSomeCredit dataset [2]. This dataset comprises 15,000 samples, each consisting of a single label and 10 features. The first client was assigned the first 5 features for each sample, while the second client received the remaining 5 features. Given the dataset's significant class imbalance, we address this issue by downsampling the majority (negative) class to achieve an equal size with the positive class. Subsequently, we divide the dataset into a 75% training set and a 25% testing set. we employ a straightforward linear model ($y = Wx$) on the client side. This model takes the local features of the client as input and generates two predictions: one for the positive class and another for the negative class. We set the batch size to 64 during training, and the model undergoes 100 epochs. The learning rate is chosen as 0.01 from the option of [0.1, 0.01, 0.001]. Additionally, we select the value of $\mu$ as 0.001 from the options [0.1, 0.001, 0.0001, 0.00001] through preliminary experiments. We set the sampling time $q = 10$ for our framework. The experiment results for different methods' test accuracy and the communication cost is shown in table 6. As demonstrated in the table our method significantly reduces the communication cost of training.

Table 6: Test Accuracy and Evaluation of the Total Communication Cost.

| | Test Accuracy | Cost (70%) | Cost (total) |
|---|---|---|---|
| Split learning [18] | $72.18 \pm 0.01$ | 5.7 MB | 38.3 MB |
| Compressed-VFL [4] | $72.13 \pm 0.03$ | 3.8 MB | 24.1 MB |
| VAFL [5] | $72.26 \pm 0.29$ | 5.4 MB | 38.3 MB |
| Syn-ZOO-VFL | $71.74 \pm 0.53$ | 9.2 MB | 38.3 MB |
| ZOO-VFL [21] | $71.85 \pm 0.70$ | 4.6 MB | 38.6 MB |
| Ours | $72.76 \pm 0.29$ | 0.7 MB | 5.8 MB |

Figure 5 displays the corresponding convergence of all the frameworks, The figure shows that while all the frameworks converge similarly, our approach notably reduces the communication cost for each epoch.

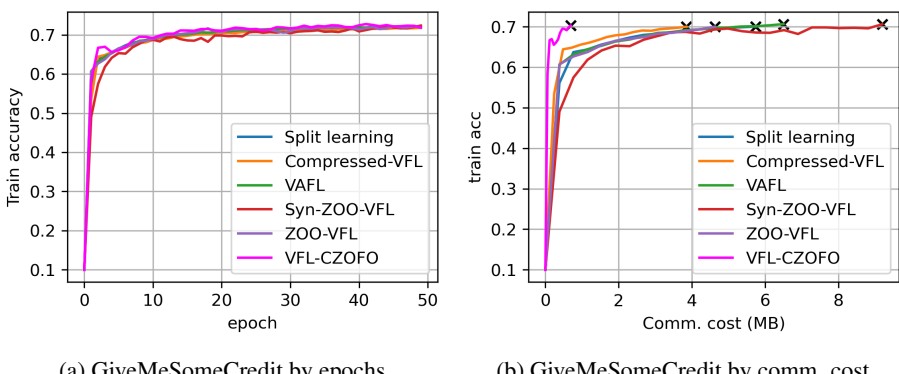

(a) GiveMeSomeCredit by epochs

(b) GiveMeSomeCredit by comm. cost

Figure 5: Comparing with other VFL Framework on GiveMeSomeCredit Task

### E.5 Experiments on a9a Dataset

The a9a dataset [20, 1] encompasses a total of 32,561 training samples and 16,281 testing samples. Each sample has one label and 123 features. In our experiment, the first client was assigned the first 62 features of each sample, while the second client received the remaining 61 features. Our approach employs a linear model similar to the one presented in section E.4. Specifically, client 1's model has an input size of 62, whereas client 2's model has an input size of 61. Both models have an output size of 2. The training procedure is the same as the experiment in section E.4. We set the batch size to 64 during training, and the model is trained 100 epochs. The learning rate is chosen as 0.01. Additionally, we select the value of $\mu$ as 0.001. We set the sampling time $q = 10$ for our framework. The experiment results for different methods' test accuracy and the communication cost is shown in table 7. As demonstrated in the table our method significantly reduces the communication cost of training.

Table 7: Test Accuracy and Evaluation of the Total Communication Cost.

|                     | Test Accuracy   | Cost (82%) | Cost (total) |
|---------------------|-----------------|------------|--------------|
| Split learning [18] | $84.84 \pm 0.01$ | 2.0 MB     | 99.4 MB      |
| Compressed-VFL [4]  | $84.85 \pm 0.02$ | 1.2 MB     | 62.5 MB      |
| VAFL [5]            | $85.08 \pm 0.01$ | 2.0 MB     | 99.4 MB      |
| Syn-ZOO-VFL         | $84.55 \pm 0.05$ | 10.0 MB    | 99.6 MB      |
| ZOO-VFL [21]        | $84.84 \pm 0.01$ | 2.0 MB     | 100.1 MB     |
| Ours                | $84.86 \pm 0.01$ | 0.3 MB     | 14.9 MB      |

Figure 6 displays the corresponding convergence of all the frameworks, The figure shows that while all the frameworks converge almost identically (with the exception of Syn-ZOO-VFL, whose lines do not overlap), our approach notably reduces the communication cost for each epoch.

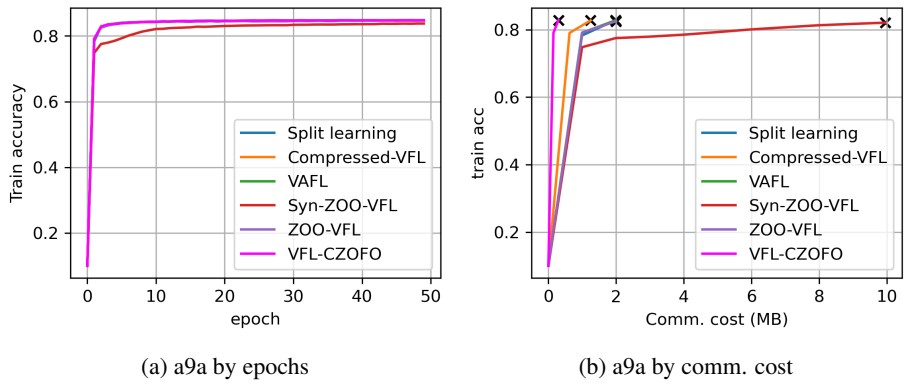

(a) a9a by epochs          (b) a9a by comm. cost

Figure 6: Comparing with other VFL Framework on a9a Dataset

## E.6 Experiment on the Privacy-utility Trade-off

In the experiments in section 6.2 and section 6.3, we only demonstrated a typical trade-off of our framework. However, our framework also has the capability to achieve the same test accuracy as Split-Learning, compressed-VFL, and VAFL, applying a corresponding privacy-utility trade-off. Note that these three baselines sacrifice privacy and get higher test accuracy ("test-accuracy-focused trade-off"), while the last three baselines (VAFL+DP, Syn-ZOO-VFL, ZOO-VFL) take a balance between privacy and utility ("balanced trade-off").

To achieve the test-accuracy-focused trade-off, we use the coordinate-wise gradient estimator (Coord-GradEst) to separately estimate the partial for each dimension[14, 11]:

$$\hat{\nabla}_{h_{m,i}} f_i\left(w_0, h_{1,i}, \cdots, h_{M,i}\right) = \frac{1}{2\mu_m} \sum_{l=1}^{d_{h_m}} [\underbrace{f_i\left(h_{m,i} + \mu_m e_m^l\right) - f_i\left(h_{m,i} - \mu_m e_m^l\right)}_{\delta_{m,i}^l}] e_m^l$$

where $e_m^l \in \mathcal{R}^{d_{h_m}}$ is a $d_{h_m}$-dimensional standard bias vector with 1 at its $l$-th dimension, and 0s otherwise. To apply the coordinate-wise estimation, the server sends $\{\delta_{m,i}^l\}_{l=1}^{d_{h_m}}$ back to the client. It is noteworthy that the backward message $\{\delta_{m,i}^l\}_{l=1}^{d_{h_m}}$ has the same size as $\frac{\partial f_i}{\partial h_{m,i}}$. Both are vectors of decimal numbers with dimensions of $d_{h_m}$. Therefore, if neither method uses compression, the communication cost for VAFL-CZOFO (CoordGradEst) is identical to that of VAFL.

Besides, regarding the "balanced trade-off", the basic zeroth-order estimator (ZOE) we used in section 6 has a large forward bias. To improve this, we applied a slightly "advanced" centralized version of ZOE so that we reached higher test accuracy and better convergence:

$$\hat{\nabla}_{h_{m,i}} f_i\left(w_0, h_{1,i}, \cdots, h_{M,i}\right) = \frac{\phi(d_{h_m})}{q\mu_m} \sum_{j=1}^{q} [\underbrace{f_i(h_{m,i} + \mu_m u_{m,i}^j) - f_i\left(h_{m,i} - \mu_m u_{m,i}^j\right)}_{\delta_{m,i}^j}] u_{m,i}^j$$

With this centralized ZOE, we can achieve a smoother convergence and a similar privacy budget.

Table 8 illustrates our method's capacity to achieve diverse privacy-utility trade-offs when compared to the baselines. In each scenario, our framework successfully achieves the specified privacy budget while maintaining a test accuracy similar to that of the baselines.

Table 8: Privacy-utility Trade-off of VFL-CZOFO

|  | Privacy | Trade-off type | Test Accuracy |
|---|---|---|---|
| VAFL | ✗ | Test-accuracy-focused trade-off | $97.36 \pm 0.14$ |
| VFL-CZOFO (CoordGradEst) | ✗ | Test-accuracy-focused trade-off | $97.35 \pm 0.05$ |
| VAFL + DP | $\epsilon = 95$ | Balanced trade-off | $95.94 \pm 0.29$ |
| VFL-CZOFO (Avg-RandGradEST) | $\epsilon = 95$ | Balanced trade-off | $96.32 \pm 0.22$ |

### E.7 Experiments on More Clients

In section 6 of the paper, we only consider a typical scenario with only two clients. Therefore, we conducted experiments with four and eight clients to further assess the performance of our framework on a larger scale.

The dataset spliting setting of the experiments follows the dist-MNIST experiment. For the experiment involving four clients in section, the first client received the uppermost 1/4 of each image; the second client obtained the segment spanning from the upper 1/4 to 1/2; the third client from the lower 1/2 to 3/4; finally, the fourth client was assigned the bottommost 1/4. A similar split was implemented for the experiment involving eight clients.

The models deployed on each client are identical to the one presented in Section 6. Similarly, the server model is described in detail in Section 6. However, it is worth noting that with the number of clients changed, the input size of the first layer of the server has been adjusted to $4 \times 64 = 256$ for the 4-client experiments and $8 \times 64 = 512$ for the 8-client experiments.

#### E.7.1 Training Efficiency and Communication Cost

We conducted the same experiment on training efficiency and communication cost as in section 6.3. The outcomes for four clients are depicted in Figure 7 and detailed in Table 9. Similarly, the outcomes for eight clients are presented in Figure 8 and detailed in Table 10. These results collectively substantiate the efficacy of our method in diminishing communication costs, particularly within scenarios involving a higher number of clients.

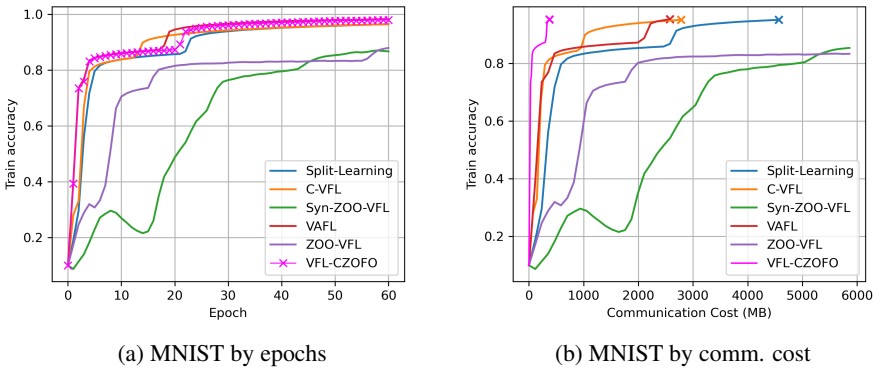

(a) MNIST by epochs        (b) MNIST by comm. cost

Figure 7: Training Efficiency and Communication Cost Experiment on 4 Clients

Table 9: Test Accuracy and Comm. Cost (4 Clients)

|  | Privacy | Test Accuracy | Cost (95%) | Cost (total) |
|---|---|---|---|---|
| Split learning | ✗ | $97.67 \pm 0.03$ | 4570 MB | 11718 MB |
| Compressed-VFL | ✗ | $97.78 \pm 0.12$ | 2783 MB | 7325 MB |
| VAFL | ✗ | $97.60 \pm 0.07$ | 2703 MB | 12288 MB |
| VAFL+DP | ✓ | $96.72 \pm 0.21$ | 3179 MB | 12288 MB |
| Syn-ZOO-VFL | ✓ | $83.97 \pm 0.51$ | - | 11722 MB |
| ZOO-VFL | ✓ | $87.42 \pm 0.25$ | - | 12291 MB |
| Ours | ✓ | $96.60 \pm 0.08$ | 537 MB | 1579 MB |

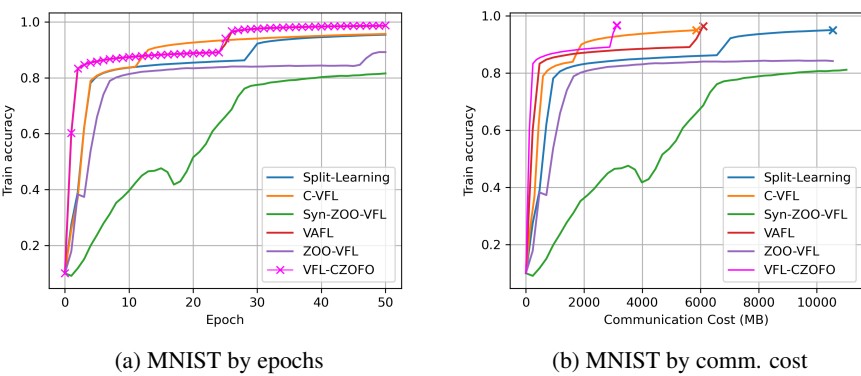

(a) MNIST by epochs       (b) MNIST by comm. cost

Figure 8: Training Efficiency and Communication Cost Experiment on 8 Clients

Table 10: Test Accuracy and Comm. Cost (8 Clients)

|  | Privacy | Test Accuracy | Cost (95%) | Cost (total) |
|---|---|---|---|---|
| Split learning | ✗ | $97.46 \pm 0.08$ | 10547 MB | 23438 MB |
| Compressed-VFL | ✗ | $97.51 \pm 0.09$ | 5860 MB | 14649 MB |
| VAFL | ✗ | $97.41 \pm 0.04$ | 6390 MB | 24579 MB |
| VAFL+DP | ✓ | $96.62 \pm 0.17$ | 8132 MB | 24579 MB |
| Syn-ZOO-VFL | ✓ | $82.64 \pm 0.57$ | - | 23443 MB |
| ZOO-VFL | ✓ | $89.49 \pm 0.38$ | - | 24093 MB |
| Ours | ✓ | $96.81 \pm 0.12$ | 3272 MB | 12590 MB |

### E.7.2 The Computational Cost

With more clients, the server may take more computational costs on the server. Therefore, we also conducted an experiment on the computational cost of the server and the clients. The setting of this experiment follows the experiment in section E.2 but changes the number of clients to four and eight. The result is shown in Table 11.

Table 11: Computational Cost for Propagation (More Clients)

| The number of Clients | Clients' Comp. Time per Epoch (s) | Server's Comp. Time per Epoch (s) |
|---|---|---|
| 2 | 1.15 | 49.02 |
| 4 | 2.65 | 122.20 |
| 8 | 9.81 | 384.29 |