# OpenReview forum: "A Unified Solution for Privacy and Communication Efficiency in Vertical Federated Learning"
_NeurIPS.cc/2023/Conference — NeurIPS 2023 poster_

### Official Review · Reviewer_uLCn · 2023-07-03

**Soundness:** 3 good
**Presentation:** 2 fair
**Contribution:** 2 fair
**Rating:** 6
**Confidence:** 4

**Summary:**

This research proposes a novel cascaded hybrid optimization approach named VFL-CFOZO for VFL that combines zeroth-order (ZO) gradient and first-order (FO) gradient optimization techniques. The critical output layer of the clients uses the ZO gradient, while other parts utilize the FO gradient. This approach enhances convergence while maintaining privacy protection. Experimental results demonstrate that VFL-CZOFO achieves similar utility as the Gaussian Mechanism in DP privacy preservation while significantly reducing communication costs compared to state-of-the-art communication-efficient VFL frameworks.

**Strengths:**

- A novel idea and good motivation. Based on existing research about FOO and ZOO frameworks in VFL, this research focuses on privacy in transmitted parameters and efficiency on other layers to load balance.

- Complete analysis and proofs. Compared to the previous works in ZOO-VFL, such as [40], this research supplies clear proof of DP privacy.

- Moreover, convergence proofs are also sufficient.

**Weaknesses:**

- Lack of scalability. In VFL, the dimension of the dataset often increases with the scale of clients, which means higher sample times should be applied to guarantee convergence. Thus, the current experiments with two clients are insufficient in this aspect. The evaluation of the correlations of client size and “enough” sample times should be added.

- The potential risk of offline clients. The identical random sequence demands that the server and client maintain high synchronization. If a client drops for a while and reconnects with high delay, renegotiating the random seed may be better than using the original random ID because the completeness and timeliness of the information in an offline client are usually uncertain for the server.

- Some minor mistakes. In the explanation of Figure 4 in Section 6.4, the symbol q is miswritten as p. Inconsistent expression of the paper. ‘VAFL-ZOFO’ in Figure 5 of Appendix E.4 but ‘VAFL-CZOFO’ throughout the paper. Table 1 of Appendix A.2 shows that the output embeddings without delay formula should be Φ(w^t) instead of the w^t with the wide tide.

**Questions:**

1. Why not add VAFL+DP to Figure 3 in Section 6.3? This work also incorporates privacy and communication efficiency. Though the paper is compared for training accuracy in 6.2, it needs to be clarified and intuitive enough.

2. Is there any basis or analysis of the application of the chain rule on a cascaded hybrid gradient? [40] indicates that the chain rule is inapplicable between two ZOE gradients due to extra variance. However, this research directly uses it between FO and ZO gradients.

**Limitations:**

As mentioned in Section 7, the extra computational costs for the server do limit the scalability and utility of this work.

---

> ### Author Rebuttal · Authors · 2023-08-09
>
> >Weakness 1 (Experiment with more clients needed):
>
> We conduct experiments with 4 and 8 clients to further assess the performance of our framework on a larger scale. The results are presented in the Figure 1&2, Table 1&2 in the **attached PDF (in the topmost rebuttal box)**. It is essential to note that Vertical Federated Learning (VFL) typically involves a **small number of participants** (e.g. [5, 6, 10, 15, 27, 40] all used less than 5 clients). Unlike Horizontal Federated Learning (HFL), which involves millions of smartphones, tablets, and similar devices as clients, VFL primarily engages big companies and institutions. The added experiments will be included in the appendix of the manuscript.
>
> >Weakness 2 (Synchronization of the random seed):
>
> The synchronized random sequence does not need to be generated streamy (repeatedly) during the entire training process. Note that in Eq. 2, the random sequence $u_{m, i}^j $ **has no superscript for iteration time $T$**. Therefore, the participants only need to negotiate $u_{m, i}^j $ once, rather than repeatedly negotiate it during the entire training. Here we also provide one lazy method to generate the synchronized random sequence without communication, which is directly using the sample ID as the random seed to generate the random sequence.
>
> In our work, we assume that generating the same random sequence with the same seed is feasible and not very expensive. However, further exploration of the engineering aspects related to generating random sequences is **beyond the scope** of this paper.
>
> >Weakness 3 (Minor issues):
>
> Thank you very much for pointing them out! The typo $p$ in Section 6.4 has been corrected to $q$. Figures 5 and 6 in Appendix E are also typos, we corrected them to “CZOFO”. We corrected the symbol for the $\Phi(w^t) $ in the notation table in Appendix A, along with some layout problems in that table.
>
> >Question 1 (Add VAFL+DP in Figure 3 of the manuscript):
>
> We attached the figure where VAFL+DP is added in **Figure 6 of the PDF attached**.
>
> Yes, our primary objective of section 6.2 is **comparing training efficiency and communication cost**, while carefully excluding the influence of other factors. Moreover, it is worth noting that VAFL has better convergence performance compared to "VAFL+DP," as shown in the attached figure, where the latter incurs significantly higher communication costs. Therefore, we compared the VAFL baseline with the optimal convergence rate in that section.
>
> Besides, another fundamental concern is that the algorithm “VAFL[6]+DP[1]“ presented in our paper is **not a published baseline**; rather, it is a customized algorithm designed by us that offers equivalent protection to our proposed framework. Therefore, we did not treat it as a baseline for section 6.2 when we presented our work.
>
> >Question 2 (Basis of Chain ZOE. The discussion of the chain rule in [40]):
>
> To the best of our knowledge, we are the first to propose the chain of different gradient estimators, explore its advantages in VFL, and provide a comprehensive convergence analysis.
>
> In [40], the author means that the **multiplication of two ZOE** introduces extra variance. Specifically, separately estimating the two terms in $\frac{\partial f}{\partial h} \cdot \frac{\partial h}{\partial w}$ with two ZOE will lead to extra variance. Therefore, they directly estimate $ \frac{\partial f}{\partial w}$ with ZOE to avoid this extra variance.
>
> In our framework, we also avoid the multiplication of two ZOE. We solely estimate $\frac{\partial f}{\partial h} $ with ZOE and calculate $\frac{\partial h}{\partial w}$ with FO gradient. As a result, no extra variance is introduced in our framework.
>
> >Limitation (Extra computational cost on the server):
>
> We acknowledge this limitation. However, we have several justifications that this limitation is acceptable in practice. First, the number of clients involved in VFL is typically **quite small**, therefore the increase in the number of clients is controllable. Second, it is worth noting that **the bottleneck of VFL is communication cost** rather than computation cost [5, 27]. Local computations consume significantly less time compared to communication with other participants. Third, the server as the initiator and beneficiary **typically possesses more computational resources**, therefore using the higher computation cost of the server to trade off communication cost/privacy is favorable in the strategy of building VFL.

---

> > ### Comment · Reviewer_uLCn · 2023-08-15
> >
> > Thanks for your reply. Since the paper focuses on VFL, you should clarify how data samples are partitioned for different clients. In the paper, there is only one statement, "The datasets were vertically partitioned among all participants in our experiments. Each client held a portion of the features of each sample." Please give a more detailed explanation about how an image sample is partitioned.

---

> > > ### Author Response · Authors · 2023-08-15
> > > **Detailed Explanation of the Data Partitioning**
> > >
> > > Thank you sincerely for your response and valuable suggestions. We will incorporate additional details regarding the dataset partitioning to enhance the manuscript.
> > >
> > > Regarding the MNIST experiment, we flattened the image and then equally distributed the dimensions among each client. Specifically, in the context of the two-client experiment detailed in Section 6, the first client received the **upper half** of each image, while the second client was allocated the **lower half**. For the experiment involving four clients in the attached PDF, the first client received the **uppermost 1/4** of each image; the second client obtained the segment spanning from the **upper 1/4 to 1/2**; the third client from the **lower 1/2 to 3/4**; finally, the fourth client was assigned the **bottommost 1/4**. A similar split was implemented for the experiment involving eight clients. (This dataset has also been employed in other VFL research [6, 27, 40], where the features are equally distributed among clients.)
> > >
> > > Regarding the CIFAR10 Experiment in Appendix E.3, we split the image by the last dimension. Therefore, the first client was assigned the **left half**, while the second client received the **right half** of each image. (This dataset has also been employed in other VFL research [5, 6, 10], where the features are equally distributed among clients.)
> > >
> > > Regarding the GiveMeSomeCredit experiment in Appendix E.4, each sample comprises 10 distinct features. The first client was assigned **the first 5 features** for each sample, while the second client received **the remaining 5 features**. (This dataset has also been employed in other VFL research [13, 40], where the features are equally distributed among clients.)
> > >
> > > Regarding the a9a dataset experiment in Appendix E.5, each sample comprises 1 label and 123 features. The first client was assigned **the first 62 features** of each sample, while the second client received the **remaining 61 features**. (This dataset has also been employed in other VFL research, [40] and Zhang et al. 2022)
> > >
> > >
> > > ---
> > >
> > > ### Extra Comments on Data Partitioning in VFL:
> > > It is worth noting that certain VFL studies with **specific objectives** may have employed a different data partitioning method, e.g., distributing the features via random selection when studying Graph Neural Networks (Ni et al., 2021), distributing the features unevenly when studying feature unbalanced (Zhang et al. 2022), or using multimodality datasets when studying heterogeneity among parties (Castiglia et al. 2022).
> > >
> > > ### References
> > >  *Castiglia, Timothy, Shiqiang Wang, and Stacy Patterson. "Flexible vertical federated learning with heterogeneous parties." arXiv preprint arXiv:2208.12672 (2022).*
> > >
> > >   *Ni, Xiang, et al. "A vertical federated learning framework for graph convolutional network." arXiv preprint arXiv:2106.11593 (2021).*
> > >
> > >   *Zhang, Jie, et al. "Adaptive vertical federated learning on unbalanced features." IEEE Transactions on Parallel and Distributed Systems 33.12 (2022): 4006-4018.*

---

> > > > ### Comment · Reviewer_uLCn · 2023-08-16
> > > >
> > > > Thanks for your reply.

---

### Official Review · Reviewer_WL8R · 2023-07-06

**Soundness:** 3 good
**Presentation:** 3 good
**Contribution:** 3 good
**Rating:** 5
**Confidence:** 3

**Summary:**

The paper proposes a solution for privacy and communication efficient vertical FL training framework by utilizing zero-order-optimisation (ZOO). Since the convergence of ZOO-based VFL is significantly slower than standard gradient-based VFL, the paper proposes to only implement ZOO on the cut layer.

**Strengths:**

The paper provides a strong motivation and a realistic setup of VFL. It also provides a detailed comparison with the SOTA VFL framework in Section 2 with convergence rate. The method of apply different optimization for different layer is very interesting.



**Weaknesses:**

- The method proposes to use a compressor for communication since the ZOO will generate extra communication cost, but didn’t analyse which compressing method is the best for the method.

- It seems from Fig 2 that the method give lower performance compared to Gaussian.


**Questions:**

- Why choose the uniform scale compressor in particular for the experiment? Have you compared it with other compressors?

- It mentions that the paper proposes an asynchronous VFL, but how does asynchronous work? To finish the forward prop, the model will need to have all the intermediate output from all clients. I'm surprised to see that the synchronous version (Syn-VFL-ZO) perform much worse than asynchronous version (the paper version). Is there any reason why? Also, the synchronous version doesn't seem to use the Avg-RandGradEst.



**Limitations:**

- Only 2 clients for experiments. Especially in this case, ZOO can potentially lower the performance when the number of clients increase.

- Only work on the image datasets, which each client holds half of the image. This particular experimental setup is not realistic in the real-life scenarios.

---

> ### Author Rebuttal · Authors · 2023-08-09
>
> >W1&Q1 (The choice of compressor):
>
> Yes, applying different compressors may lead to different communication costs. We will add an experiment on the Top-K compressor in the appendix to enrich the discussion. However, comparing the performance of different compressors is **not the focus** of our paper. The goal for this part is to show that compression, as a common practice in distributed learning, is compatible with our framework. Therefore, we theoretically proved the convergence of our method with **any compression method**, and only used a basic compressor for demonstration in the experiment.
>
> >W2 (Convergence of Figure 2):
>
> When we did the experiment, we only applied the basic zeroth-order estimator (ZOE) because our primary objective was to demonstrate comparable performance with VAFL+DP while significantly reducing the communication cost. The basic ZOE (Eq. 2) has a large forward bias because the perturbation is only on the function's forward side. Applying a slightly “advanced” centralized version of ZOE, we can reach higher test accuracy and better convergence. More specifically, we convert Eq.2 to its centralized version which reduces the bias by sampling from both sides of the function:
> $ \frac{\phi(d_{h_m}) }{2 q \mu_m } \sum_{j=1}^{q} [{f_i(h_{m, i} + \mu_m u_{m, i}^j )  - f_i ({h_{m, i}}- \mu_m u_{m, i}^j } )] u_{m, i}^j$
>
> With this centralized ZOE, we achieve a smoother convergence, similar privacy budget ($\epsilon=95$), and better test accuracy (96.32%) than the corresponding VAFL+DP of (95.94%). We attached the result in **Figure 4 in the attached PDF (in the topmost rebuttal box)**.
>
> >Q2 (How does asynchronous work? Why is Syn-VFL worse than Asyn-VFL in Figure 3? Why ZOO-VFL does not use Avg-RandGradEst?):
>
> Regarding “how does asynchronous work”: Yes, the server requires embeddings from all clients. Therefore, to support asynchronous updates, the server maintains a table of embeddings from all clients. During each round, only the embedding of the activated client undergoes an update. This specific detail has been explained in lines #238-#240 of the manuscript. Furthermore, a comprehensive explanation of Asynchronous VFL can be found in Appendix D.1 of the manuscript.
>
> Regarding “Syn worse than Asyn”: the primary reason is that the fundamental difference between Asyn-VFL and Syn-VFL causes some ambiguity in measuring the convergence. To address this ambiguity, we will add the definition of the x-axis in the manuscript.
>
> We explain this ambiguity here:
>
> In some other works, the "communication round (**CR**)" may be used to measure the convergence of Asyn-VFL and Syn-VFL (**Figure 5 in the PDF attached** shows the "Syn outperforms Asyn" with the x-axis changed to CR). In one CR, Asyn-VFL updates only **one client** and the server, while Syn-VFL updates **all clients** and the server. This difference in the definition of CR is the reason behind the observed superiority of "Syn" over "Asyn" in this particular case. It is worth noting that, in each communication round, the communication cost for Syn-VFL is greater than that of Asyn-VFL.
>
> In our study, we extend the concept of **“epoch”** in Asyn-VFL, to advocate an intuitive understanding comparable to traditional ML. For Asyn-VFL the number of CR in one “epoch” precisely matches the number of CR required for all clients to traverse the dataset in an ideal no-delay case. As per this definition, the server in Asyn-VFL undergoes more updates on the server compared to its corresponding Syn-VFL within one "epoch". Therefore shows “Syn worse than Asyn”. Note that the communication cost for one “epoch” is the same for Asyn-VFL and the corresponding Syn-VFL, which brings some convenience in the discussion of communication cost.
>
> Regarding "why ZOO-VFL does not use Avg-RandGradEst": The reason is that applying Avg-RandGradEst in ZOO-VFL will **significantly increase its forward communication cost**. To implement Avg-RandGradEst, these frameworks would need to generate $q$ perturbations on its local model and send all of these embeddings to the server, i. e. forwarding $q$ different $h(x+\mu u)$ with different $u$. And this large cost is inevitable. Besides, a more basic consideration is that the original work of ZOO-VFL [40] did not apply Avg-RandGradEst, and we followed the implementation of this baseline.
>
> Our method effectively circumvents the expense associated with multiple sampling because the perturbation is on the layer of clients’ output, i.e. $h(x)+\mu u$, with $q$ different $u$. This random sequence of $u$ can be shared via sharing the random seed, and only need to be generated once. Therefore, in each iteration, our framework only needs to send $h(x)$ instead of $ h(x+\mu u)$, thereby avoiding the communication cost incurred by multiple sampling of Avg-RandGradEst.
>
> >L1 (Experiment on more clients needed):
>
> We conducted experiments with 4 and 8 clients to further assess the performance of our framework on a larger scale. The results are presented in the Figure 1&2, Table 1&2 in the **PDF**. It is essential to note that Vertical Federated Learning (VFL) typically involves a **small number of participants** (e.g. [5, 6, 10, 15, 27, 40] all used less than 5 clients). Unlike Horizontal Federated Learning (HFL), which involves millions of smartphones, tablets, and similar devices as clients, VFL primarily engages big companies and institutions. The added experiments will be included in the appendix of the manuscript.
>
> >L2 (Need experiment in real-life scenarios):
>
> We not only conducted experiments on the mainstream CV dataset (MNIST and CIFAR10) but also on the **real-world dataset** for VFL in the Appendix of the manuscript. Specifically, the experiments conducted on GiveMeSomeCredit and Adult(a9a) datasets can be found in Appendix E.4 and E.5. The results from these experiments demonstrate that we achieve comparable test accuracy to other baseline approaches, while significantly reducing communication costs.

---

> > ### Comment · Reviewer_WL8R · 2023-08-16
> >
> > I extend my gratitude to the authors for their insightful rebuttal and for providing the accompanying new results. The explanation is clear.

---

> > > ### Author Response · Authors · 2023-08-16
> > > **Thank you and further results on different compressors.**
> > >
> > > Thank you sincerely for your response and valuable suggestions.
> > >
> > > We also conducted further experiments incorporating other compressors as you suggested in W1. The table below shows the ablation study of applying Top-K and Random-K (Stich et al. 2018, Shi et al. 2019) on the backward message of VFL-CZOFO. This experiment will be added in the Appendix of the manuscript. In terms of convergence rate, Top-K (K=10) shows similar convergence to Uniform (8-bit), while Random-K (K=10) converges notably slower.  However, the sparsification technique has advantages in enhancing test accuracy, possibly due to preventing overfitting.
> > >
> > > |           | Compressor on Backward Message | Test Accuracy    | Backward Cost (95%) | Backward Cost (total) |
> > > |-----------|--------------------------------|------------------|---------------------|-----------------------|
> > > | VFL-CZOFO | None                           | 95.30 $\pm$ 0.25 | 19 MB               | 75 MB                 |
> > > |           | Uniform (8bit)                 | 94.58 $\pm$ 0.21 | 5 MB                | 19 MB                 |
> > > |           | Top-K (K=10)                   | 96.35 $\pm$ 0.24 | 5 MB                | 15 MB                 |
> > > |           | Random K (K=10)                | 95.56 $\pm$ 0.23 | 8 MB                | 15 MB                 |
> > >
> > >
> > > ### Reference
> > > *Stich, Sebastian U., Jean-Baptiste Cordonnier, and Martin Jaggi. "Sparsified SGD with memory." Advances in Neural Information Processing Systems 31 (2018).*
> > >
> > > *Shi, Shaohuai, et al. "Understanding top-k sparsification in distributed deep learning." arXiv preprint arXiv:1911.08772 (2019).*

---

### Official Review · Reviewer_1JdL · 2023-07-07

**Soundness:** 3 good
**Presentation:** 3 good
**Contribution:** 3 good
**Rating:** 8
**Confidence:** 5

**Summary:**

The paper is aiming at solving two critical problems of Vertical Federated Learning (VFL), the convergence rate of ZOO-based VFL and the privacy guarantee of ZOO-based VFL. This study provides a simple solution of using different optimizations i.e., the first-order optimization (FOO) and zeroth-order optimization method (ZOO) in the VFL framework, and theoretically explained the privacy guarantee with differential privacy. Experiments are conducted with regard to differential privacy, training efficiency, and communication cost which showed significant improvement in communication cost compared with SOTA and baselines.

**Strengths:**

This study discusses the training efficiency and privacy in applying ZOO to VFL, which is a significant problem in this area. The idea of cascaded different optimization is novel and interesting. The motivation of balancing the advantages of FOO&ZOO sounds. I am appreciated that it proposes a simple solution that is easy to understand and will inspire more following studies in this area. The solution is validated from different dimensions and thus I think the effectiveness of the solution is reliable.

The paper is also theoretically contributed. Theorem 4.1 explains the privacy of ZOO. Theorem 5.2 guarantees training efficiency. The experiments are solid from my perspective, comprehensively covering the essential aspects of the algorithm, including the privacy budget of DP, the learning curve, and the communication cost at each stage. The method has high performance with a substantial improvement in communication efficiency, making this work great potential of being a new SOTA baseline.

**Weaknesses:**

According to Table 1, solely applying ZOO on the client layer only reduce the backward message size from $d_h B$ to $q$ compared with FOO-based VFL. However, the forward communication size was not improved. I agree with the improvement of your work in improving the convergence rate, however, a large reduction in the total communication cost could be due to the compression. Could you illustrate more on the contribution of compression and ZOO in reducing the total communication cost?

As you mention in #261-#263, the fundamental difference between ZOO and Gaussian Mechanism is that your privacy budgets ($\epsilon, \delta$) are implicitly controlled by the parameter of ZOO. Will this limit the application of your solution when a certain degree of privacy budget is required?

**Questions:**

It is less clear to me whether the reduction of total communication cost is from “applying ZOO” or “applying compression”. The current ablation study did not tell the ratio of the reduction of the communication cost between “applying ZOO” and “applying compression”.

How your approach solves the problem if a certain privacy budget is required? Since the privacy budget of your framework is not directly controlled by the magnitude of the noise.

**Limitations:**

Yes, the limitation was discussed.

---

> ### Author Rebuttal · Authors · 2023-08-09
>
> >Weakness 1 & Question1 (The contribution of ZOO and compression in reducing communication cost):
>
> Yes, both methods contribute to enhancing communication efficiency. However, typically ZOO makes a more significant contribution to reducing the communication cost.
>
> Take the experiment presented in Table 2 as an example, The application of the ZOO results in a reduction of backward message cost **from 3073MB to 75MB**, achieving a total reduction of **2998MB**.  Subsequently, by using compression, the forward message size diminishes from 3073MB to 769MB, and backward costs additionally reduce from 75MB to 19MB, leading to a total reduction of **2360MB**. For this experiment, the ratio of the absolution contribution between ZOO and compression is around **1.27: 1**.
>
> >Weakness 2 & Question 2 (How to achieve a certain privacy budget):
>
> Yes, that is the fundamental difference between our method and Gaussian Mechanism. If a certain privacy budget ($\epsilon, \delta$) is required. We need to run the parameter tuning process for the ZOO to achieve that privacy budget. For example, if the privacy budget is too large, we need to reduce the sampling time $q$ to make the gradient estimation less accurate or reduce the number of iterations $T$ to make the attacker accumulate less information (early stopping). We acknowledge that this tuning process is less convenient than Gaussian Mechanism where the corresponding magnitude of the Gaussian noise can be directly calculated. However, with our scheme, we achieve DP-guarantee by reducing the communication amount, while Gaussian Mechanism achieves DP-guarantee by adding noisy information.
>
> >Other notes
>
> We also included **additional experiments in the attached PDF** (located in the topmost rebuttal section), containing the experiment involving 4 and 8 clients.

---

> > ### Comment · Reviewer_1JdL · 2023-08-17
> >
> > Thank you very much for your reply and the additional experiment.
> >
> > Apart from those experiments, I am also interested in the privacy-utility trade-off of CZOFO in your discussion with other reviewers. Could you demonstrate on a set of different privacy budgets and their corresponding test accuracy? This would be a good experiment to demonstrate the privacy-utility trade-off of your method.

---

> > > ### Author Response · Authors · 2023-08-18
> > > **Thank you and further experiment result**
> > >
> > > Thank you sincerely for your response and valuable suggestions.
> > >
> > > Following your suggestion, we conducted further experiments on the privacy-utility trade-off across a comprehensive range of privacy budgets. Through this experiment, we demonstrate our framework's versatility in achieving varying trade-offs between privacy budget and test accuracy. The corresponding results are presented in the table provided below. This result will be included in the appendix of the manuscript. It is worth noting that the last column demonstrates that VFL-CZOFO can achieve comparable test accuracy as VAFL, given the corresponding privacy-utility trade-off.
> > >
> > > **Table 1: Privacy Budget and Corresponding Test Accuracy**
> > >
> > > | $\bar{\epsilon}=$ | 12               | 20               | 35               | 95               | >>100            |
> > > |-------------------|------------------|------------------|------------------|------------------|------------------|
> > > | VAFL+DP           | 72.34 $\pm$ 0.59 | 84.17 $\pm$ 2.83 | 93.18 $\pm$ 0.52 | 95.94 $\pm$ 0.29 | 97.36 $\pm$ 0.14 |
> > > | VFL-CZOFO         | 75.92 $\pm$ 3.51 | 85.86 $\pm$ 2.78 | 93.34 $\pm$ 0.15 | 96.32 $\pm$ 0.22 | 97.35 $\pm$ 0.05 |
> > >
> > > ---
> > >
> > > The following section provides the experimental details: First, to ensure consistency with the experiments in the manuscript, we did not use any compressor on VFL-CZOFO. Besides, we further employed the centralized zeroth order estimator to enhance the convergence stability, i.e. converting the right-hand side of Eq.2 into $ \frac{\phi(d_{h_m}) }{2 q \mu_m } \sum_{j=1}^{q} [{f_i(h_{m, i} + \mu_m u_{m, i}^j )  - f_i ({h_{m, i}}- \mu_m u_{m, i}^j } )] u_{m, i}^j $. The $\bar{\delta}$ are tuned to 0.01 for all trials. To achieve different accumulated privacy budget $\bar{\epsilon}$, we reduce the sampling time $q$ of Avg-RandGradEst and reduce the number of iterations $T$ of VFL-CZOFO as mentioned in our response to Weakness 2&Question2.

---

> > > > ### Comment · Reviewer_1JdL · 2023-08-21
> > > > **All concerns are addressed well. I will raise my rating.**
> > > >
> > > > Thank you for your response and additional experiments. The proposed solution has clear motivation and the potential for application across a wide range of scenarios. I will raise my rating and support this paper.

---

### Official Review · Reviewer_Mz1s · 2023-07-25

**Soundness:** 3 good
**Presentation:** 2 fair
**Contribution:** 3 good
**Rating:** 6
**Confidence:** 3

**Summary:**

This paper presents a pioneering Zero-Order Optimization (ZOO)-based VFL algorithm that effectively ensures privacy preservation while significantly enhancing communication efficiency. Regarding privacy, the paper demonstrates theoretically that ZOO can inherently offer $(\epsilon,\delta)$-differential privacy, providing a strong foundation for understanding the privacy preservation achieved by ZOO. Concerning communication efficiency, the method ingeniously combines first-order and zero-order gradient optimization, resulting in remarkable improvements in training and communication efficiency. Moreover, the paper rigorously proves the convergence of the proposed algorithm. Extensive experiments are conducted, further affirming the superiority of the method.

**Strengths:**

1. The paper presents a novel VFL method that applies different optimization methods to different layers of the global model in each iteration, significantly improving the convergence rate of ZOO-based VFL while preserving privacy.
2. The theoretical proof that ZOO can inherently offer $(\epsilon,\delta)$-differential privacy provides a strong foundation for understanding the privacy preservation achieved by ZOO in VFL.
3. This paper conducts extensive experiments, offering concrete evidence of its effectiveness.


**Weaknesses:**

1. The experiments are currently conducted with only two clients, but further experiments with varying numbers of clients are necessary.
2. The article divides experiments on different datasets into multiple chapters for presentation, which hinders a comprehensive display of the model.
3. The computation cost of the server is extremely high, and it escalates with an increase in the number of clients due to the average random gradient estimation for each client.


**Questions:**

1. Does the number of clients significantly affect the computational efficiency?
2. In Appendix E.4 and E.5, Figure 5 and Figure 6 present VFL-ZOFO, which is different from VFL-CZOFO. Do they have any differences?


**Limitations:**

1. The computation cost of the server is extremely high.
2. This method is not suitable for many client situations.

---

> ### Author Rebuttal · Authors · 2023-08-09
>
> >Weakness 1& Question 1& Limitation 2 (Experiment on more clients needed):
>
> We conduct experiments with 4 and 8 clients to further assess the performance of our framework on a larger scale. The results are presented in the Figure 1&2, Table 1&2 in the **attached PDF (in the topmost rebuttal box)**. It is essential to note that Vertical Federated Learning (VFL) typically involves a **small number of participants** (e.g. [5, 6, 10, 15, 27, 40] all used less than 5 clients). Unlike Horizontal Federated Learning (HFL), which involves millions of smartphones, tablets, and similar devices as clients, VFL primarily engages big companies and institutions. The added experiments will be included in the appendix of the manuscript.
>
> >Weakness 2 (Presentation of the experiment):
>
> We aimed to conduct a comprehensive evaluation of our algorithm, putting the experiment for all datasets together could be the optimal choice. However, because of space limitations, we had to put the experiment on other datasets in the appendix. We considered each dataset as a distinct scenario of applying VFL, therefore we separated them into different chapters for clarity.
>
> >Weakness 3 & Limitation 1 (Computational cost of the server):
>
> We acknowledge this limitation, and we also include an experiment on the computational cost on the server with more clients in **Table 4 of the PDF attached** (the experiment setting is the same as Appendix E.2, but changing the number of clients).
>
> However, we have several justifications that this limitation is acceptable in practice. First, the number of clients involved in VFL is typically **quite small**, therefore the increase in the number of clients is controllable. Second, it is worth noting that **the bottleneck of VFL is communication cost** rather than computation cost [5, 27]. Local computations consume significantly less time compared to communication with other participants. Third, the server as the initiator and beneficiary **typically possesses more computational resources**, therefore using the higher computation cost of the server to trade off communication cost/privacy is favorable in the strategy of building VFL.
>
>
> >Q2 (Figure 5&6 in Appendix E):
>
> That is a typo for Figures 5 and 6 in Appendix E, both should be “CZOFO”. Thank you very much for pointing that out!

---

> > ### Comment · Reviewer_Mz1s · 2023-08-16
> >
> > Thanks for your response. I'd like to keep the initial rating.

---

> > > ### Author Response · Authors · 2023-08-18
> > > **Thank you.**
> > >
> > > Thank you sincerely for your commitment of time in evaluating the manuscript and the valuable suggestions to help us improve the quality of the manuscript.

---

### Official Review · Reviewer_wequ · 2023-07-25

**Soundness:** 3 good
**Presentation:** 3 good
**Contribution:** 2 fair
**Rating:** 5
**Confidence:** 3

**Summary:**

The paper introduces a hybrid Federated Learning (FL) framework, named VFL-CZOFO, which aims to provide intrinsic privacy protection while also significantly improving the convergence rate when compared to existing ZOO-based frameworks.

**Strengths:**

1. Faster Convergence: The paper demonstrates that VFL-CZOFO achieves faster convergence compared to other ZOO-based frameworks.

2. Theoretical Solidity: The paper is theoretically sound, with a proof of convergence, and guarantees $(\epsilon, \delta)$-DP (differential privacy).

**Weaknesses:**

The major concern lies in the experimental performance of VFL-CZOFO.

1. The experiments only involve 2 clients, which are considered too small for a conclusive evaluation. The proposed algorithm's performance in a large-scale federated learning system remains unclear.

2. The usage of only 2 datasets and lack of clarity regarding heterogeneity raise concerns about the model's generalizability and applicability in diverse scenarios.

3. The choice of epsilon (around 90) for privacy evaluation seems excessively high, raising doubts about the actual privacy protection offered. Additionally, the training doesn’t seem to converge well at epoch 50 in Fig. 2, potentially affecting the total privacy budget.

4. Table 3 shows that the proposed method sacrifices accuracy compared to the FOO-based method while improving communication cost. This trade-off is questionable, as accuracy is generally more critical in most cases.

**Questions:**

VFL-CZOFO is designed for better privacy protection while reducing communication costs. In the case where VFL-CZOFO guarantees a significantly smaller privacy budget than VAFL+Gaussian, what will the accuracy be?

**Limitations:**

The paper introduces a framework that provides intrinsic privacy protection while also improving communication costs. However, there are weaknesses and questions listed above.

---

> ### Author Rebuttal · Authors · 2023-08-09
>
> >Weakness 1 (Experiment on more clients needed):
>
> We conduct experiments with 4 and 8 clients to further assess the performance of our framework on a larger scale. The results are presented in Figure 1&2, Table 1&2 in the **attached PDF (in the topmost rebuttal box)**. It is essential to note that Vertical Federated Learning (VFL) typically involves a **small number of participants** (e.g. [5, 6, 10, 15, 27, 40] all used less than 5 clients). Unlike Horizontal Federated Learning (HFL), which involves millions of smartphones, tablets, and similar devices as clients, VFL primarily engages big companies and institutions. The added experiments will be included in the appendix of the manuscript.
>
> >Weakness 2 (The model’s generalizability in diverse scenarios):
>
> We not only conducted the experiment on the mainstream CV dataset (MNIST and CIFAR10) but also on the **real-world dataset** for VFL. Specifically, the experiments conducted on GiveMeSomeCredit and Adult(a9a) datasets can be found in Appendix E.4 and E.5 of the manuscript. The results from these experiments demonstrate that we achieve comparable test accuracy to other baseline approaches, while significantly reducing communication costs.
>
> Besides, it is worth noting that there is no data distribution heterogeneity in VFL because, in VFL, each client shares different features of the **same sample**. Other types of heterogeneity in VFL such as system heterogeneity and feature unbalanced, is beyond the scope of our paper, including them will obscure the clarity of our theoretical and experimental result.
>
>
> >Weakness 3 (Privacy budget seems high. Deviation from convergence at epoch 50 in Figure 2.):
>
> The $\epsilon$ in that figure is the accumulated privacy budget for the entire training, demonstrating the guarantee in the **the worst case** in VFL. The worst case means that the attacker can acquire all the messages from the server by colluding all clients, during the entire training procedure of 50 epochs. Besides, the sampling time of 100 is very large in this experiment, whose margin gain for convergence is small. Therefore, the privacy budget demonstrated here is larger than practical. To acquire a smaller $\epsilon$, we can use fewer iterations to reduce the accumulated information or use smaller sampling times to estimate the gradient less accurately.
>
> Regarding the “epoch 50 in Fig. 2”: the basic zeroth-order estimator (ZOE) in Eq.2 has a large forward bias. This is possibly the reason that causes an unstable convergence around epoch 50 in Figure 2. Applying a slightly “advanced” centralized version of ZOE, we can get a more stable convergence and higher test accuracy, with a trade-off of extra computational cost on the server. More specifically, we convert Eq.2 to its centralized version:
> $ \frac{\phi(d_{h_m}) }{2 q \mu_m } \sum_{j=1}^{q} [{f_i(h_{m, i} + \mu_m u_{m, i}^j )  - f_i ({h_{m, i}}- \mu_m u_{m, i}^j } )] u_{m, i}^j$
>
> We add an experiment that combines the above techniques in **Figure 3 of the PDF attached**. Applying early stopping, smaller sampling times, and centralized ZOE, we demonstrate a smaller accumulated privacy budget of $\epsilon=32$, without significantly influencing the convergence. The privacy budget can be further reduced with more techniques. However, it is worth noting that there is a privacy-utility trade-off in DP, a small privacy budget will cause low test accuracy [31, 35]. For example, in [35], when $\epsilon = 100$, they achieve around 90% test accuracy, while $\epsilon=50$, the accuracy drops to around 68%.
>
>
> >Weakness 4 (Sacrifice in test accuracy):
>
> Our framework also has the capability to achieve the same test accuracy as the corresponding VAFL baseline (not sacrificing the test accuracy), applying a corresponding privacy-utility trade-off. Note that the first three baselines sacrifice privacy and get higher test accuracy (“test-accuracy-focused trade-off”), while the last three baselines take a balance between privacy and utility (“balanced trade-off”). In Table 3, we only demonstrated the “balanced trade-off” for our method. However, our framework can also demonstrate the “test-accuracy-focused trade-off”, achieving the comparable test accuracy of the corresponding asynchronous VFL baseline (VAFL [6]). The cost is a larger privacy budget and increasing communication-computation costs. The experiment result is attached in **Table 3 of the PDF attached**.
>
> Besides, regarding the "balanced trade-off", the basic zeroth-order estimator (ZOE) we used has a large forward bias. To correct this, we apply a slightly “advanced” centralized version of ZOE so that we can reach higher test accuracy and better convergence. When we did the experiment in the manuscript, we only applied the basic ZOE because our prime goal was the communication cost, therefore we demonstrated a comparable test accuracy with the target baseline in Table 3. With this centralized ZOE, we can achieve a smoother convergence, a similar privacy budget ($\epsilon=95$), and better test accuracy (96.32%) than the corresponding VAFL+DP of (95.94%). (Experiment result is shown in **Table 3 of PDF attached**.)
>
> >Question (What would be the accuracy if VFL-CZOFO has a smaller privacy budget than VAFL+DP):
>
> The privacy-utility trade-off exists within all DP algorithms. If we achieve a smaller privacy budget than VAFL+Gaussian, it means that we apply a trade-off that places a greater emphasis on privacy, leading to lower utility and less test accuracy. However, the advantage of our framework is that we provide comparable protection as the DP mechanism while reducing the communication cost simultaneously.

---

> > ### Author Response · Authors · 2023-08-18
> > **Thank you and we provide further experiment on Weakness 3&4**
> >
> > Thank you sincerely for your commitment of time and effort in evaluating the manuscript. We would be happy to offer additional clarification if needed.
> >
> > We conducted further experiments on the privacy-utility trade-off across a comprehensive range of privacy budgets. Through this experiment, we demonstrate our framework's versatility in achieving varying trade-offs between privacy budget and test accuracy. The corresponding results are presented in the table provided below. This result will be included in the appendix of the manuscript. It is worth noting that the last column demonstrates that VFL-CZOFO can achieve comparable test accuracy as VAFL, given the corresponding privacy-utility trade-off.
> >
> > **Table 1: Privacy Budget and Corresponding Test Accuracy**
> >
> > | $\bar{\epsilon}=$ | 12               | 20               | 35               | 95               | >>100            |
> > |-------------------|------------------|------------------|------------------|------------------|------------------|
> > | VAFL+DP           | 72.34 $\pm$ 0.59 | 84.17 $\pm$ 2.83 | 93.18 $\pm$ 0.52 | 95.94 $\pm$ 0.29 | 97.36 $\pm$ 0.14 |
> > | VFL-CZOFO         | 75.92 $\pm$ 3.51 | 85.86 $\pm$ 2.78 | 93.34 $\pm$ 0.15 | 96.32 $\pm$ 0.22 | 97.35 $\pm$ 0.05 |
> >
> > ---
> >
> > The following section provides the experimental details: First, to ensure consistency with the experiments in the manuscript, we did not use any compressor on VFL-CZOFO. Besides, we further employed the centralized zeroth order estimator to enhance the convergence stability, i.e. converting Eq.2 into $ \frac{\phi(d_{h_m}) }{2 q \mu_m } \sum_{j=1}^{q} [{f_i(h_{m, i} + \mu_m u_{m, i}^j )  - f_i ({h_{m, i}}- \mu_m u_{m, i}^j } )] u_{m, i}^j$. The $\bar{\delta}$ are tuned to 0.01 for all trials. To achieve different accumulated privacy budget $\bar{\epsilon}$, we reduce the sampling time $q$ of Avg-RandGradEst and reduce the number of iterations $T$ of VFL-CZOFO as mentioned in our response to weakness 3.

---

> ### Comment · Reviewer_wequ · 2023-08-21
>
> Thank you very much for the clarifications and comments. I will raise my score to 5

---

### Author Rebuttal · Authors · 2023-08-09

**We would like to thank the reviewers for dedicating their time and effort to assess the manuscript.**

**Attached is the PDF for the figures and tables.**

---

### Decision · Program_Chairs · 2023-09-21

**Decision:**

Accept (poster)

**Comment:**

A vertical FL method is presented that combines zeroth-order (ZO) gradient (used by the output layer of the clients) and first-order (FO) gradient optimization techniques.  This approach enhances convergence while maintaining privacy protection. Experimental results show that this method is able to achieve similar privacy guarantee as the DP Gaussian mechanism while significantly reducing communication costs compared to other VFL frameworks.  There was a good amount of correspondence and discussion, resulting in all reviews on the positive side.